# Characterization of organic aerosol across the global remote troposphere: A comparison of ATom measurements and global chemistry models

Alma Hodzic[1], Pedro Campuzano-Jost[2,3], Huisheng Bian[4], Mian Chin[4], Peter R. Colarco[4], Douglas A. Day[2,3], Karl D. Froyd[2,8], Bernd Heinold[6], Duseong S. Jo[2,3], Joseph M. Katich[2,8], John K. Kodros[5], Benjamin A. Nault[2,3], Jeffrey R. Pierce[5], Eric Ray[2,8], Jacob Schacht[6], Gregory P. Schill[2,8], Jason C. Schroder[2,3], Joshua P. Schwarz[2,8], Donna T. Sueper[2,3], Ina Tegen[6], Simone Tilmes[1], Kostas Tsigaridis[7,9], Pengfei Yu[8,10], Jose L. Jimenez[2,3]

[1]*National Center for Atmospheric Research, Boulder, CO, USA*

[2]*Cooperative Institute for Research in Environmental Sciences (CIRES), University of Colorado, Boulder, CO, USA*

[3]*Department of Chemistry, University of Colorado, Boulder, CO, USA*

[4]*NASA Goddard Space Flight Center, Greenbelt, MD, USA*

[5]*Department of Atmospheric Science, Colorado State University, Fort Collins, CO, USA*

[6]*Leibniz Institute for Tropospheric Research, Leipzig, Germany*

[7]*Center for Climate Systems Research, Columbia University, New York, NY, USA*

[8]*NOAA Earth System Research Laboratory (ESRL), Chemical Sciences Division, Boulder, CO, USA*

[9]*NASA Goddard Institute for Space Studies, New York, NY, USA*

[10]*Institute for Environmental and Climate Research, Jinan University, Guangzhou, Guangdong, China*

**Key words**: organic aerosol, remote atmosphere, ATom field campaign.

**Abstract**.
The spatial distribution and properties of submicron organic aerosols (OA) are among the
key sources of uncertainty in our understanding of aerosol effects on climate.
Uncertainties are particularly large over remote regions of the free troposphere and
Southern Ocean, where very little data has been available, and where OA predictions from
AeroCom Phase II global models span two to three orders-of-magnitude, greatly
exceeding the model spread over source regions. The (nearly) pole-to-pole vertical
distribution of non-refractory aerosols was measured with an aerosol mass spectrometer
onboard the NASA DC8 aircraft as part of the Atmospheric Tomography (ATom) mission
during the northern hemisphere summer (August 2016) and winter (February 2017). This
study presents the first extensive characterization of OA mass concentrations and their
level of oxidation in the remote atmosphere. OA and sulfate are the major contributors by
mass to submicron aerosols in the remote troposphere, together with sea salt in the marine
boundary layer. Sulfate was dominant in the lower stratosphere. OA concentrations have
a strong seasonal and zonal variability, with the highest levels measured in the lower
troposphere in the summer and over the regions influenced by the biomass burning from
Africa (up to 10 $\mu$g sm$^{-3}$). Lower concentrations (~0.1-0.3 $\mu$g sm$^{-3}$) are observed in the
northern mid- and high- latitudes and very low concentrations (< 0.1 $\mu$g sm$^{-3}$) in the
southern mid- and high- latitudes. The ATom dataset is used to evaluate predictions of
eight current global chemistry models that implement a variety of commonly used
representations of OA sources and chemistry, as well as of the AeroCom-II ensemble.
The current model ensemble captures the average vertical and spatial distribution of
measured OA concentrations, and the spread of the individual models remains within a
factor of 5. These results are significantly improved over the AeroCom-II model ensemble,
which shows large overestimations over these regions. However, some of the improved
agreement with observations occurs for the wrong reasons, as models have the tendency
to greatly overestimate the primary OA fraction, and underestimate the secondary fraction.
Measured OA in the remote free troposphere are highly oxygenated with organic aerosol
to organic carbon (OA/OC) ratios of ~ 2.2-2.8 and are 30-60% more oxygenated than in
current models, which can lead to significant errors in OA concentrations. The
model/measurement comparisons presented here support the concept of a more dynamic
OA system as proposed by Hodzic et al. (2016), with enhanced removal of primary OA,
and a stronger production of secondary OA in global models needed to provide a better
agreement with observations.

# 1    Introduction

Organic aerosols (OA) are a complex mixture of directly emitted primary OA (POA) and chemically produced secondary OA (SOA) from anthropogenic and biogenic emission sources. They are associated with adverse health effects (Mauderly and Chow, 2008, Shiraiwa et al., 2017) and contribute radiative forcing in the climate system (Boucher et al., 2013). The currently limited understanding of processes involved in the formation, ageing, and removal of organic compounds results in large uncertainties in (i) the predicted global OA burden, (ii) relative contributions of emissions vs. chemistry to OA formation, (iii) spatial distribution, and (iv) impacts on radiation and clouds (Kanakidou et al., 2005, Hallquist et al., 2009, Heald et al., 2011, Spracklen et al., 2011, Tsigaridis et al., 2014, Hodzic et al., 2016, Shrivastava et al., 2017, Tsigaridis and Kanakidou, 2018, Zhu et al., 2019). The uncertainties are particularly large in the estimated global burden of SOA that range from 12 to 450 Tg y$^{-1}$ (see Fig. 9 of Hodzic et al., 2016), and in their direct and indirect radiative forcing that range from -0.08 to -0.33 W m$^{-2}$, and -0.60 to −0.77 W m$^{-2}$, respectively (Spracklen et al., 2011, Myhre et al., 2013, Scott et al., 2014, Hodzic et al., 2016, Tsigaridis and Kanakidou, 2018). Reducing these uncertainties is becoming more important as OA is on a path to becoming the dominant fraction of the submicron anthropogenic aerosol mass globally due to the ongoing efforts to reduce $SO_2$ emissions and associated sulfate aerosols.

Model performance has been especially poor in the remote regions of the atmosphere where OA measurements available for model evaluation have been sparse (especially aloft). Using data from 17 aircraft campaigns mostly located in the northern hemisphere Heald et al. (2011) showed that the skill of the global GEOS-Chem model in predicting the vertical distribution of OA was significantly decreased in remote regions compared to polluted near-source regions. The study pointed out the limitations of commonly used SOA formation mechanisms that are based on chamber data; these have the tendency to underpredict OA in source regions and overpredict OA in the remote troposphere. For a subset of 9 recent aircraft campaigns, Hodzic et al. (2016) showed that OA is likely a more dynamic system than represented in chemistry-climate models, with both stronger production and stronger removals. These authors suggested that additional removal mechanisms via e.g. photolytic or heterogeneous reactions of OA particles are needed to explain low OA concentrations observed in the upper troposphere where direct cloud scavenging is less efficient. The recent global multi-model comparison study (Tsigaridis

et al., 2014) within the AeroCom Phase II project illustrates well the amplitude of model
uncertainties simulating OA mass concentrations and the contrast in model performance
between near-source and remote regions. The results indicate that model dispersion (the
spread between the models with the lowest and highest predicted OA concentrations)
increases with altitude from roughly 1 order of magnitude near the surface to 2-3 orders
of magnitude in the upper troposphere. Our own analyses of the AeroCom-II models
shown in Figure 1a indicate that model dispersion (quantified as the ratio of the average
concentration of the highest model to that of the lowest one, in each region) increases not
only with altitude but also with distance from the northern mid-latitude source (and data-
rich) regions. The model spread is a factor of 10-20 in the free troposphere between the
equator and northern mid-latitudes, and increases to a factor of 200-800 over the Southern
Ocean and near the tropopause. It is not surprising that model spread is lower closer to
source regions where it is mostly driven by uncertainties in emissions and SOA production
yields. Spread is expected to be larger in remote regions where models are also impacted
by uncertainties in transport, chemical ageing and removal. The lowest model dispersion
also coincides with the regions of the northern hemisphere (NH) or the African biomass
burning outflow where models have been evaluated the most (Figure 1b), emphasizing
the need for further model/observation comparison studies in remote regions (of the
southern hemisphere (SH) in particular).
Here, we present a unique data set of airborne aerosol mass spectrometer measurements
of OA mass concentrations collected onboard the NASA DC-8 as part of the Atmospheric
Tomography (ATom) mission. The aircraft sampled the vertical structure of the
atmosphere from near-surface (0.2 km) to the lower-stratosphere (LS) regions (12 km
altitude) over both the Pacific and Atlantic basins (to limit the influence of source regions)
with a quasi-global spatial coverage from 82°N to 67°S. This dataset is used to perform
the first systematic global-scale multi-model evaluation of the chemistry-climate models
focusing on OA in the remote troposphere over the remote oceans. We focus on the NH
summer (August 2016, ATom-1) and NH winter (February 2017, ATom-2) deployments.
Overall these ATom missions sampled the marine boundary layer (MBL) for 10% of the
flight tracks, 12% of the time the remote lower stratosphere, and the rest the free
troposphere. The model-observation comparisons are aimed at identifying discrepancies
in terms of OA mass concentrations and vertical distribution, their fractional contribution
to submicron aerosols, and their oxidation level in global models.

The modeling framework is described in Section 2. Section 3 describes the ATom dataset and the spatial and vertical distributions of OA over the Atlantic and Pacific regions. Section 4 presents the comparisons of ATom-1 and -2 data to multi-model predictions from both the AeroCom-II models, and the ensemble of eight current model simulations of the ATom campaign. Section 5 presents the conclusions of the study and discusses its implications.

## 2    Modeling framework

### 2.1    ATom model simulations

ATom measurements were compared with results of eight global models that simulated the time period of the ATom-1 and 2 campaigns (August 2016 and February 2017), using the emissions and reanalysis meteorology corresponding to this period (and a spin-up time of at least six to twelve months). These are referred hereafter as ATom models and include the NASA global Earth system model GEOS5, the aerosol-climate model ECHAM6-HAM, three versions of the NCAR Community Earth System Model (CESM), and three versions of the global chemistry GEOS-Chem model. Simulations were performed at various horizontal resolutions ranging from relatively high ~50km (GEOS5) and ~100km (CESM2 models) resolutions to somewhat coarser grids of ~200km (CESM1-CARMA, GEOS-Chem) and ~400km for GC10-TOMAS. The advantage of using the same host model (in the cases of variants of CESM2 and GEOS-Chem) is that the dynamics and emissions remain comparable. Models differ greatly in their treatment of emissions, gas-phase chemistry, aerosol chemistry and physical processes, and aerosol coupling with radiation and clouds, among others. Table 1 describes the configuration of various models (e.g. meteorology, emissions), and their treatment of OA. In this section we only summarize the main features and parameters directly impacting the OA simulations. Some models do not include SOA chemistry and instead assume that SOA is directly emitted proportional to the emissions of its precursors (ECHAM6-HAM, CESM2-SMP, GEOS5, GC10-TOMAS), while others have more complex treatments of organic compounds, their chemistry, and partitioning into particles (GC12-REF, GC12-DYN, GC10-TOMAS, CESM1-CARMA, CESM2-DYN). It should be noted that models that directly emit SOA assume that SOA is a non-volatile species that remains irreversibly in the particle phase. In all models POA is treated as a non-volatile directly emitted species. In most models (see below) the primary emitted organic aerosol is artificially aged to transition between hydrophobic to hydrophilic POA. There are some commonalities between simulations for

the treatment of biogenic emissions, which are based in all models on the Model of
Emissions of Gases and Aerosols from Nature (MEGAN, Guenther et al., 2012) to
generate meteorology-dependent emissions of volatile organic compounds. None of the
models includes the marine production of OA which is estimated to be ~3 orders of
magnitude smaller than the continental production of OA from both isoprene and
monoterpene precursors (Kim et al., 2017), but could be important in the MBL. This
contribution could however be larger for sea-spray biological material from phytoplankton
with predicted contributions of 0.01 to 0.1 $\mu$g m$^{-3}$ to surface submicron aerosol over remote
oceanic regions (Vergara-Temprado et al., 2017, Middlebrook et al., 1998). Below we only
provide a brief description of most important processes that influence OA for each model.
GEOS5 was run in a configuration similar to Bian et al. (2019) using the anthropogenic
emissions from HTAP v2 (Janssens-Maenhout et al., 2015) and biomass burning
emissions from the Quick Fire Emission Dataset (QFED v2.54). Aerosols are simulated
within the GOCART bulk aerosol module and include externally mixed particles of black
carbon (BC), organic carbon (OC), sulfate, ammonium, nitrate, dust and sea salt (Colarco
et al., 2010, Bian et al., 2017). The formation of SOA is based on a prescribed 10%
formation yield from the monoterpene emissions. The primary emitted OC and SOA are
separated into hydrophobic (50%) and hydrophilic (50%) species, with a 2.5 days e-folding
time conversion from hydrophobic to hydrophilic organic particles. All SOAs from biogenic,
anthropogenic, and biomass burning sources are treated as hydrophilic particles. Both
types of organic particles are dry deposited. The hydrophilic OA is removed by large-scale
and convective warm clouds, while hydrophobic OA is removed by ice clouds. The
hydrophilic particles undergo hygroscopic growth according to the equilibrium
parameterization of Gerber (1985).
The ECHAM6.3-HAM2.3 standard version (Tegen et al., 2019) was run using updated
anthropogenic emissions (Schacht et al, 2019) combining the ECLIPSE (Klimont et al.,
2017) emissions, with the Russian anthropogenic BC emissions from Huang et al. (2015).
For biomass burning the Global Fire Assimilation System (GFAS, Kaiser et al., 2012)
biomass burning emissions are used, however, without the scaling factor of 3.4 suggested
by Kaiser et al. (2012). Aerosol composition and processes are simulated using the
Hamburg Aerosol Model (HAM2, Zhang et al., 2012), that considers an aerosol internal
mixture of sulfate, BC, OC, sea salt, and mineral dust. The aerosol population and their
microphysical interactions are simulated using seven log-normal modes, including the
nucleation mode, soluble and insoluble Aitken, accumulation and coarse modes. In the
model configuration used in this publication the formation of SOA is based on a prescribed
15% mass yield from monoterpene emissions only (Dentener et al., 2006). Aerosol
particles are removed by dry and wet deposition. The wet deposition includes the below
cloud scavenging by rain and in-cloud cloud scavenging for large-scale and convective
systems (Croft et al., 2010).
The two simulations with the GEOS-Chem 12.0.1 global chemistry model (Bey et al.,
2001) use emissions based on CMIP6 global inventory (CEDS historical emissions up to
2014 and future emissions based on climate scenarios, Hoesly et al., 2018; Feng et al.,
2019) with regional improvements for anthropogenic sources, and on GFED v.4 for
biomass burning emissions (Giglio et al., 2013). Both simulations use the bulk aerosol
representation and differ only in the treatment of SOA formation and removal. The first
configuration    (called    hereafter    GC12-REF)    includes    the    default
(http://wiki.seas.harvard.edu/geos-chem/index.php) representation of SOA formation
based on Marais et al. (2016) for isoprene-derived SOA, and on the volatility basis set
(VBS) of Pye et al. (2010) for all other precursors. Note that this GEOS-Chem REF
simulation is similar to the version 12 default "complex option" which includes non-volatile
POA and semi-volatile SOA (semi-volatile POA is an optional switch within this version
used in Pai et al. 2020). The second configuration (referred to as GC12-DYN) includes a
more dynamic representation of the SOA lifecycle based on Hodzic et al. (2016), with the
exception of the treatment of isoprene SOA that is formed in the aqueous aerosols as in
Marais et al. (2016). As in Hodzic et al. (2016) the GC12-DYN model version includes
updated VBS SOA parameterization, updated dry and wet removal of organic vapors, and
photolytic removal of SOA (except for isoprene-SOA that is formed in aqueous aerosols,
where we follow Marais et al. 2016). SOA formation is based on wall-corrected chamber
yields (Zhang et al., 2014) for the traditional precursors (isoprene, monoterpenes,
sesquiterpenes, benzene, toluene, xylene) and on yields derived from an explicit chemical
mechanism for higher molecular weight n-alkanes and n-alkenes species (Hodzic et al.,
2016). The removal of gas-phase oxidized volatile organics uses updated Henry's law
solubility coefficients from Hodzic et al. (2014), and photolytic removal of SOA (Hodzic et
al., 2015). In addition to OA, the model includes BC and dust, and simulates the chemistry
and gas-particle partitioning of inorganic compounds such as sulfate, ammonium, nitrate
and sea salt using the ISORROPIA II thermodynamic model (Fountoukis and Nenes,
2007). In both GEOS-Chem configurations, BC and primary OC are simulated with a
hydrophobic and hydrophilic fraction for each. At the time of emission, 80% of BC and
50% of primary OC are considered as hydrophobic. Hydrophobic aerosols are converted
to hydrophilic aerosols with an e-folding lifetime of 1.15 days. An OA/OC ratio of 2.1 is
assumed to convert POC to POA, and SOA is simulated as OA mass (i.e. no OA/OC ratio
assumption is needed for SOA, except for comparison with OC measurements). Soluble
gases and aerosols are removed by both dry and wet deposition. Wet deposition includes
scavenging in convective updrafts, and in-cloud and below-cloud scavenging from large-
scale precipitation (Liu et al., 2001). Hydrophobic aerosols (BC and POA) are scavenged
in convective updrafts following Wang et al. (2014).
GC10-TOMAS is based on the GEOS-Chem version 10.01 coupled with TwO Moment
Aerosol Sectional microphysics scheme (TOMAS) and ran in a similar configuration to that
described in Kodros et al. (2016). The model computes the evolution of sulfate, sea salt,
primary and secondary OA, BC, and dust aerosols described by 15 internally mixed size
bins (of which six were analyzed for these comparisons, cf. Table 1). Anthropogenic
emissions are based on the EDGAR v4 global inventory with regional improvements, while
the biomass burning emissions are from GFED v3. SOA are irreversibly made from the
emitted parent precursor, considering a 10% mass yield from monoterpene emissions,
and an emission flux of 0.2 Tg of SOA per Tg of CO for the anthropogenic CO emissions.
The removal of gases and aerosols are treated similar to the GEOS-Chem 12.0.1 model
(GC12-REF, see above).
Simulations based on the CESM2.0 Earth system model use the standard version of the
Whole Atmosphere Community Climate Model (WACCM6, Gettelman et al., 2019,
Emmons et al., 2019). Details on the specific of the model configurations are described in
detail in Tilmes et al. (2019) i.e. CESM2-SMP and CESM2-DYN correspond to the
specified dynamics WACCM6-SOAG and WACCM6-VBSext simulations described in that
work, respectively. Emissions are based on the CMIP6 global inventory for the year 2014
for anthropogenic sources, and on the QFED version 2.4 for the wildfires inventory.
Aerosols are represented with the modal aerosol scheme (MAM4, Liu et al., 2012) that
includes BC, primary and secondary OA, sulfate, dust and sea salt. Four modes are
considered including Aitken, accumulation and coarse size modes, and an additional
primary carbon mode. Only the accumulation mode was used in this work. The CESM2-
SMP and CESM2-DYN simulations differ in their treatment of OA. CESM2-SMP forms OA
directly using fixed mass yields from primary emitted precursors (isoprene, monoterpenes,

aromatics) without explicitly simulating their oxidation and partitioning. These mass yields are increased by a factor of 1.5 to match the anthropogenic aerosol indirect forcing (Liu et al., 2012). The second configuration (referred to as CESM2-DYN) includes the formation and removal parameterizations of organics of Hodzic et al. (2016), as implemented into CESM2 by Tilmes et al. (2019) for all species based on low-NOx VBS yields only. This is a similar SOA scheme as used in GC12-DYN (with differences in the treatment of isoprene-SOA based on Marais et al. 2016 in GC12-DYN, and the use of both low- and high-NOx VBS yields in GC12-DYN). Organic gases and aerosols undergo dry and wet deposition as described in Liu et al. (2012). It should be noted that CESM2-SMP does not include deposition of intermediate organic vapors. Aerosol wet scavenging considers in-cloud scavenging (the removal of cloud-borne particles that were activated at the cloud base) and below-cloud scavenging for both convective and grid-scale clouds.

CESM1-CARMA simulations use the configuration described in Yu et al. (2019) which is based on CESM1 and the sectional Community Aerosol and Radiation Model for Atmospheres (CARMA v3.0). Anthropogenic emissions are those from the Greenhouse gas-Air pollution Interactions and Synergies (GAINS) model, and biomass burning emissions are from the Global Fire Emission Database (GFED v3, van der Werf et al., 2010). In CARMA, 20 size bins are used for both pure sulfate particles (bins from 0.2 nm to 1.3 µm in radius, only used up to 500 nm) and mixed aerosols composed of BC, primary and secondary OC, dust, sea salt, and sea-spray sulfate (bins from 0.05–8.7 µm in radius, again, only analyzed up to 500 nm). SOA formation is based on the VBS approach from Pye et al. (2010). The removal of OA occurs only by dry and wet deposition. Compared to the CESM2-SMP and CESM2-DYN simulations, the convective removal of aerosols uses the modified scheme described in Yu et al. (2019) which accounts for aerosol secondary activation from the entrained air above the cloud base, and the scavenging of activated aerosols in convective updrafts. The default CESM can transport aerosols from the cloud base to the top of the cloud in strong convective updrafts in one time step without scavenging them, while the new scheme allows for a more efficient removal off all aerosols inside convective clouds. A sensitivity simulation is performed for ATom-1 to quantify the effect of this improved removal on OA concentrations (Section 4.5).

### 2.2 AeroCom-II model climatology

The ATom measurements are also compared to the global model OA predictions generated within the Phase II Aerosol Comparisons between Observations and Models

(AeroCom-II) project (Schulz et al., 2009). We consider the monthly average results of 28
global models, which is a subset of those presented in Tsigaridis et al. (2014), based on
the availability of model results. It should be noted that the meteorological forcing used in
these models is mostly based on the year 2006, while the anthropogenic and biomass
burning emissions are mostly representative of the year 2000. For comparison purposes,
the monthly mean model outputs for the months of August (ATom-1) and February (ATom-
2) are interpolated along the flight path (latitude, longitude, and altitude), and averaged
the same way as the measurements (see section 3.2).
## 3    Description of ATom measurements
### *3.1    Submicron aerosol data*
The measurements of non-refractory submicron aerosols were performed onboard the
NASA DC8 aircraft as part of the ATom field study (Wofsy et al., 2018) using the University
of Colorado Aerodyne High-Resolution Time-of-Flight Aerosol Mass Spectrometer (AMS
in the following, Canagaratna et al., 2007, DeCarlo et al., 2006).
We use measurements from both the NH summer (August 2016, ATom-1) and winter
(February 2017, ATom-2) deployments. Figure 2a shows the flight path and the vertical
extent of the ATom-1 dataset colored by OA mass concentrations (see Figure S1 for
ATom-2). The aircraft performed systematic vertical sampling with ~140 vertical profiles
per campaign throughout the troposphere from the near surface ~0.2 km to the upper
troposphere/lower stratosphere region at ~13 km altitude. Details on the operation of the
CU AMS on board the DC-8 are reported in Schroder et al. (2018), Nault et al. (2018), and
Jimenez et al. (2019b). AMS data was acquired at 1 Hz time resolution and independently
processed and reported at both 1 s and 60 s time resolutions (Jimenez et al., 2019a). The
later product, with more robust peak fitting at low concentrations was exclusively used as
the primary dataset in this work. Detection limits at different time resolutions/geographical
bins relevant to this study are discussed in Section 3.3. The overall 2σ accuracies of the
AMS measurement (38% for OA, 34% for sulfate and other inorganics) are discussed in
Bahreini et al. (2008) and Jimenez et al. (2019b).
For ATom, the AMS reported the standard non-refractory aerosol species OA, sulfate,
nitrate, ammonium, and chloride, with the response for all the nominally inorganic species
characterized by in-field calibrations. In addition, it also reported methanesulfonic acid
(MSA, Hodshire et al., 2019a describes the AMS MSA methods and calibrations for ATom)
and sea salt for for $D_{geo}$<450 nm (based on the method of Ovadnevaite et al., 2012). Both
of these species were important to achieve closure with the volume calculated from the
on-board sizing instruments in the marine boundary layer (Jimenez et al., 2019b). Another
important refractory submicron species not captured by the AMS measurements is BC.
This was measured on ATom with the NOAA SP2 instrument (Katich et al., 2018). It should
be noted that aircraft measurements of aerosol mass concentrations are given in $\mu g\ sm^{-3}$
(i.e., under standard conditions of 1 atm and 273.15 K).
For ATom the AMS measured particles with geometric diameters (based on the campaign-
wide average density of 1640 kg m$^{-3}$, Jimenez et al., 2019b) of between $D_{geo}$~60 and 295
nm with ~100% efficiency (and between 35 and 460 nm with 50% efficiency). Here we
denote the AMS aerosol data as "submicron" mass (based on the more usual definition
using aerodynamic diameter, which is larger than the geometric diameter; DeCarlo et al.,
2004), with the assumption that non-refractory aerosol are small contributors to mass
above the AMS size range. As shown in Brock et al., 2019, the accumulation mode for the
ATom sampling environment only extended up to 500 nm, and hence, as expected for a
background tropospheric environment, this approximation is appropriate. Very good
agreement was observed with the integrated volume calculated from the number size
distributions for ATom (Brock et al., 2019). A low bias compared to a typical submicron
definition can occur in thick biomass burning plumes and in the lower stratosphere at times
(Jimenez et al., 2019b). As detailed in Table 1, the accumulation mode for the bulk models
discussed in this study overlaps with the size range of the AMS, and for the sectional
models (CESM1-CARMA, GEOS-Chem-TOMAS, ECHAM6-HAM) only the bins that
match the AMS size range were used. As expected based on the previous discussion,
however, a comparison of the total OA calculated by these sectional models with the
modeled OA inside the AMS size-range showed small differences (Slopes for ATom-1
linear regressions: CESM1-CARMA:0.91, GC10-TOMAS: 0.94, ECHAM6-HAM 1.00)
mostly influenced by the high concentration points in the biomass plumes off Africa that
have a large effect on the regression since they are about 10 times larger than the bulk of
the dataset).
Refractory and non-refractory aerosol composition was also measured using the NOAA
Particle Analysis by Laser Mass Spectrometry (PALMS) instrument. PALMS classifies
individual aerosol particles into compositional classes including biomass burning (Hudson
et al., 2004), sea salt (Murphy et al., 2019), mineral dust (Froyd et al., 2019), and

others. Mass concentrations for these particles types are derived by combining PALMS composition data with aerosol size distribution measurements (Froyd et al., 2019). Good agreement overall was found for OA, sulfate and seasalt between the two particle mass spectrometers during ATom once the AMS and PALMS instrument transmissions were accounted for (Jimenez et al., 2019b). For all PALMS data used in this work (biomass burning fraction and dust) the AMS transmission function was applied to ensure that both instruments were characterizing approximately the same particle range.

For a particular airmass, the mass fraction of biomass burning (BB) aerosol reported by the PALMS instrument f(BB)$_{PALMS}$ (Thompson and Murphy, 2000; Froyd et al., 2019) was then used to evaluate the degree of BB influence. This parameter correlates quite well with other gas-phase BB tracers (Figure S20), and is more useful as a particle tracer since its lifetime follows that of the particles. Importantly, it is not impacted by the long lifetimes of the gas-phase tracers (e.g. 9 months for CH$_3$CN) and unrelated removal processes (e.g. ocean uptake for CH$_3$CN and HCN) that result in highly variable backgrounds. Hence f(BB)$_{PALMS}$ has a much higher contrast ratio and linearity for particle BB impacts, compared to the available gas-phase tracers in the ATom dataset. An airmass was classified as non-BB influenced when f(BB)$_{PALMS}$ was lower than 0.30 (Hudson et al, 2004) as shown in Figure 2b. For both ATom-1 and 2, about 74% of measurements were classified as not influenced by biomass burning. f(BB)$_{PALMS}$ was also used to assess the impact of POA on the total OA burden (next section); note that no thresholding was applied in that case.

### 3.2   Estimation of the POA fraction for the ATom dataset

For model evaluation purposes, it is important to know whether the source of OA is primary or secondary. For ground studies close to sources (e.g. Jimenez et al., 2009) Positive Matrix Factorization of AMS mass spectra (PMF, Ulbrich et al., 2009) can be used to estimate the contribution of primary sources (mostly from transportation, heating, cooking, and biomass burning) to total OA. This approach is not suitable for ATom. To accurately resolve a minor factor such as POA in an AMS dataset, there needs to be a combination of: (a) Sufficient OA mass concentration, so that the signal-to-noise of the spectra is sufficient; (b) Enough fractional mass for the factor to be resolved (>5% in urban areas per Ulbrich et al. (2009), probably a larger fraction at low concentrations such as in ATom); (c) Sufficient spatio-temporal variability ("contrast") in the relative contributions of different factors, since that is part of what PMF uses to extract the factors; (d) Sufficient difference in the spectra of the different factors (for the same reason as (c)), and (e) relatively

invariant spectra for each factor across the dataset (as this is a key assumption of the
PMF algorithm). As an example of a near ideal case, in Hodshire et al (2019) we extracted
MSA by PMF from the ATom-1 data, and were able to match that factor with our
independently calibrated MSA species. A very distinct and nearly invariant mass spectrum
was measured repeatedly near sources (MBL) (and was mostly absent elsewhere, thus
providing strong spatio-temporal contrast) and accounted for about 6% of the fractional
mass and 15% of the variance in time. Thus all the conditions were met. For POA, on the
other hand, the air sampled in ATom and coming from e.g. Asia has POA and SOA very
well mixed, with little change on their relative mass fractions vs. time (as the aircraft flies
through that airmass). POA is very low, as documented later in this paper. Atmospheric
aging makes the spectra from all OA sources more and more similar as measured by AMS
spectra (Jimenez et al., 2009). Thus most of the conditions above are not satisfied for
extracting POA by PMF analysis of this dataset.
Instead, in this work we have estimated POA based on the fact that it is co-emitted with
BC as part of the combustion processes releasing both species in source regions, and
that BC is not impacted by chemical aging processes over the lifetime of the airmass. Note
that BC can physically age but it is not lost in any significant amount to the gas-phase due
to chemical processes in the atmosphere. We assume non-differential removal (and
transport) of the BC fraction relative to the rest of the POA (the two are generally internally
mixed, Lee et al., 2015). Table S1 summarizes recent POA/BC and POC/EC emission
ratio determinations for urban background sites, which best represent real mixes of
pollution sources, and for individual sources of POA (from mobile sources – commonly
referred as HOA – and cooking aerosol – COA). Based on Table S1 data, we assume
POA to be co-emitted with BC for anthropogenic fossil fuel / urban region POA (herein
called FF$_{ratio}$ for simplicity, even though much of it is non-fossil, Zotter et al., 2014; Hayes
et al., 2015) at a ratio of 1.5±0.82 (average ±1$\sigma$ of all urban ambient air studies that report
POA and BC for best intercomparability to the ATom dataset; including all urban studies
results in a very similar number, 1.48±0.65, median: 1.41). Measurements where mobile
source are the main contributor in general exhibit lower ratios (POA/OA ratio 0.5-1.5),
while COA determination typically ranges from 2 to 3. Hence, the ratio used here is a good
estimate for a diverse mix of urban sources as appropriate for ATom. The studies used to
derive the emission ratio used ambient data in urban air, where all sources mix together
and impact the POA/BC ratio, and thus the ratios include the impact of POA sources that
may not emit BC. It should be noted that urban model ratios do not include emissions
associated with fugitive dust from road, tire and construction, as those are typically found
in larger particles than those studied here (Zhao et al., 2017). For biomass burning
sources, we use a value of POA/BC = 11.8 ($BB_{ratio}$), based on the average of the recent
review by Andreae (2019), which included over 200 previous determinations for a variety
of fuels and burning conditions (since Andreae (2019) used and OA/OC ratio of 1.6 in his
work, we have used that value to calculate POA/BC; we note that this is different from the
1.8 OA/OC ratio used for other studies listed in Table S1). We note the measured total
OA/BC of ~3.5 (conservatively assuming that all OA is POA) observed on both ATom
missions for the large African-sourced BB plumes over the Equatorial Atlantic. We note
that using the larger $BB_{ratio}$ from Andreae (2019) leads to a POA fraction >> 100% in the
ATom African plumes. We also perform sensitivity studies with values of both $FF_{ratio}$ and
$BB_{ratio}$ within the literature range.
The PALMS determined mass fraction of biomass impacted aerosol ($f(BB)_{PALMS}$) can then
be used to determine a total POA contribution from both types of sources:
$$POA = [BC]*(FF_{ratio}+(BB_{ratio}- FF_{ratio})*f(BB)_{PALMS}) \qquad \text{(Eq. 1)}$$
Further detail is provided in Table S2, which summarizes the POA/BC ratios used in the
emission inventories implemented in current models. Overall, there is reasonable
agreement with the measurements in Table S1, with $FF_{ratio}$ ranging from ~0.5 for diesel
fuels, to >2 for energy production and ~5 for residential emissions (which include some
BB).  On the other hand, for biomass burning, the emission inventories ratios range from
~5 for crop, to ~15 for forest, and up to ~50 for peatland. While generally consistent with
the values discussed by Andreae (2019), they are on the lower end of the ranges
discussed in that work. The averages and ranges of the measurement and model ratios
are similar, and thus no significant model bias on the ratios is apparent.
PALMS detection efficiency increases with size across the accumulation mode, and
therefore the f(BB) number fraction is weighted to the larger size end of the accumulation
mode. In very clean regions of the upper troposphere (typically <0.15 µg sm$^{-3}$ submicron
mass) particles below the PALMS size range can contribute significantly to aerosol mass
(Williamson et al., 2019; Jimenez et al., 2019b). If BB particles are not evenly distributed
across the entire accumulation mode (due to preferential removal in convective updrafts
of primary aerosol, cf. Yu et al., 2019 and Section 4.5; and preferential condensation of
SOA on smaller particles), then the f(BB) reported by PALMS will be an overestimation.
For the final analysis these periods where left in the dataset, and therefore for the LS the
reported POA is likely overestimated for these regions, although their impact on the mass-
weighted campaign average is negligible.
The contribution of POA from sea spray is difficult to constrain. As an order-of-magnitude
estimate, marine POA is roughly calculated based on preliminary calibrations of OA on
mineral dust particles from the PALMS instrument (personal communication K. Froyd).
Using this calibration, the average OA by mass on sea salt was <10% for the large majority
of MBL sampling (>85%). Since sea salt contributed 4% (11%) of mass in the AMS size
range for ATom-1(2) (Figure 2), we estimate that marine POA is on the order of ~1% of
aerosol mass in the AMS size range, and possibly much lower. Thus we think that it is
reasonable to neglect the contribution of marine POA to this dataset. Future studies will
refine this estimate.
### 3.3   Data processing for comparisons
For the comparisons between the measurements and the various global models, data
were averaged both vertically and zonally to minimize the impact of smaller plumes or
vertical gradients in aerosol concentrations that might not be captured by coarse resolution
models. For the same reason, all data near airports was removed from the datasets prior
to analysis (up to about 3 km on the climb in/out). In order to restrict this analysis to the
remote troposphere, the last leg of the ATom-1 mission (over the continental US) was
taken out of the dataset as well. Data was binned into 5 large latitude regions as shown in
Figure 2a including southern polar (55-80°S, "S.Polar"), southern mid-latitudes (25-55°S,
"S.Mid"), equatorial (25°S-25°N, "Equatorial"), northern mid-latitudes (25-55°N, "N.Mid"),
northern polar (55-80°N, "N.Polar") and analyzed separately for the Pacific and Atlantic
basins. For data in each of these latitude regions, altitude profiles were calculated with a
constant 600 m altitude resolution. According to both variability in the cleanest air and
statistical analysis of the organic background subtraction (Drewnick et al. 2009), the 1σ
precision at low concentrations for one-minute data ranged between 20 and 50 ng sm$^{-3}$,
or a 3σ detection limit between 60 and 150 ng sm$^{-3}$ for the one-minute data (confirmed by
frequent filter blanks). Per standard statistics, the precision of a measurement decreases
(i.e., gets better) with the square root of the number of points (or time interval) sampled.
I.e. the precision of an average can be approximated by the standard error of the mean
(σ/sqrt($n$), where $n$ is the number of measurements averaged), and it is better than the
precision of the individual data points (σ). This also applies to the detection limit, since it
is just 3 times the precision. Note that a detection limit is not meaningful unless the
averaging time is specified. For example let's assume that the detection limit is 20 ng m$^{-3}$
(1-second), and the data points over 60 consecutive seconds are all 10 ng m$^{-3}$. All 1-
second measurements are below the 1-second DL. However the average (10 ng m$^{-3}$) is
now above the DL for 1-minute averages, which is 20/sqrt(60) = 2.6 ng m$^{-3}$. On average,
each individual point in the profiles represents the average of about 25 min of ATom flight
data. At that time resolution, the OA 1σ precision was about 10 ng sm$^{-3}$. Hence with very
few exceptions (10 points for both missions combined), the OA concentrations in the
averaged profiles reported are well above the instrumental detection limit in those regions.
For model-measurement comparisons along flight tracks, model outputs and
measurements were considered at 1-minute time resolution, which corresponds to ~0-700
m vertical resolution and ~0.05-0.15 degrees horizontal resolution. Note that a large
fraction of the 1-minute OA values in the remote free troposphere were below the local 3σ
detection limit. The data of periods of zero concentration (sampling ambient air through a
particle filter) do average to zero. Some negative measurements are present, and this is
normal for measurements of very low concentrations in the presence of instrumental
noise. Averaging of longer periods, as done for the figures in this paper, reduces the
detection limit. We therefore caution future data users that the reported data should be
averaged as needed, as replacing below-detection limit (or negative) values by other
values introduces biases on averages. For fractional ratio analysis, measurements were
averaged to 5-minute time resolution to reduce the noise in the ratios due to noise in the
denominator. The results are not very sensitive to the 5-minute averaging (compared to
1-minute) as shown in Figure S12 for OA to sulfate ratios. The same figure also illustrates
that excluding ratios affected by negative concentrations (the non-bracketed case, overall
these are about 15% of the dataset) does not really affect the fractional distribution, with
the variance between the two cases diminishing as the averaging interval increases. To
further confirm that there is no inherent bias in the fractional products regardless of the
treatment of low concentration values, an additional sensitivity analysis was performed
where data was filtered by an independent measurement proxy for aerosol mass, the
aerosol volume measured in ATom (Brock et al., 2019). Using a range of value that
encompasses the regime where the AMS calculated volume to aerosol measured volume
exhibited increased noise (Jimenez et al., 2019b), no systematic bias was found (Figure
S13), with variations of about 10% in fractional volume for different filtering conditions.
Some of the performed analysis required separating the dataset into vertical subsets. In
this manuscript, we define the marine boundary layer (MBL) as the region below 1.5 times
the calculated boundary layer height in the NCEP global model reanalysis. The free
troposphere (FT) includes all data points between the top of MBL and the NCEP
tropopause height, and the LS region includes all points above the NCEP tropopause
height. The tropopause height varied during ATom between 8 and 16.5 km; given the DC-
8 ceiling (42 kft, 12.8 km) the stratosphere was only sampled at latitudes higher than 30
degrees in both hemispheres. The MBL height varied between up to 1.5 km in the mid-
latitudes, ~1 km in the tropics, and sometimes <150 m (lowest DC-8 altitude) for some of
the sampling in the polar troposphere.

### 538 *3.4 Submicron aerosol composition*

Figure 2b shows that during both NH summer and winter ATom deployments, OA is one
of the three dominant components of the measured submicron aerosol in the remote
troposphere, together with sulfate and sea salt. During ATom-1, average submicron
aerosol concentrations were close to 0.8 $\mu$g sm$^{-3}$ in the marine boundary layer and
biomass burning outflow regions, and ~2 times lower in the free troposphere and lower
stratosphere regions. ATom-2 had overall lower average concentrations below 0.4 µg sm$^-$
$^3$ (vs. 0.5 µg sm$^{-3}$ for ATom-1). As expected, sulfate (sulfuric acid in the lower stratosphere)
is the dominant constituent in the MBL (~50%) and LS (50-70%), while the OA contribution
is generally below 10% and 40%, respectively in those regions. A large fraction of sea salt
aerosol is found in the MBL especially during the NH winter deployment (~30%, see
Murphy et al., 2019).
OA is found to be a major constituent (~50%) of submicron aerosol in the clean (non-BB
influenced) free troposphere. The contribution of OA is 1.4 times larger than that of sulfate
during the NH summer, and 1.2 times lower than that of sulfate during the NH winter,
which is likely due to a large contribution of the NH sources to SOA production in the NH
summer. Biomass-burning events increase the OA contribution relative to that of sulfate,
and lead to a higher contribution of OA to total during the ATom-1 mission (stronger BB
influence).

### 557 *3.5 Spatial and vertical distribution of OA*

Figure 2a (and Fig. S1) shows the spatial and vertical distribution of OA mass
concentrations measured during ATom-1 (and ATom-2) campaigns. Most data were taken
over remote oceanic regions (and a few remote continental regions, primarily over the
Arctic). The measured OA varies between extremely clean conditions (< 0.1 µg sm$^{-3}$)
encountered mostly in the Pacific and Southern Ocean regions and moderately polluted
conditions (> 2 $\mu$g sm$^{-3}$) in the biomass burning outflow regions. During ATom-1 (August
2016), a strong BB influence is observed in the lower troposphere (below 6 km) over the
Atlantic basin off the African coast and over California with OA concentrations exceeding
10 $\mu$g sm$^{-3}$. OA associated with biomass burning is also present in the upper troposphere
over equatorial regions and over Alaska, associated with the deep convective transport of
biomass burning aerosols. The biomass burning contribution to carbonaceous aerosols in
those regions during ATom-1 was also apparent in the black carbon measurements
(Katich et al., 2019). ATom-2 was generally less polluted than ATom-1, likely due to a
more limited global influence of biomass burning emissions during that period, and also to
a less active photochemistry during winter months in the NH.
The measured OA is characterized by a strong latitudinal gradient. Figure 2c shows the
average vertical profiles of measured OA over the selected latitudinal bands during August
2016. The cleanest airmasses are observed over the remote oceanic regions of the
Southern Hemisphere (SH, 25-80°S) with OA mass concentrations below 0.06 $\mu$g sm$^{-3}$.
These extremely low OA concentrations can be explained by the very low influence from
continental emission sources, and presumably low marine POA and SOA precursor
emissions. This is consistent with low concentrations of gas-phase pollutants (e.g. CO,
ethane, propane). An enhancement can be noticed above 10 km in the lower stratosphere.
In some cases, this could be related to the long-range transport of biomass burning
aerosols from the tropics. By comparison, the Arctic region is more polluted with an order
of magnitude higher OA levels compared to its analog of the SH (i.e. OA loadings ranging
from 0.1 to 0.5 $\mu$g sm$^{-3}$). These concentrations are comparable to FT levels measured in
the extratropical regions (25-55°N) of the NH. The equatorial marine regions (25°S-25°N)
display the highest OA concentrations with a strong gradient between lower and upper
troposphere. In the lower troposphere OA, concentrations are close to 1 $\mu$g sm$^{-3}$, and
decrease down to 0.1 $\mu$g sm$^{-3}$ at altitudes above 4km. The highest OA levels are
associated with the African outflow over the southeastern Atlantic Ocean, which results
from the transport of the biomass burning smoke from the sub-Saharan regions and
increasing urban and industrial air pollution in southern West Africa (Flamant et al., 2018).
Figure 2d shows that the Atlantic basin is often more polluted than the Pacific basin, not
only because of the African biomass burning influence but also due to the contribution of
anthropogenic pollution in the lower troposphere of the NH. It should be noted that Asian
pollution was likely an important contributor to the North Pacific Basin, especially between
2 and 6 km, in both ATom deployments (see figures 2a and S1). Several-fold higher OA
concentrations are found near the surface (below 1km) over the southern Pacific
compared to that same location in the southern Atlantic, which could be indicative of the
stronger emission of marine OA in the Pacific basin.
In addition to spatial gradients, a strong summer-to-winter contrast is observed in OA
concentrations. Figure 2e shows the ratio between OA vertical profiles measured in the
NH summer ATom-1 vs. in the NH winter ATom-2. The NH is more polluted during the NH
summer due to the photochemical production of SOA, as well as biomass burning
emissions, leading to the tripling of OA concentrations in the extratropical regions (25-
80°N) on average regardless of altitude. The doubling of OA loading in the lower
troposphere at the equator (25°S-25°N) in the NH summer (August, ATom-1) is strongly
influenced by the biomass burning activity in the sub-Saharan African region as already
mentioned above. Likewise, OA concentrations are found to be generally higher in the SH
during the SH summer. These zonal trends are broadly similar to the ones described in
Katich et al (2018) for BC.
**4    Model-measurement comparisons**
*4.1    Evaluation of predicted OA concentrations*
Prior to evaluating model performance in simulating OA, we have assessed the ATom
models' ability to simulate sulfate aerosols. According to the model evaluation shown in
Table S3, the predicted sulfate concentrations are generally within 40% of the measured
values, which is comparable to the AMS measurement uncertainties. The only exception
is found for the ECHAM6-HAM model, which overestimates sulfate aerosols by a factor of
two. These results imply that most ATom models capture relatively well the overall sulfate
burden. However, large root mean square error (RMSE > 0.4 µg sm$^{-3}$ for ATom-1 and >
0.2 µg sm$^{-3}$ for ATom-2) is indicative of their limited skill in reproducing the observed
variability in sulfate concentrations.
For OA, model evaluation metrics for the entire ATom-1 and ATom-2 campaigns are given
in Table 2 for the eight ATom models and their ensemble, as well as the AeroCom-II
ensemble. The results show that the normalized mean bias is substantially lower for the
ATom model ensemble compared to AeroCom-II decreasing from 74% to 4% for ATom-1
and from 137% to 23% to ATom-2, which is within the measurement uncertainty range.
The mean temporal correlations are substantially improved from 0.31 (0.38) for AeroCom-
II to 0.66 (0.48) for ATom model ensemble during ATom-1 (ATom-2). However, results
vary strongly among ATom models. Models using prescribed emissions of non-volatile
SOA have the tendency to overestimate the OA concentrations during both NH summer
and winter deployments (with ~35-60% overestimation for CESM2-SMP, ~70-100% for
ECHAM6-HAM, and up to 150% for GC10-TOMAS during ATom-2), with the exception of
the GEOS5 model that on the contrary underestimates OA concentrations by 5-25%.
During the NH summer (ATom-1), models using the VBS parameterization from Pye et al.
(2010) tend to underpredict the OA concentrations by 43% for GC12-REF and 33% for
CESM1-CARMA for ATom-1, most likely due to the excessive evaporation of the formed
SOA in remote regions and low yields for anthropogenic SOA (Schroder et al., 2018; Shah
et al., 2019). Models using the VBS parameterization from Hodzic et al. (2016) (CESM2-
DYN and GC12-DYN) where OA is less volatile and also OA yields are corrected for wall
losses show an improved agreement with observations especially for CESM2-DYN (with
NMB of ~5%), and to a lesser extent for GC12-DYN (NMB of ~33%). During the NH winter
(ATom-2) characterized by a lower production of SOA, both VBS approaches lead to an
overestimation of the predicted OA. This is likely caused by excessively high levels of
primary emitted OA as discussed in section 4.4.
Figure 3 compares the average median ratios between modeled and observed OA
concentrations for the ATom and AeroCom-II model ensembles for different regions (BB,
MBL, FT, LS). The results show that the median ratio for the ATom model ensemble is
close to unity in all regions. This is at least a factor of two improvement compared to
AeroCom-II models, which were almost always biased high for the remote regions
sampled in ATom. The model spread has also been reduced by a factor of 2-3 in all
regions. This reduction in the ensemble spread may partially be explained by a smaller
size of the ATom model ensemble (see Fig. S2), which also includes models with a more
up-to-date OA representation. In order to explore this point further, results for a subset of
AeroCom-II models (using earlier versions of models in the ATom ensemble) show only a
slight reduction (~10%) in the model spread, with however some regional differences i.e.
an improved agreement with observations in the MBL, but an increase in the model bias
and spread in the LS (Figure S2). Thus, model improvement for the more recent models
appears to be the main reason for the reduced spread.

## 4.2   Evaluation of predicted OA vertical distribution

Figure 4 compares the mean vertical profiles of OA measured during ATom-1 and -2 with the predictions of the model ensemble average based on the eight ATom models (Table 1) and 28 AeroCom-II models for the different latitudinal regions of the Pacific and Atlantic basins. Note that the use of a wide logarithmic scale (to be able to span all the observations) may make the observed differences appear small, although they often reach factors of 2-10 and larger (Figure S5 shows the results on a linear scale). For AeroCom-II, large latitudinal differences exist in the results with a better performance closer to source regions and large disagreement in the lower stratosphere and remote regions, as already suggested by the mission medians shown in Figure 3. The best AeroCom-II model performance is found over the equator in both basins, where the model ensemble captures within a factor of 2 the observed OA concentrations throughout the troposphere in the Pacific basin, and matches remarkably well the observations in the lower troposphere of the Atlantic basin that is heavily influenced by biomass burning emissions. Reasonable agreement is found for the OA vertical distribution over the NH Atlantic and Pacific oceans, especially in the lower troposphere (< 4 km). The largest model discrepancies (1-2 orders of magnitude) are found in the remote regions of the Southern Ocean and SH mid-latitudes during both seasons and basins. The model overestimation is also large over the NH mid-latitude Pacific basin in the upper troposphere. A spread of 2-3 orders of magnitude is observed around the ensemble average indicating a very large variability in individual model predictions. This evaluation of AeroCom-II models in remote regions is an extension of that performed at the surface for urban and remote stations by Tsigaridis et al. (2014) (as in that previous study, the data and model simulations compared are not synchronous in time). The tendency of the model ensemble to overpredict OA concentrations by a factor of 2 on average in the remote regions is consistent with the transition from the large underprediction in OA near the source region to a slight overprediction of OA in remote continental sites that was reported for most AeroCom-II models (Tsigaridis et al., 2014), and also observed for default parameterizations in other studies (Heald et al., 2011; Hodzic et al., 2016).

By comparison, the results of the ATom model ensemble show a much better agreement with observations. The model spread is still substantial, but mostly below a factor of 5. Figures S6 and S7 show OA vertical profiles for individual ATom models and the spread in their results. In most regions, the ATom model ensemble captures reasonably well both

the absolute concentrations as well as the shape of the vertical profiles. In the biomass
burning outflow and NH mid-latitude regions, the ATom ensemble average better captures
the higher OA concentrations in the boundary layer and lower OA concentrations in the
lower stratosphere than the AeroCom-II ensemble. We note that using the ensemble
median OA profiles instead of ensemble mean OA profiles (as shown in Figure 5 and S7)
results in a slightly lower values of OA but does not change the conclusions of the model-
measurement comparisons (Figure S18).

### *4.3   Oxidation level of organic aerosols (OA/OC ratios)*

In addition to OA mass concentrations, we also evaluate the model's ability to simulate
their degree of oxygenation, an indicator of their oxidation and aging (Aiken et al., 2008;
Kroll et al., 2011). Ambient measurements of the oxidation level of organic particles are
limited (Aiken et al., 2008, Canagaratna et al., 2015), and the ATom dataset provides the
first global distribution of O/C and OA/OC ratios for the remote aerosol. The OA/OC ratio
is an estimate of the average molecular weight of organic matter per carbon weight, and
it mostly depends on the oxygen content (i.e. the O/C ratio), in the absence of significant
concentrations of organonitrates and -sulfates. It is needed to compare measurements of
organic aerosol mass (from e.g. AMS) with organic carbon measurements (from e.g.
thermooptical methods). It is also needed to compare the various types of measurements
to model concentrations, which are sometimes carried internally as OA and sometimes as
OC. A low OA/OC ratio is indicative of freshly emitted OA from fossil fuel combustion
(typically ~1.4), and its value increases with increased processing of organics in the
atmosphere.  Figure 5 shows that in the remote regions the bulk of measured OA/OC
ratios during ATom-1 and -2 range between 2.2 and 2.5, and are larger than values of 2.1
$\pm$ 0.2 found in the polluted US continental outflow regions that were sampled during
SEAC4RS, WINTER and DC3 field campaigns (Schroder et al., 2018). These values
indicate that remote OA is highly oxidized and chemically processed.
Note that for organosulfates (R-O-$SO_2$H and organonitrates (R-O-$NO_2$, p$RONO_2$ in the
following) only one oxygen is included in the reported OA/OC, as the fragments of these
species are typically the same as for inorganic species in the AMS (Farmer et al., 2010).
However in ATom organosulfates are estimated to account for ~1% of the total sulfate
(based on PALMS data, see Liao et al., 2015 for the methodology). Since sulfate and OA
concentrations are comparable, organosulfates would only increase the OA/OC by ~1%
on average. Organonitrates are reported from the AMS for ATom. Their impact on OA/OC
is not propagated for the default values, to maintain consistency with a large set of OA/OC
measurements by AMS in the literature, and since they would increase OA/OC on average
by only 4.5% (ATom-1) and 2.2% (ATom-2), which is smaller than the uncertainty of this
measurement. However, we show the results with both methods in Fig. 5 to fully document
this topic.
Importantly, this ratio is also used to calculate the total OA mass concentration for models
that provided their outputs in terms of organic carbon concentrations ($[OA]_i$ = $[OC]_i$ x
OA/OC$_{ratio}$). Most Models use a constant OA/OC ratio, but the value used varies
substantially. OA/OC of 1.4 is used in ECHAM6-HAM, whereas 1.8 is used in GEOS5 and
GC10-TOMAS simulations for both POA and SOA. Other models calculated directly SOA
concentrations without applying this conversion (CESM1-CARMA, CESM2-SMP, CESM2-
DYN, GC12-REF and GC12-DYN), but for POA used the ratio of 1.8 (CESM1-CARMA,
CESM2-SMP, CESM2-DYN) and 2.1 (GC12-REF and GC12-DYN). Most of the AeroCom-
II models used the ratio of 1.4 for all primary and secondary OA (Tsigaridis et al., 2014).
The comparison with measurements shows that the measured values are ~40% larger
than those assumed in some of the ATom models, and 60-80% larger than used in
AeroCom-II models. The comparison between the observed and predicted OA/OC vertical
profiles (Fig. S3) shows that AeroCom-II models tend to generally underpredict this ratio,
and do not capture its increase in remote regions. As a result, this underestimation of
OA/OC ratios and the use of a constant value could substantially impact the comparisons
of OA mass concentrations for several models considered in this study (ECHAM6-HAM,
GEOS5, CESM1-CARMA and GC10-TOMAS). If we correct for the underestimated
OA/OC ratio using the ATom measured values of 2.2 (to be conservative) and compare
to previously discussed biases in Table 2, the overprediction of the ECHAM6-HAM model
is increased to ~110-160%, and that of GC10-TOMAS to 180% during ATom-2 while
having ~15% bias in ATom-1, whereas GEOS5 results now overestimate up to 30% during
ATom1, and perform much better during ATom-2.
These results demonstrate that current global chemistry-climate models use unrealistically
low OA/OC ratios, which results in a large underestimate of the degree of oxidation of OA
in remote regions. Inaccurate prediction of OA oxidation as it ages could impact not only
the calculations of OA burden, but also its optical properties as the absorption of OA
changes with its degree of oxidation (through the formation and destruction of brown
carbon, Laskin et al., 2015, Forrister et al., 2015). However, models used in this study did
not include these effects.

### 4.4   Contribution of primary vs. secondary OA

We further assess whether global models can adequately predict the relative contributions
of primary and secondary OA. We strive to quantify these fractions with the most
straightforward methods (with the fewest assumptions) for both models and
measurements. POA concentrations were estimated from the BC measurements by using
an emission ratio appropriate to the airmass origin (biomass burning vs. anthropogenic),
as quantified by the f(BB) mass fraction from the PALMS single particle instrument (see
Section 3.2), with f(BB)=1 taken as a BC and OA being of pure BB airmass origin and
f(BB)=0 exclusively from a non-biomass burning source. By using the POA/BC ratio at the
source regions after most evaporation, but before POA chemical degradation evaporation
has taken place, we implicitly assume POA to be chemically inert, while in reality it can
slowly be lost to the gas-phase by heterogeneous chemistry (e.g. George and Abbatt,
2010; Palm et al., 2018). Thus, the observation-based method provides an upper limit to
the fraction of POA. The model/measurement comparison is only shown for the CESM
and GEOS-Chem model variants, as other participating models do not separate or did not
report their POA and SOA fractions. In all simulations, POA was treated as a chemically
inert directly emitted primary aerosol species that only undergoes transport,
transformation from hydrophobic to hydrophilic state with ageing (1-2 days typically),
coagulation, and dry and wet deposition. Importantly, the treatment of POA as non-volatile
(rather than semi-volatile) in models is fully consistent with the assumptions for POA
estimation from the measurements.
Figure 6 compares the vertical profiles of measurement-derived POA during ATom-1 and
predicted by the CESM2-DYN model over clean remote regions of the Pacific basin and
northern polar Atlantic that are not influenced by biomass burning. Comparisons for other
models are similar (not shown). Observations show that POA is extremely small in remote
regions, whereas the model predicts that about half of the OA is made of POA in those
areas. Although the model reproduces quite well the measured total OA, it tends to
severely overpredict the amount of POA and underpredict that of SOA over clean remote
regions (with the two errors canceling each other when it comes to total OA). Over the
biomass burning regions (not shown here) it can be difficult to directly quantify POA and
SOA with this method, as total OA remains about constant, while POA decreases with
aging and SOA increases (Cubison et al., 2011; Jolleys et al., 2015; Hodshire et al.,
2019b). However, given this evolution the method used here would lead to an
overestimate of POA for this reason.
A more general comparison is made in Figure 7, using the frequency distributions of the
measured and simulated fraction of POA/OA, for the free troposphere only (Figure S8
shows the corresponding cumulative distributions). Observations indicate that most
remote FT airmasses contain less than 10% POA, except for biomass burning plumes that
are considered mostly primary. A slightly higher proportion of POA is seen in ATom-2,
which is consistent with a slower photochemical production of SOA during NH winter.
These results indicate that the remote OA is consistently dominated by SOA regardless
of the season and location. The comparison with models reveals a very large discrepancy
in the predicted vs. measured POA vs. SOA contributions. Models have a general
tendency to severely overpredict the fraction of POA and underpredict that of SOA,
displaying a much wider frequency distribution than the measurements (as also shown for
POA and SOA vertical profiles for individual models on Figures S6 and S7). In GC12-REF,
CESM2-DYN and CESM1-CARMA (without improved in-cloud removal) predictions for
ATom-1, more than a half of the remote OA is POA, while that is very rarely observed in
the free troposphere (possibly only during strong biomass burning events). Most models
fail to reproduce the overwhelming dominance of SOA that is inferred from the
measurements during ATom-1, while the discrepancies are less severe during NH winter
(ATom-2). These seasonal differences suggest that model errors could be partially due to
inefficient production of SOA and/or too high POA emissions, although removal errors also
probably play a major role (see next section).
The differences are so large that they are pretty insensitive to details of the POA estimation
method from the measurements, mostly because for the vast majority of the ATom track
BC/OA ratios were extremely low and hence the exact magnitude of the multiplicative
factor is secondary to the estimation of POA (Figure S11). As Figure S9 illustrates, the
choice of $FF_{ratio}$ has very little impact on the overall distribution of POA. On the other hand,
while the $BB_{ratio}$ does impact the overall distribution of POA, it mostly affects the points in
the vicinity of the large Atlantic plumes. Since the POA/BC ratio in those plumes is fairly
low, (see Section 3.2), using a very large $BB_{ratio}$ mostly leads to an increase of the fraction
of the points where POA > 100%. While the large range of published $BB_{ratio}$ for different
sources precludes a more accurate estimation by our method, for the purposes of the
comparison with the model results we emphasize that even using the largest $BB_{ratio}$,
fraction of SOA is still significantly larger in the ATom dataset that in any of the models.
Additional sensitivity tests were performed to investigate the impact of noisy data and
uncertainties of f(BB) on the estimation of POA. Figure S11 clearly shows that the impact
of a misattribution of the aerosol type by the stated PALMS uncertainty (Froyd et al., 2019)
is completely negligible. Figure S10 details how the choice of averaging interval (with
longer averaging times reducing both the fraction of OA measurements under the DL and
below zero) impact the distribution of POA. Overall, no large changes are observed for
averaging times >5 min, and hence a 5 min averaging interval was used for the analysis
in Figure 7. Figure S10 also illustrates how capping the histogram impacts the POA
distribution. To capture the most realistic $f_{(POA)}$ distribution, the data in Fig 7 was capped
at the extremes (so $f_{(POA)}<0$ is taken as $f_{(POA)}=0$, and $f_{(POA)}>1$ is taken as $f_{(POA)}=1$). As Figure
S10 shows, data with $f_{(POA)}<0$ is almost exclusively due to very small (and always positive,
since BC cannot go negative) POA values being divided by small, negative noise in total
OA, and hence treating that fraction of the histogram as essentially fPOA~0 is justified.
On the other end of the distribution, data where POA is larger than OA is mostly due to
our average $BB_{ratio}$ being larger than the one encountered in most of the BB plumes in
ATom. Choosing a lower $BB_{ratio}$, as Figures S9b and S9d illustrate, leads to $f_{(POA)}>1$
basically trending to zero, confirming our interpretation. This is a limitation of the dataset,
and it does not seem appropriate to remove these points, since some fraction are likely
dominated by POA. However, it shows that the POA estimation, especially for this part of
the distribution likely overstates the importance of POA.
A comparison between simulations that have the same treatment of POA, and only differ
in their chemistry and removal of SOA (e.g. CESM2-SMP vs. CESM2-DYN; GC12-REF
vs. GC12-DYN) indicate that a more complex SOA treatment does not always result in a
more accurate simulation of the primary/secondary character of OA, a result that was also
found in the AeroCom-II multi-model intercomparison (Tsigaridis et al., 2014).
Finally, we have examined whether the non-volatile treatment of POA in models could
lead to these unrealistically high POA fractions in the remote regions. Figure S16 shows
a comparison of POA vertical profiles as predicted by the GC12-REF simulations that use
non-volatile POA and a sensitivity simulation GC12-REF-SVPOA that uses semi-volatile
POA similar to the standard treatment in GEOS-Chem as described in Pai et al. (2020).
Note, however, that Pai et al. (2020) included marine POA emissions, used different
reanalysis meteorology, and a different model version (12.1.1 rather than 12.0.1 here), so
their resulting comparisons to ATom measurements are somewhat different than found
here for GC12-REF-SVPOA. The comparison indicates that the POA concentrations
increase substantially in most regions when the semi-volatile POA parameterization is
used. These results suggest that non-volatile treatment of POA is not responsible of the
model bias.
*4.5    Sensitivity to OA formation and removal*
In this section, we further investigate some of the possible reasons for the incorrect model
predictions of the relative contributions of POA and SOA in remote regions. Given the
tendency of models to underestimate OA close to anthropogenic source regions and
overestimate OA downwind in past studies (e.g. Heald et al., 2011; Tsigaridis et al., 2014;
Hodzic et al., 2016), in this section we investigate the sensitivity of OA to increasing
sources and increasing removals. We have performed two additional model simulations
to test the sensitivity of the POA/SOA fractions to uncertainties in the representation of (i)
wet scavenging, based on the CESM1-CARMA simulations in which we have removed
the improvements in the aerosol removal by the convective updrafts (Yu et al., 2019); and
of (ii) SOA formation based on the GC12-REF simulations in which we have replaced the
SOA formation VBS mechanism (Pye et al., 2010) by an updated VBS mechanism that
uses chamber wall-loss corrected SOA yields (Hodzic et al., 2016, same formation
scheme that is used in GC12-DYN and CESM2-DYN runs, but with removals kept identical
to GC12-REF). The results of these two sensitivity simulations are displayed on Figure 8,
which shows measured and predicted mass concentrations of OA, POA, SOA and sulfate
for ATom-1 as a function of the number of days since the air mass was processed through
convection. One should keep in mind that this is an averaged plot that included airmasses
from various regions and altitudes, and not a Lagrangian plot following the same airmass.
***Sensitivity to in-cloud scavenging in convective clouds.*** Inefficient wet removal of
primary OA could contribute to the POA overprediction in global models, especially in the
tropics. Previous global model studies have reported two to three orders of magnitude
overestimation of primary carbonaceous species such as BC in the free troposphere when
the removal in the convective updrafts was not included (e.g. Schwarz et al., 2013, Yu et
al., 2019). A strong reduction due to convective removal is also expected for POA
concentrations, as POA is a primary species co-emitted with BC at the surface and
internally mixed with it (Lee et al., 2015), and that is typically coated by secondary
inorganics and organics over short timescales (Petters et al., 2006; Jiang et al., 2010;
Wang et al., 2010). Figures 7a and 8 compare the simulations of CESM1-CARMA with
and without improved convective in-cloud scavenging during ATom-1. The improved in-
cloud scavenging scheme considers aerosol activation into cloud droplets from entrained
air above the cloud base, which is more realistic and results in a more efficient removal of
aerosols in the upper troposphere by convection. E.g. a two order of magnitude reduction
in BC in the upper FT was reported by Yu et al. (2019), resulting in much improved
agreement with observations. Similar results were observed for sea salt aerosols in
Murphy et al. (2019). Figure 8 shows that all submicron aerosol species simulated in
CESM1-CARMA are strongly impacted by the in-cloud removal above the cloud base.
POA concentrations are reduced by an order of magnitude while sulfate is reduced by
30% leading in both cases to a much-improved agreement with observations. SOA is
reduced by ~30% as well, which leads to an underprediction of measured SOA
concentrations. The overall impact on OA concentrations is a significant reduction, which
leads to ~20% underestimation of OA in the aged remote air during ATom-1.
For the CESM2-DYN model that does not have improved in-cloud removal, the reasonable
agreement (within 20%) with the observed OA concentrations thus results from
coincidental error compensation between the overpredicted POA and underpredicted
SOA. The prescribed SOA formation and the artificial 50% adjustment of SOA emissions
based on Liu et al. (2012) in CESM2-SMP leads to an overestimation of observed SOA in
aged remote airmasses.
***Sensitivity to SOA formation.*** In addition, we have also tested the sensitivity of the OA
composition to the choice of the SOA formation mechanism. Figure 8 compares the results
of the GC12-REF model that uses SOA formation yields derived from traditional chamber
experiments (Pye et al., 2010) and those corrected for loses of organic vapors onto
chamber walls as proposed in Hodzic et al. (2016). Previous studies have reported that
chamber wall losses could lead up to a factor of 4 underprediction of formed SOA (Zhang
et al., 2014; Krechmer et al., 2016). It should be noted that in both cases, isoprene-SOA
is formed in aqueous aerosols following Marais et al. (2016). The comparison shows a
factor of 3 increase in SOA concentrations when the updated SOA formation is considered
leading to a much better agreement with the observed SOA as well as the observed total
OA. GC12-REF predicts well the amount of POA and overpredicts somewhat the amount
of sulfate aerosols, which is expected as it already includes the improved aerosol removal
in convective updrafts (Wang et al., 2014). Figure S6 also shows that POA vertical
distribution is well captured in GEOS-Chem in most regions, except over the polar north
Pacific. It should be noted that these results are consistent with the POA/OA frequency
distribution shown in Figure 7 (the POA/OA ratio predicted by GC12-REF is larger than
the measured ratio, which is consistent with the fact that POA is about the right amount,
and OA is underpredicted in Figure 8).
These sensitivity simulations suggest that a stronger convective removal of POA and a
stronger production of SOA might be needed to correctly represent not only the total OA
concentrations but also its primary and secondary nature in remote free troposphere and
remote ocean regions. Accurate predictions of the OA concentration, composition, and
source contributions for the right reasons are key for accurately predicting their lifecycle
and radiative impacts. Only when there is confidence that the sources are accurately
predicted, we can have confidence in OA predictions for pre-industrial and future
conditions, as well as to evaluate PM mitigation strategies.
### *4.6 OA and sulfate relative contributions in FT*
Finally, we assess the model ability to predict relative amounts of OA and sulfate in the
free troposphere where they are the two major constituents of the submicron aerosol
(Figure 2b). Accurate predictions of their relative contributions are crucial to determine the
hygroscopicity of the submicron aerosol, and its ability to serve as a cloud condensation
nuclei (CCN) in the remote free troposphere (Carslaw et al., 2013; Brock et al., 2016).
Figure 9a compares the average measured relative fractions of sulfate (36%) and
carbonaceous aerosols (OA=59% and BC=5%) in the FT with those predicted by
individual models during ATom-1. The CESM2 models best reproduce the observed
relative contributions, with a slight underestimation of OA (57% instead of 59%) for
CESM2-DYN, and a slight overestimation of OA (63% instead of 59%) for CESM2-SMP.
GEOS5 has 15% more OA relative to sulfate than observed. All other models
underestimate both OA and BC relative fractions. For instance, in GC12-REF and -DYN,
both the BC and OA fractions are ~40% (relative) lower than observed.
Figure 9b shows the frequency distribution of observed and predicted fractions of OA
relative to sulfate during ATom-1 and -2 in the free troposphere. Most models fail to
reproduce the relatively uniform nature of the observed distributions during ATom-1, with
typically narrower model shapes around a preferred ratio. The NH summer measurements
indicate that OA > sulfate in ~55% of the samples (consistent with Fig. 2b), while models
generally tend to underestimate the relative OA contribution. In particular, GEOS-Chem
and ECHAM6-HAM tend to overestimate the relative contribution of sulfate. A better
agreement is found for GEOS5, CESM1-CARMA and CESM2-DYN, which follow more
closely the shape of the observed distribution. The comparisons also suggest that the
more complex SOA treatment of SOA formation and removal proposed by Hodzic et al.
(2016) in the same host model leads to an improved agreement with observations (e.g.
CESM2-DYN vs. CESM2-SMP; GC12-DYN vs. GC12-REF). It should be noted that
CESM2-SMP uses fixed SOA yields that were increased by 50% as suggested by Liu et
al. (2012), leading to an overestimation of the relative contribution of OA compared to that
of sulfate in the free troposphere. During the NH winter (ATom-2), measurements show a
somewhat higher proportion of sulfate aerosols (vs. ATom-1), which is consistent with a
slower production of SOA in the NH during winter and a reduced influence of biomass
burning. Similar conclusions are found for the evaluation of different models. It is worth
mentioning that the comparison performed for the whole ATom-1 and 2 dataset (not
shown) leads to similar results with even slightly stronger overestimation of the sulfate
relative contribution compared to OA.
The discrepancies between the observed and predicted composition of submicron aerosol
over remote regions can be quite large for other constituents as well. Figure 10 shows the
comparison of measured and predicted composition of the submicron aerosol over the
Southern Ocean (during the NH winter) where the disagreement in simulated sea salt,
nitrates, ammonium, and MSA often exceeds the contribution of OA. While the
observations show a more uniform distribution of non-marine aerosol with higher values
in the mid and upper troposphere, respectively, most models tend to simulate highest
fractions of OA (and sulfate) towards the tropopause. This may also be explained by the
uncertainties in modeled wet removal of aerosol that has been discussed above. Specific
studies have discussed and continue to investigate the ATom measurements and
simulations of different components in more detail, including particle number (Williamson
et al., 2019), black carbon (Katich et al., 2018; Ditas et al., 2019), MSA (Hodshire et al.,
2019), sulfate-nitrate-ammonium (Nault et al., 2019), and sea salt (Yu et al, 2019; Bian et
al., 2019; Murphy et al., 2019).

## 5    Conclusions and implications

Our understanding and representation in global models of the lifecycle of the OA remain highly uncertain, especially in remote regions where constraints from measurements have been very sparse. We have performed a systematic evaluation of the performance of eight global chemistry climate models and of 28 AeroCom-II models in simulating the latitudinal and vertical distribution of OA and its composition in the remote regions of the Atlantic and Pacific marine boundary layer, free troposphere and lower stratosphere, using the unique measurements from the ATom campaign. Our simulations are conducted for both ATom-1 and ATom-2 deployments that took place in August 2016 and February 2017, respectively. The main conclusions of the comparison are as follows:

- The AeroCom-II ensemble average tends to be biased high by a factor of 2-5 in comparison to measured vertical OA profiles in the remote atmosphere during both NH summer and NH winter. The ensemble spread increases from a factor of 40 in the NH source regions to a factor of 1000 in remote regions of the Southern Ocean. The evaluation of AeroCom-II models in the remote regions provides an extension of the previous evaluation with continental ground data by Tsigaridis et al. (2014). We note that the data from the AeroCom-II models were based on monthly mean values from a different simulated year than the ATom campaigns; however, the consistent model biases are strong enough that we would not expect our conclusions to change for a different modeled year.

- The results of the ATom model ensemble used in this work show a much better agreement with the OA observations in all regions and reduced model variability. However, some of the agreement is for the wrong reasons, as most models severely overestimate the contribution of POA and underestimate the contribution of SOA to total OA. Sensitivity simulations indicate that the POA overestimate in CESM could be due to an inadequate representation of primary aerosol removal by convective clouds, (additional convective removal per Yu et al. (2019) in CESM1-CARMA led to a better agreement with observations). Most models have insufficient production of SOA, and sensitivity studies indicate that a stronger production of SOA is needed to capture the measured concentrations. The photochemical ageing of POA which was not considered here (unlike for SOA) could also contribute to the model overestimation. The non-volatile POA treatment in models is consistent with the assumption of inert POA particles used to estimate POA from measurements, and cannot explain the

model bias. Indeed, sensitivity simulations with semi-volatile POA lead to a much
larger model bias for OA in the upper troposphere and remote regions. The
compensation between errors in POA and SOA in remote regions is however a
recurring issue in OA modeling (de Gouw and Jimenez, 2009). For instance, it was
found in the urban outflow regions such as Mexico City during MILAGRO 2006 field
campaign (Fast et al., 2009; Hodzic et al., 2009); Paris during MEGAPOLI 2009
(Zhang et al., 2013); the Los Angeles area during CalNex-2010 (Baker et al., 2015;
Woody et al., 2016); the NE US outflow during WINTER 2015 (Schroder et al., 2018;
Shah et al., 2019).

•   Additional errors in simulated OA concentrations can arise from the use of too low
OA/OC ratios when model results (often calculated as OC) are converted to OA for
comparison with measurements. We note that OA is the most atmospherically-relevant
quantity, while OC is an operational quantity, partially a relic from a period in which
only OC could be separately quantified (although also of some use for carbon budget
studies). It should also be noted that most emission inventories still use OC as the
primary variable, which is why the use of accurate OA/OC ratios is still key for all
models. We show that the OA/OC ratio used in most models is too low compared to
measured values that range mostly from 2.2 to 2.5, resulting in errors in OA mass of
~70% for AeroCom-II models and ~30% for current models that use organic carbon to
track OA mass. Remote OA is thus highly oxidized and chemically processed. These
results demonstrate that current global chemistry-climate models underestimate the
degree of oxidation of OA in remote regions and need to consider further chemical
ageing of OA, which could impact the calculations of its burden, and optical and
hygroscopic properties.

•   The results also show that in most models (except CESM2) the predicted OA
contribution to the total submicron aerosol is underestimated relative to sulfate in the
remote FT where OA and sulfate are the dominant submicron aerosols (important for
climate). Accurate predictions of composition of submicron particles remains
challenging in remote regions and should be the topic of future studies.

Key implications of our results are: (i) Model errors on the relative contribution of POA and
SOA to OA reduce our confidence on the ability to simulate radiative forcing over time or
OA health impacts; (ii) Model errors for the relative contributions of sulfate and organics
to the submicron aerosol in the free troposphere could lead to errors in the predicted CCN
or radiative forcing of aerosols as inorganics are more hygroscopic than OA; (iii) the OA
system seems to be more dynamic with a need for an enhanced removal of primary OA,
and a stronger production of secondary OA in global models to provide a better agreement
with observations.
**Acknowledgements.** The authors want to thank the ATom leadership team and the
NASA logistics and flight crew for their contributions to the success of ATom. Authors
acknowledge Dr. Rebecca Buchholz (NCAR) for providing the emissions used for the
CESM2 simulations. We thank C. Brock and C. Williamson (NOAA) for the aerosol volume
data, Paul Wennberg (Calthech) for HCN data and Eric Apel and Rebecca Hornbrook
(NCAR) for $CH_3CN$ data used in Fig, S20. The ATOM measurements and analyses were
supported by NASA grants NNX15AH33A, NNX15AJ23G, and 80NSSC19K0124. AH was
supported by the National Center for Atmospheric Research, which is operated by the
University Corporation for Atmospheric Research on behalf of the National Science
Foundation. JRP and JKK were supported by the US Department of Energy's Atmospheric
System Research, an Office of Science, Office of Biological and Environmental Research
program, under grant no. DE-SC0019000. This project has received support from the
European Research Council under the European Union's Horizon 2020 research and
innovation programme (grant agreement No. 819169), and from EPA STAR grant
83587701-0. This manuscript has not been reviewed by EPA, and no endorsement should
be inferred. We would like to acknowledge high-performance computing support from
Cheyenne provided by NCAR's Computational and Information Systems Laboratory. We
thank C. Brock for the aerosol volume data, and D. Murphy for useful discussions. We
thank the ATom leadership team, science team and the NASA DC-8 crew for their
contributions to the success of the ATom mission.
**Code/Data availability:** Data can be obtained from the ATom website:
https://doi.org/10.3334/ORNLDAAC/1581.
L2 Measurements from CU High-Resolution Aerosol Mass Spectrometer (HR-AMS) can
be obtained from the ORNL DAAC, Oak Ridge, Tennessee, USA.
https://doi.org/10.3334/ORNLDAAC/1716.
**Author contribution:** A. Hodzic, P. Campuzano-Jost and J.L. Jimenez performed the
measurement / model comparisons, wrote and revised the manuscript. P. Campuzano-
Jost, D.A. Day, B.N. Nault, J.C. Schroder, D.T. Sueper, and J. L. Jimenez performed and
analyzed the AMS measurements. K.D. Froyd and G.P. Schill performed and analyzed
the PALMS measurements. J.P. Schwarz and J.M. Katich performed the BC

measurements. H. Bian, M. Chin, P.R. Colarco, B. Heinold, A. Hodzic, D.S. Jo, J.K. Kodros, J.R. Pierce, E. Ray, J. Schacht, I. Tegen, S. Tilmes, K. Tsigaridis, and P. Yu provided model output. All authors provided comments on the manuscript.

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

Tables
Table 1: ATom global model configurations and their treatment of the most important processes affecting organic aerosols.

| Models & horizontal res. & met. fields & config. reference | Aerosol module | Submicron size[6] OA (dust/sea salt) | SOA precursors[1] | | | | | SOA production | Emission | POA/POC (SOA/SOC) | Removal | | |
|---|---|---|---|---|---|---|---|---|---|---|---|---|---|
| | | | ISO | MT | SQ | ANT | C>12 | | | | Standard[2] | Improved | Photolytic |
| CESM1-CARMA (1.9°lon x 2.5°lat) MERRA-2 (Yu et al. 2019) | 20 bins | < 500 nm (< 800 nm) | x | x | | x | | Semi-volatile using VBS (Pye et al. 2010) | GAIS and GFED v3 | 1.8 (N/A) | x | For convective updrafts (Yu et al. 2019)[3] | |
| CESM2-DYN (0.9° lon x 1.25° lat) GEOS5 (Tilmes et al. 2019) | 4 modes | < 270 nm (< 800 nm) | x | x | x | x | x | Semi-volatile using VBS (Hodzic et al. 2016) | CMIP6 and QFED v2.4 | 1.8 (N/A) | x | Water solubility of organic gases per Hodzic et al. (2014) | For SOA (Hodzic et al. 2016) |
| CESM2-SMP GEOS5 (0.9° lon x 1.25° lat) (Tilmes et al. 2019) | 4 modes | < 270 nm (< 800 nm) | x | x | | x | | Non-volatile with prescribed mass yields for all precursors[4] | CMIP6 and QFED v2.4 | 1.8 (N/A) | x | | |
| ECHAM6-HAM ECHAM6 (1.87°lon x1.87°lat) (Tegen et al. 2019) | 7 modes | < 500 nm (< 500 nm) | | | x | | | Non-volatile with 15% prescribed mass yields (Dentener et al. 2006) | ECLIPSE[5] and GFAS | 1.4 (1.4) | x | | |
| GC12-REF (2° lon x 2.5° lat) GEOS-FP (Bey et al. 2001) | Bulk | Bulk (< 500 nm) | x | x | x | x | | Semi-volatile using VBS (Pye et al. 2010); non-volatile isoprene- | CMIP6 and GFED v4 | 2.1 (N/A) | x | For convective updrafts per Wang et al. 2014 | |

| | | | | | | | | | | | | | |
|---|---|---|---|---|---|---|---|---|---|---|---|---|---|
| | | | | | | | | SOA (Marais et al. 2016) | | | | | |
| GC12-DYN (2° lon x 2.5° lat) GEOS-FP (Bey et al. 2001) | Bulk | Bulk (< 500 nm) | x | x | x | x | x | Semi-volatile using VBS (Hodzic et al. 2016); non-volatile isoprene-SOA (Marais et al. 2016) | CMIP6 and GFED v4 | 2.1 (N/A) | x | For convective updrafts (Wang et al. 2014); Water solubility of organic gases (Hodzic et al. 2014) | For SOA (Hodzic et al. 2016) |
| GC10-TOMAS (5° lon x 4° lat) GEOS-FP (Kodros et al. 2016) | 15 bins | < 316 nm (< 316 nm) | | x | | x | | Non-Volatile using 10% mass yields for MT, 0.2 Tg SOA per Tg CO for anthropogenic emissions | EDGAR v4 and GFED v3 | 1.8 (1.8) | x | For convective updrafts (Wang et al. 2014) | |
| GEOS5 (0.5°lon x 0.625°lat) MERRA-2 (Bian et al. 2019) | Bulk | bulk (< 1 μm for dust, 500 nm for seasalt) | x | x | | x | | Non-Volatile, 10% mass yields for all precursors | HTAP and QFED v2.54 | 1.8 (1.8) | x | | |

(1) SOA precursors include isoprene (ISO), monoterpenes (MT), sesquiterpenes (SQ), anthropogenics (ANT) including aromatics such as benzene, toluene and xylene, as well as lumped shorter chain alkanes and alkenes; and higher molecular weight n-alkanes and n-alkenes (C>12).

(2) Standard removal includes dry deposition and sedimentation, as well as convective and large-scale scavenging of soluble organic gases and aerosols, and below-cloud scavenging of aerosols.

(3) A sensitivity simulation is performed with CESM1-CARMA without the improved scavenging in convective updrafts.

(4) 5% for lumped C<12 alkanes, 5% for lumped C<12 alkenes, 15% for aromatics, 4% for isoprene, 25% for monoterpenes.

(5) Anthropogenic BC emission are replaced in Russia with the dataset of Huang et al. (2015).

(6) Submicron size range (diameter) used in various models for comparison with the AMS data.

Table 2: Comparison of observed and simulated OA concentrations along ATom-1 and
ATom-2 flights for eight global model simulations and their ensemble. The results of the
model ensemble are also indicated.  The statistical indicators are calculated as normalized
mean bias $NMB(\%) = 100 \times \sum_i (M_i - O_i)/\sum_i O_i$; normalized mean error $NME(\%) =$
$100 \times \sum_i |(M_i - O_i)|/\sum_i O_i$;    root    mean    square    error    $RMSE(\mu g\, m^{-3}) =$
$\sqrt{(1/N)\sum_i (M_i - O_i)^2}$ and correlation coefficient ($R^2$) between modeled ($M_i$) and observed
($O_i$) data points. The mean of ATom-1 observations is ~0.23 $\mu g\, m^{-3}$ and for ATom-2 is
0.11 $\mu g\, m^{-3}$. Figure S4 shows the normalized mean bias for all individual ATom model
simulations for various latitudinal regions and for both the Atlantic and Pacific basins.

| **Organic aerosols** | Avg.Mod. ($\mu g\, m^{-3}$) | NMB (%) | NME (%) | RMSE ($\mu g\, m^{-3}$) | $R^2$ | Avg.Mod. ($\mu g\, m^{-3}$) | NMB (%) | NME (%) | RMSE ($\mu g\, m^{-3}$) | $R^2$ |
|---|---|---|---|---|---|---|---|---|---|---|
| Model | *ATom-1 scores (August 2016)* | | | | | *ATom-2 scores (February 2017)* | | | | |
| AeroCom-II Ens. | 0.400 | 74.2 | 127.3 | 0.560 | 0.31 | 0.254 | 137 | 175 | 0.278 | 0.38 |
| AeroCom-II Sub.[1] | 0.335 | 47.0 | 111 | 0.557 | 0.28 | 0.242 | 127 | 178 | 0.290 | 0.27 |
| ATom Ensemble | 0.239 | -4.5 | 64.6 | 0.372 | 0.66 | 0.139 | 23 | 92.6 | 0.224 | 0.48 |
| CESM2-DYN | 0.268 | 4.6 | 83.7 | 0.867 | 0.47 | 0.140 | 25.6 | 111.7 | 0.317 | 0.36 |
| CESM2-SMP | 0.349 | 36.3 | 94.3 | 0.556 | 0.51 | 0.175 | 57.2 | 125.4 | 0.299 | 0.31 |
| CESM1-CARMA | 0.155 | -33.2 | 93.8 | 0.603 | 0.12 | 0.131 | 22.6 | 119.6 | 0.244 | 0.31 |
| ECHAM6-HAM | 0.400 | 73.6 | 143.6 | 0.714 | 0.24 | 0.214 | 100 | 184.0 | 0.363 | 0.23 |
| GC12-DYN | 0.142 | -32.6 | 79.4 | 0.560 | 0.16 | 0.174 | 14.7 | 96.6 | 0.312 | 0.39 |
| GC12-REF | 0.122 | -43.0 | 76.5 | 0.536 | 0.18 | 0.147 | 3.6 | 96.3 | 0.292 | 0.35 |
| GC10-TOMAS | 0.218 | -14.4 | 86.5 | 0.644 | 0.16 | 0.313 | 150.0 | 223.7 | 0.537 | 0.12 |
| GEOS5 | 0.242 | -5.4 | 86.6 | 0.975 | 0.38 | 0.084 | -24.9 | 86.4 | 0.268 | 0.29 |

(1) This is the subset of AeroCom-II model ensemble that includes only seven
models that are similar to those that are included in the ATom ensemble (either
the same model, or an older model version, or the same aerosol module).
AeroCom-II Sub. incudes CAM5-MAM3, CCSM4-hem, ECHAM5-HAM2,

1625   GEOSChem-APM 8.2, GEOSChem 9, GISS-TOMAS and GMI (see Tsigaridis
1626   et al., 2014 for their description).

Figures:

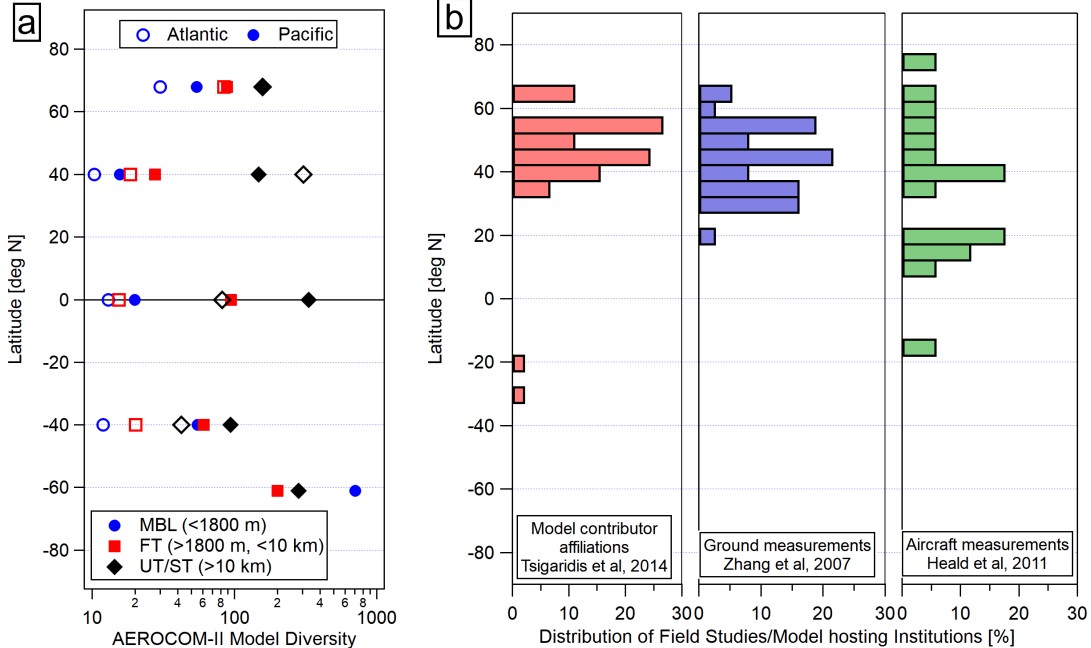


Figure 1: (a, left) The ratio between the average OA concentrations of the highest and the
lowest models (for each region) as predicted among 28 global chemistry transport models
participating in the AeroCom phase II intercomparison study (Tsigaridis et al. 2014); (b,
right) Geographical distribution of institutions at which the AeroCom-II models were
ran/developed (based on author affiliations) and of the field measurements included in two
major literature overview studies (Zhang et al., 2007; Heald et al., 2011) for the OA ground
and aircraft AMS as a function of latitude. For the aircraft campaigns, the average latitude
for the full deployment was taken.





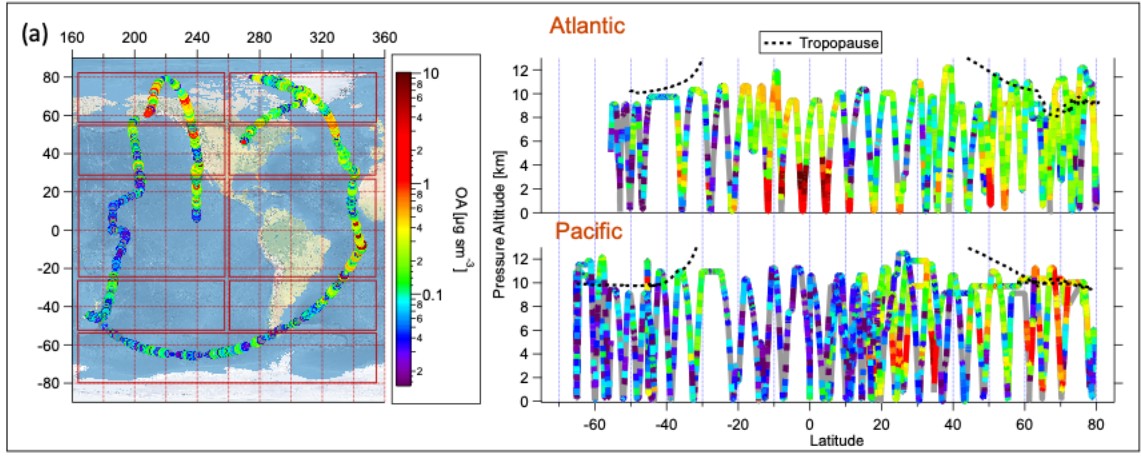

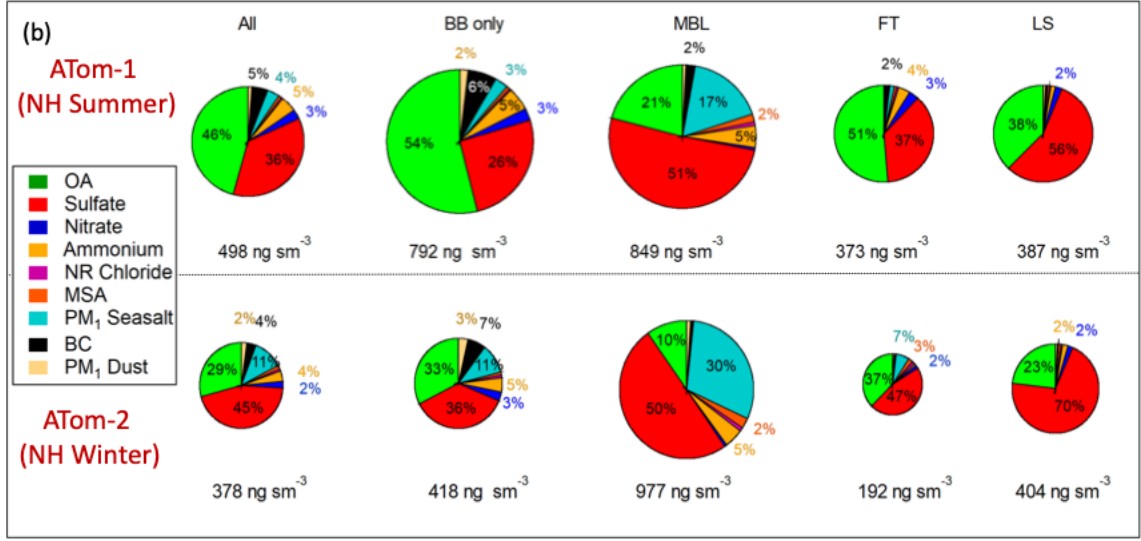

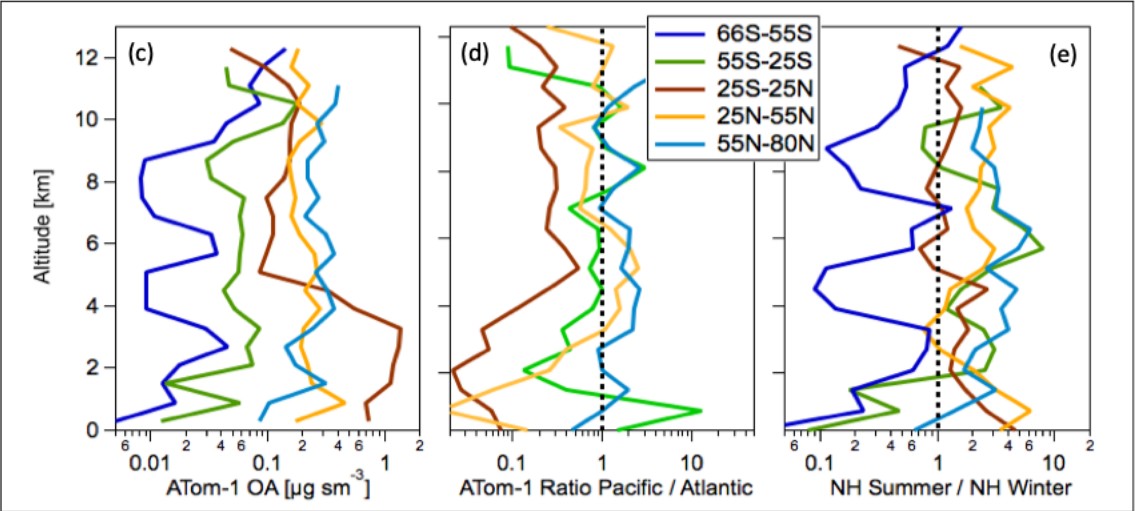


Figure 2: (a, left) ATom-1 DC-8 flights during the August 2016 deployment. Red boxes
indicate regions used for the latitude averaging of the model results. (a, right) Vertical
distribution of OA concentrations ($\mu$g sm$^{-3}$) along ATom-1 flight tracks (b) Average
submicron aerosol composition as measured in the biomass-burning influenced regions
(BB only), and the non-BB influenced regions including the marine boundary layer (MBL),
free troposphere (FT), and lower stratosphere (LS) for ATom-1 (upper plots) and ATom-2
(lower plots). The BB influenced airmasses were filtered using the PALMS data (see
section 3.1). Contributions below 2% are shown but not labeled on the pie chart graph. In
ATom-1, BB-only represents 24% of the data, clean MBL 8%, clean FT 57% and clean
UT 12%, whereas in ATom-2 BB-only represents 3%, clean MBL 8%, clean FT 74%, clean
UT 16%. (c) The average OA vertical profiles are shown for each latitude region as well
as (d) the ratios between the Pacific and Atlantic Oceans in each region. (e) The seasonal
contrast in OA concentrations as calculated as the ratio in OA concentrations between the
NH summer (ATom-1) and NH winter (ATom-2) campaigns. The corresponding plots for
ATom-2 can be found in Fig. S1.

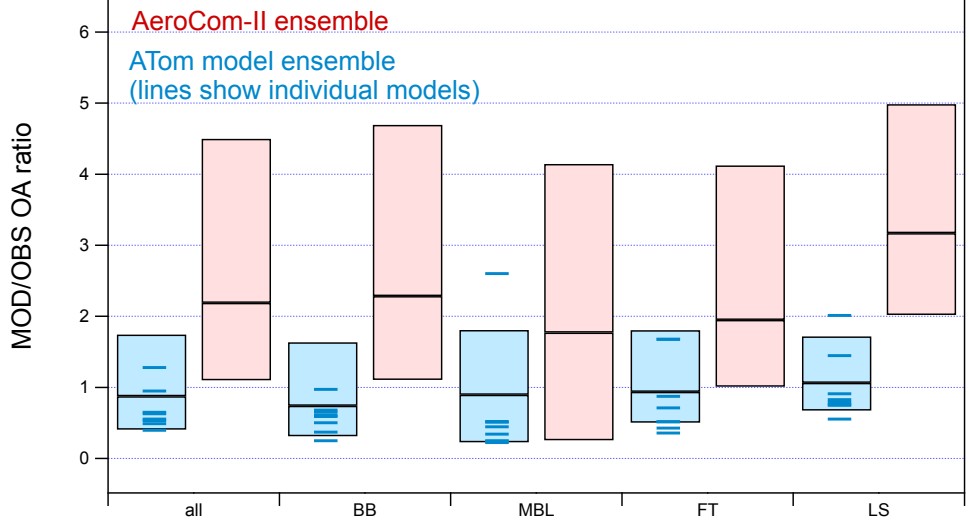

Figure 3: Ratios between predicted and observed OA concentrations for all ATom-1 flights
as calculated for the ATom and AeroCom-II model ensembles in different regions ("BB"
biomass burning influenced regions; "MBL" clean marine boundary layer; "FT" clean free
troposphere' and "LS" lower stratosphere). Median of the ensemble ratio is shown as a
horizontal line, while the boxes indicate 25th and 75th percentiles. Medians for the
individual models included in the current ATom model ensemble are also shown as blue
lines.

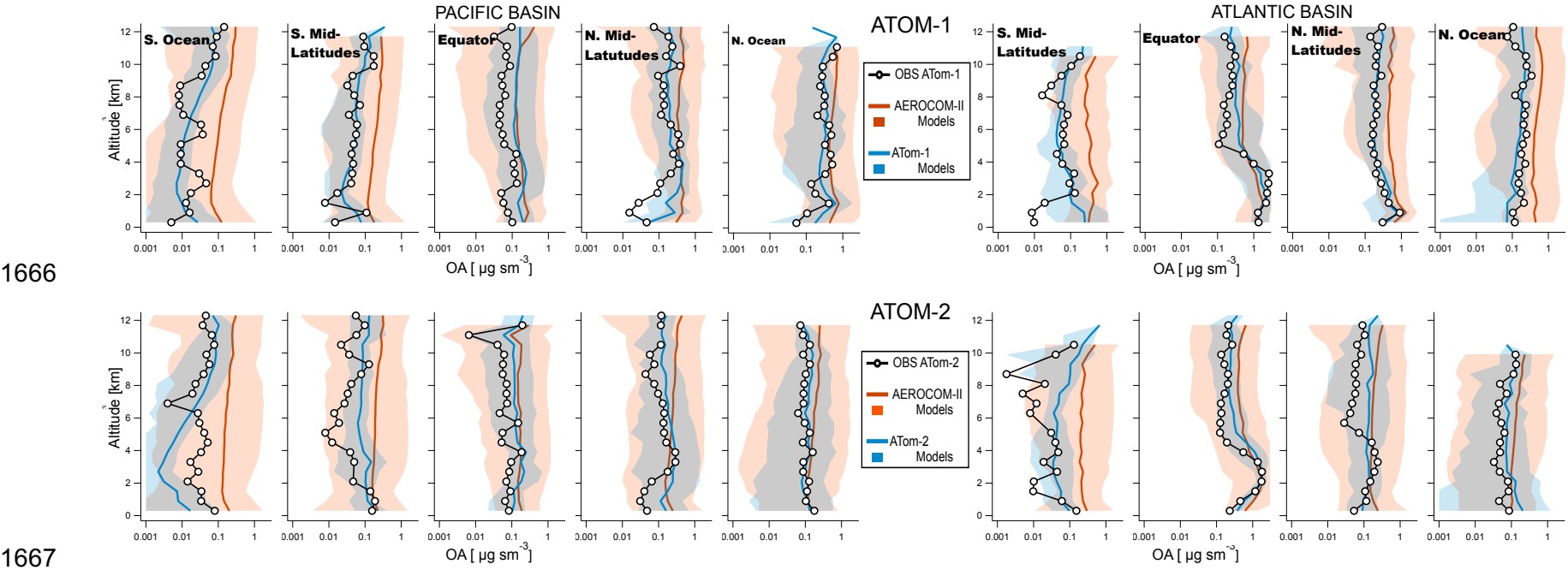



Figure 4: Comparison of latitude-averaged predicted OA vertical profiles with ATom-1 and -2 measurements taken over the Pacific (left side) and Atlantic (right side) basins. Results of the AeroCom-II model ensemble average are shown in red while those of the ATom model ensemble are shown in blue. Shaded areas indicate the variability (two standard deviations) within each model ensemble.


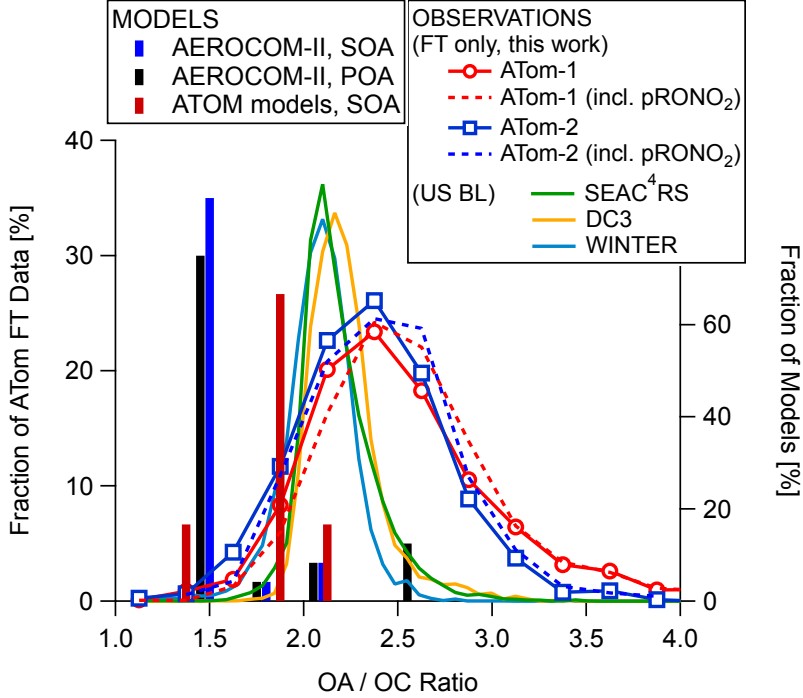


Figure 5: Distribution of the OA / OC ratio as measured during ATom-1 and -2. Values for
the recent aircraft campaigns (SEAC4RS, DC3 and WINTER) that took place over
continental US regions closer to continental source regions are also shown (Schroder et
al., 2018). The bars (right axis) show the OA/OC used for SOA and POA by the models
included in the AeroCom and ATom ensemble, with OA/OC=1.4 being the modal value for
the former and 1.8 for the latter.

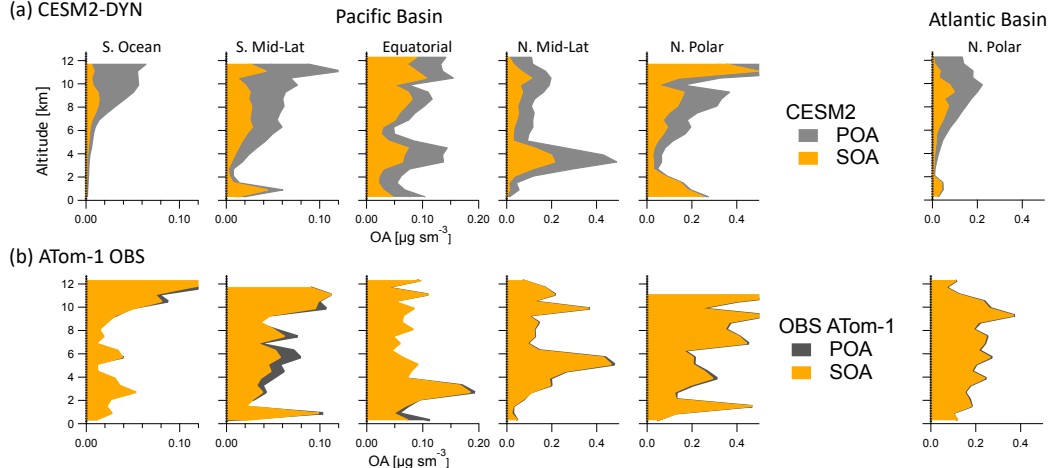


Figure 6: Comparison of averaged POA and SOA vertical profiles as observed during ATom and as predicted by the CESM2-DYN model over the non-BB influenced Pacific and Atlantic basins. The comparison is not shown for the strongly biomass burning influenced regions as all the OA is conservatively allocated to POA in those regions.

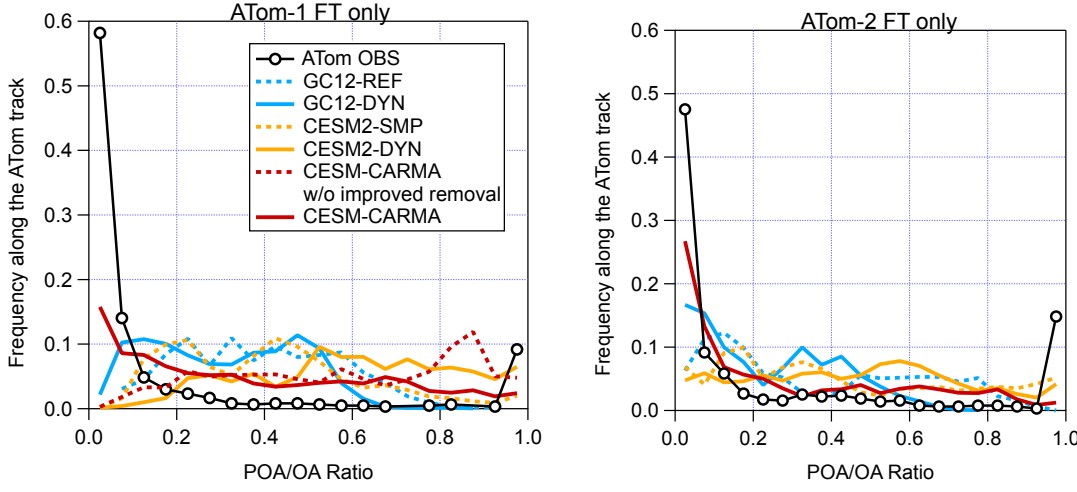

1684

Figure 7: Frequency distribution of observed and simulated ratio of POA to total OA in the free troposphere during ATom-1 and ATom-2 as computed by the GC12-, CESM2-, and CESM1-CARMA models.

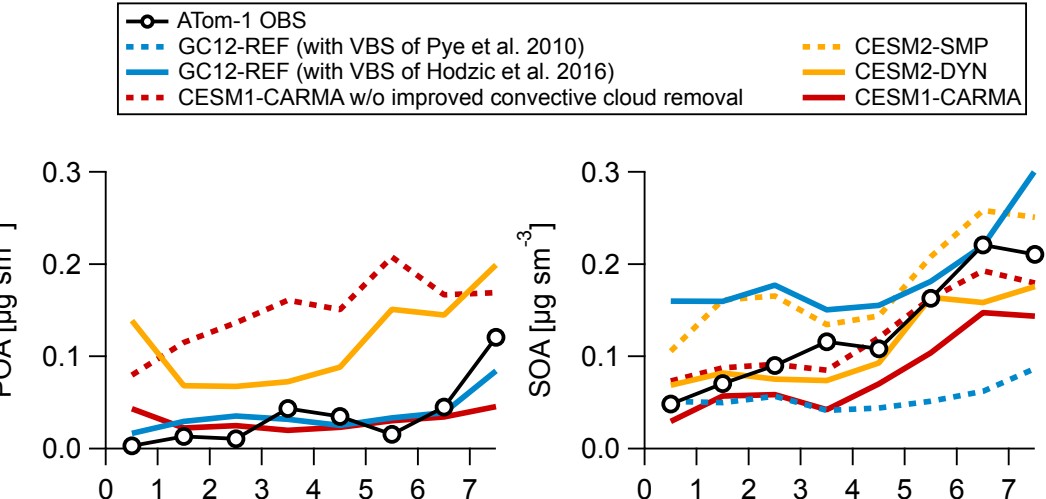

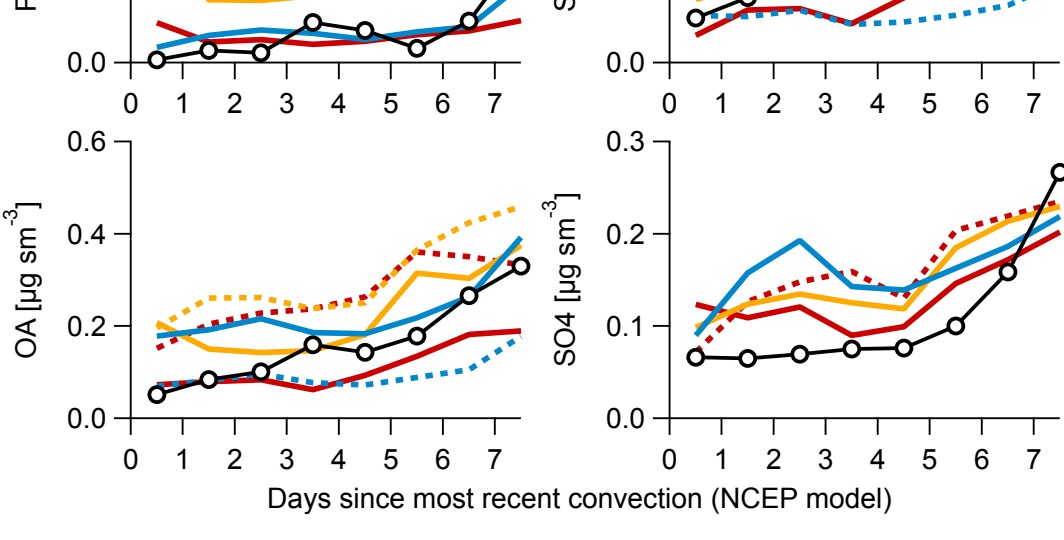

Figure 8: Measured and predicted mass concentrations of POA, SOA, OA and sulfate aerosols during ATom-1 as a function of the number of days since the air mass was processed through convection (based on a trajectory model from Bowman, 1993, and satellite cloud data from NASA Langley, https://clouds.larc.nasa.gov/). CESM2-SMP and CESM2-DYN have the same emissions and processing of POA and sulfate, and thus similar concentrations. The same is true for the two versions of GC12.

(a)

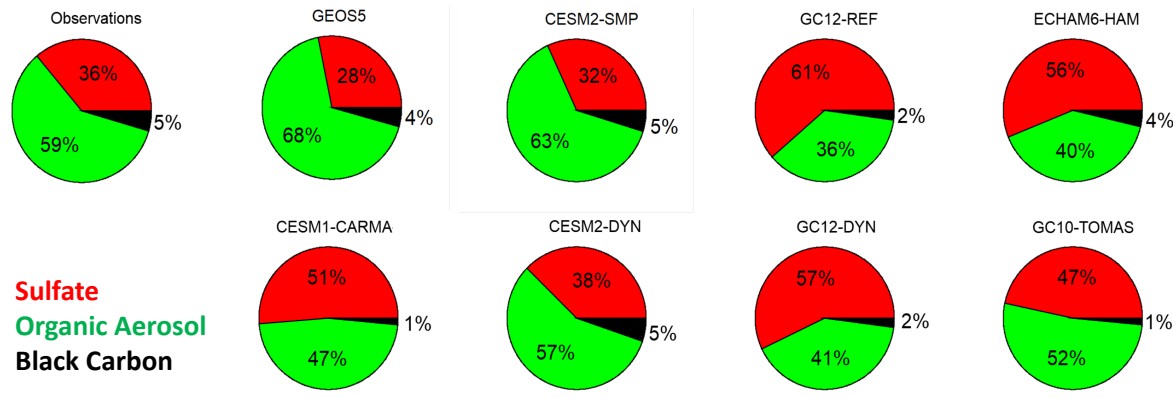


(b)

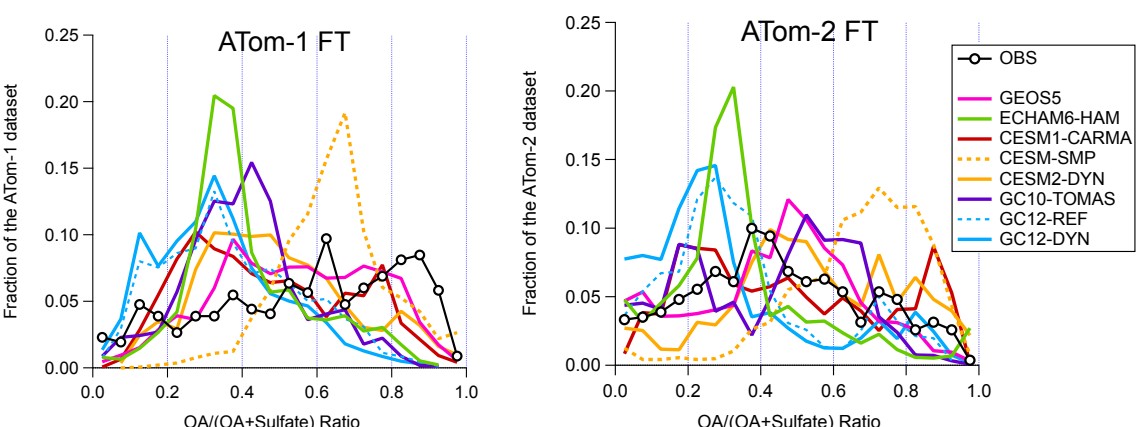


Figure 9: (a) Predicted and measured composition of submicron aerosols in the free
troposphere as a function of the submicron aerosol mass concentrations during ATom-1.
(b) Frequency distribution of observed and simulated ratio of organic to organic plus
sulfate aerosols in the free troposphere during ATom-1 and -2.



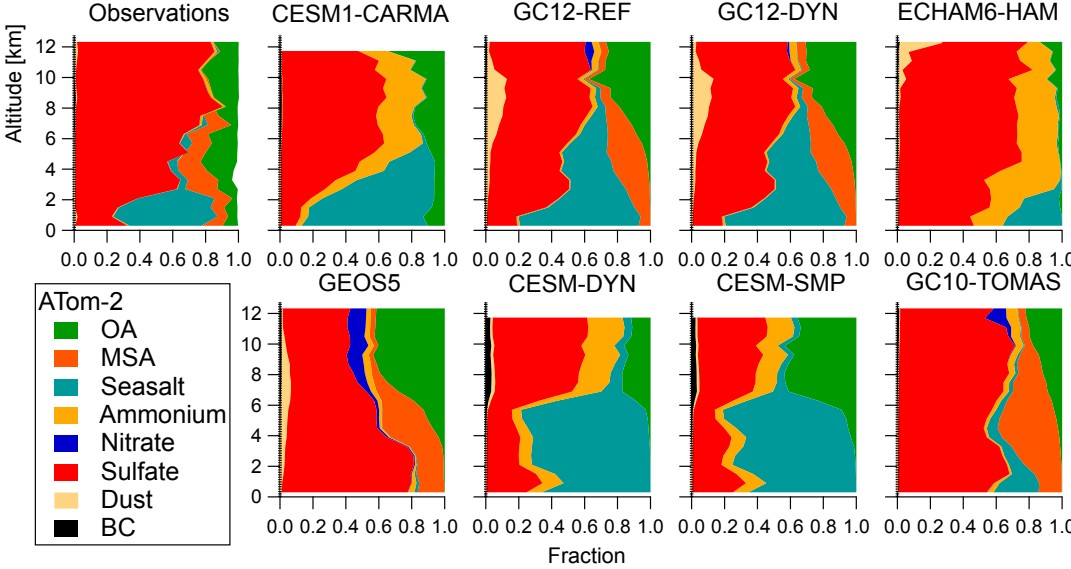


Figure 10: Comparison of measured and predicted composition of submicron aerosols as
a function of altitude over the remote Southern Ocean region during NH Winter (ATom-2).
For models that do not calculate ammonium in the aerosol (such as CESM1-CARMA,
CESM2-SMP, CESM2-DYN and ECHAM6-HAM), ammonium was estimated from the
sulfate mass assuming the formation of ammonium sulfate. Note that while the modeled
and measured submicron sea salt size ranges agree fairly well (Table 1), this is not quite
the case for dust. Given that the accumulation mode dust in the models presented
contains larger sizes than the AMS range (< 500 nm), it is expected for the modeled dust
concentration to be larger than measured.