# Peer review of "Characterization of organic aerosol across the global remote troposphere: A comparison of ATom measurements and global chemistry models"

_Atmospheric Chemistry and Physics, 2019_

## Referee Comment (RC1) · Anonymous Referee #1 · 9 Oct 2019

This is a well-written paper. I recommend accepting it, but clarifying as noted below:

Line 46-47: highest levels measured at what altitudes?

Line 75: You might consider adding Zhu et al., 2019 to the list of references here or in line 79.

Zhu, J., Penner, J. E., Yu, F., Sillman, S., Andreae, M., and Coe, H., 2019: Organic aerosol nucleation, climate and land use change: Decrease in radiative forcing, Nature Communications, 10, Article No. 423, https://www.nature.com/articles/s41467-

019-08407-7

Fig 1b: lines 1402-1407: Is the distribution shown for the AEROCOM models at the ground or at altitudes sampled by aircraft? What is meant by "distribution of studies" when referring to the models? (explain in caption, please, not just text)

Line 136: Is there something that distinguishes "ATom models" from other models? Strange terminology

Line 178-179: what fraction of hydrophilic organic material is incorporated into precipitation in GOCART? i.e. what is the Kappa value used in this model?

Line 237: add reference for CMIP6 global inventory

Line 426-429: The averaging procedure you used is not clear. If the values are < 3 ïĄ̧ detection limit, shouldn't you replace the value by zero (so as not to bias the average high)?

Line 550-551: Other than the reduction in spread of the AerocomII-sub models compared to full AeroCom II ensemble, this statement is not supported by comparing Fig S2 with Fig 3.

Line 558-559: you should plot these profiles on a linear scale. Its hard to judge how different the models are using a logarithmic scale.

Line 587-588: I would reference Fig S6 here, since it is on a linear scale. And you should change S7 to linear scale.

Line 766-769: what is meant by POA/OA being shifted rightward? Makes no sense to me.

---

## Referee Comment (RC2) · Anonymous Referee #2 · 27 Oct 2019

This is an interesting study that makes comprehensive use of a unique dataset (ATom) to evaluate a series of models. The multi-model approach is particularly valuable for pinpointing model deficiencies in these remote environments. The authors present a thorough series of comparisons, however the conclusions are not well supported. This is primarily due to the reliance on an analysis to separate POA from SOA in the measurements which is not very well justified. More work is needed to expand this analysis (see below for suggestions), or remove it and alter the text accordingly, before the manuscript would be acceptable for publication.

[Figure]

1. Figure 1: This figure is unclear and not sufficiently discussed in the main text. What does "distribution of studies" used as the x-axis of Figure 1b mean? – a more exact definition of what is plotted should be provided. In addition, the quantitative discussion of these AeroCom results in the abstract is unclear (line 37) – what does "factor of 400-1000" imply – that the spread of the means is of this range? This could more clearly be given as a percentage of the mean or median model, or as phrased in lines 100-103 as "model dispersion" in orders of magnitude. The manuscript does not fully discuss what is shown in Figure 1b.

2. Section 2.1 would benefit from a bit more discussion of the methodology in selecting these models and the differences in their configurations. Are they all standard configurations (i.e. as downloaded), including emissions used, if not why were different parameters chosen? The level of detail in the description of the various models is quite uneven –the authors should ensure that the same information is provided for all models. Finally, are simulations performed and sampled to match the spatial location of the ATom aircraft (with emissions and meteorology matched to the year of the measurements)?

3. The manuscript is missing any discussion of the role of POA treatment in these comparisons. It's not 100% clear from Section 2.1 (e.g. no info provided on POA for ECHAM-HAM, GC10-TOMAS, or any of the CESM configurations), but it appears that all of these simulations use non-volatile POA. A number of modeling studies have implemented a semi-volatile treatment of POA since Robinson et al. (2007). It seems like a major weakness to draw general conclusions on OA model performance when using a series of models which do not represent the semi-volatile nature of POA. It would be nice to see the authors add such a simulation to their suite, but if this proves impractical at this stage of the work, the manuscript should be altered considerably to acknowledge the gaps in the POA treatment and how this may have a substantial impact on the comparisons and conclusions drawn here. The lack of discussion of the simulation (and emissions) of POA also somewhat undercuts the discussion of Section

4.3. It's clear that models are underestimating the observed OM:OC during ATom, but if the models are over-estimating the POA to begin with (perhaps because it's all assumed to be non-volatile?) then this could be a compensating bias related solely to how POA is treated.

4. Some information on model configurations is missing that would be important for comparing model performance (could potentially be added to Table 1): what is the assumed OM:OC ratio, what are the global emission totals for key precursors (isoprene, monoterpenes, POA, etc.)?

5. The estimation of the POA fraction in Section 3.2 is not well supported. First, the manuscript is missing a discussion of the uncertainty on the fBB from PALMS (lines 340-342). Second, the numbers in Table S1 do not support the averages used in the text, for example EFs for urban sources range over an order of magnitude (0.16-15.4) and the authors appear to have simply averaged these values, which seems highlight inappropriate. The example provided by the authors of using a single ratio of BB from Andraea (2019) leading to a POA fraction of > 100% in African plumes also illustrates the inappropriate application of a single number. EFs range significantly with fuel type, combustion conditions, and location; use of any single value is likely to lead to uncertainties that would vastly outweigh the value of the analysis. A more appropriate approach might be to take a lower limit set of EFs and an upper limit set of EFs, and bracket the POA estimation using first one and then the other. Absent such an analysis, this POA estimate seems unreliable and the results of Section 4 are highly questionable. The analysis of Figure S9 seems to go in this direction, but the range in EFs in this Figure do not represent the full range of values shown in Table S1. Given that all the conclusions in Section 4.4 hinge on this analysis, perhaps the authors could expand this discussion: describe the range in fBB values, and then the calculated POA contributions (from FF and BB separately) estimated for all the ATom data. In order to explore the uncertainties in their methodology, the authors could also apply the same analysis to the model output of [BC] and assumed EFs (use first the same EFs as

used in the measurement analysis and then the EFs used in the model) to see how an estimated POA_model would compare to the simulated POA. This could pinpoint whether flaws in methodology for estimating POA or flaws in the model simulation of POA dominate.

6. Lines 813-821: Figure 10 seems interesting, but it feels like an aside. The details of how these models treat inorganics (including nitrate, ammonium, sea salt, and dust) and the relevant emissions, which would be necessary to understand these differences are not included in the manuscript. Thus, the authors should either eliminate this text in favor of a more focused discussion of the OA results (as suggested in point #5 above), or substantially enhance the model description section to include the relevant details.

Minor Comments

1. The mixed capitalization in the title is a bit odd.

2. Line 69: The authors might consider re-phrasing. The word "major" implies a larger role in RF than OC contributes in the AR5 assessment cited (i.e. GHG dominate the RF, and even amongst aerosols, the effect of OC is considerably less than the inorganics or BC).

3. Line 92: Hodzic et al (2016) do not use the "same field campaigns" – rather they use a subset of those previously analyzed by Heald et al. (2011) with some additional campaigns.

4. Lines 87-91: Pai et al., ACPD, 2019 provides a more recent evaluation of the standard GEOS-Chem model configurations (including comparisons with ATom) which should be discussed here and perhaps elsewhere in the manuscript, particularly as they do not see the same bias away from source that was highlighted in previous studies (Heald et al., 2011; Hodzic et al., 2016).

5. Table 1: Why are dust and seasalt sizes included here, and why are they listed in the sub-micron only? Dust and sea salt go well into the 10's of um in model simulations.

6. Lines 198-199 and 756-757: Marais et al. (2016) replace the isoprene VBS with their mechanism for isoprene SOA. Please clarify whether isoprene SOA in your simulations follows this or whether it includes both that from the VBS of Pye et al. (2010) as well as that produced using the mechanism of Marais et al. (2016), which might lead to double-counting of isoprene SOA.

7. Line 201: does "with the exception of the treatment of isoprene SOA" imply that photolytic removal does not apply to isoprene SOA in GC12-DYN?

8. Line 249-250: does this imply that CESM2-DYN uses the same SOA yield parameters, photolytic loss, and updated Henry's law constants as GC12-DYN? If not, please clarify which differ.

9. Line 275: "in a climatological way" is not defined here. Suggest remove as the later text describes how the model is sampled.

10. Figure S1 should be included in the main text given that it shows a central comparison of ATom-2 with the models

11. Section 3.1: The measurement description section should include the detection limits and uncertainties on the AMS data during ATom and how this might impact the comparisons. I noted that some of this is given in lines 415-423, but it seems like this belongs earlier in the measurement description section, or at least that the authors could refer the reader to this later discussion in their manuscript, so that they know it will be addressed.

12. Line 329: could the authors be more explicit? Does this imply that biomass burning OA from Africa is larger in size than typical?

13. Lines 330-339: what is the size range of the aerosols detected by the PALMS instrument?

14. Line 339: unclear. Why is the AMS transmission function applied to the PALMS data?

15. Line 343: Given that fBB from PALMS is a derived quantity and not a direct measurement the statement that the PALMS fBB "is more useful as a particle tracer" is a bit bold and requires a citation. Or the language should be softened to "may be more useful"

16. Lines 346-349: These sentences seem to conflate primary and biomass burning, which are not necessarily the same thing. If the implication is that the analysis assumes no SOA from biomass burning (such as suggested by Hodshire et al., 2019), that assumption should be stated explicitly here.

17. Line 382: what are the units on the POA? In units of carbon or was an OM:OC applied?

18. Line 391: EFs range orders of magnitude and these ranges in both the model and measurements are being compared. It's not clear that "no significant bias is apparent", they could easily differ by a factor of two on average – perhaps the authors rather mean something like "the ranges in values are consistent".

19. Section 3.3: The authors have focused on the means for their model-measurement comparisons. It would be useful to examine whether this is an appropriate metric – are the distributions skewed? Can the models capture the shape of the distribution? Might a comparison of medians in Section 4 provide different results?

20. Lines 435-436: could you provide the range of MBL heights and tropopause heights along the flight tracks used here?

21. Figure 2b: please provide either total number of points or percentages of total dataset for the categories should here.

22. Line 472: suggest inserting the word "likely" to "less polluted than ATom-1, likely due to a" since you haven't definitely compared emissions or source contributions.

23. Line 497-498: The statement "It should be noted.." is surprising. The authors haven't shown any analysis for this and Figures 2a and S1 clearly show elevated OA in

the North Pacific which seems likely associated with Asian source. Could the authors explain?

24. Line 537: The NMB in Table 2 for CESM1-CARMA is given as -33.2%, so the -20-30% range seems incorrect.

25. Table 2 indicates that the NMB for all models is positive for ATom-2. I didn't see this surprising result discussed in the text.

26. Line 539-541: This statement is incorrect as it only appears to apply to CESM. According to Table 2, while GC12-DYN is slightly less biased than GC12-REF for ATom-1, the reverse is true for ATom-2.

27. Section 4.4: The POA to OA ratio is derived, not directly measured (e.g. line 685)

28. Lines 726-728, and 737-738: given these statements why does Figure 8 not include a comparison of BC with and without in-cloud removal?

---

## Referee Comment (RC3) · Anonymous Referee #3 · 14 Nov 2019

Hodzic presents a comparison of a large set of global models and observations from the free troposphere with a focus on organic aerosol. I applaud the authors on this large undertaking in terms of number of models and synthesis of observations. The manuscript contains a large amount of information. My major comments are regarding the POA estimation and better illustrating the utility of DYN configuration (which may just be a bit burried).

Major comments:

[Figure]

1. Section 3.2 and POA estimation: Can the uncertainty in the POA estimates be quantified, ideally with error bars (e.g. in Fig 6 vertical profiles or Fig 8 POA)? Some ideas to consider regarding the POA estimation:

a. There are anthropogenic POA sources that do not have significant amounts of BC. See, for example, Figure 2 of Reff et al. (https://pubs.acs.org/doi/10.1021/es802930x) which indicates large emissions of dust associated with anthropogenic activity (road dust, construction dust) that have high OC relative to BC. While fossil fuel-BC may be a general proxy for anthropogenic activity, is the ratio 1.55 reflective of that general behavior? What does the EPA NEI indicate as FFratio including all anthropogenic PM sources (e.g. Fig 2 of Reff et al.)?

b. Can you plot data (obs or model) as a function of f(BB) influenced to get an intercept at f(BB)=0 and f(BB)=1 for comparison to FFratio and BBratio specified in text?

c. For the calculation of observed POA, should there be small amounts of OA associated with sea spray and dust? You have sea salt and dust observations in Figure 2b and elsewhere that could be used to estimate that POA.

d. Would it be better to label the estimate from equation 1 as "combustion POA"? I would assume models are also more specifically combustion POA?

e. Is non-differential removal of BC and POA appropriate if POA is semivolatile? Consider that near a fire, concentrations could be high enough for IVOCs to be partitioned in to the particle. As the airmass is diluted, POA will decrease more rapidly than a conserved tracer. This may explain why the larger BBratio gives POA >100% (line 378).

f. For model estimates of POA, is hydrophilic OC considered POA or SOA? Can models just label the hydrophilic OC as "SOA" and get the right properties for endpoints (health, climate) of interest?

g. Did you consider using a CTM to verify the method in equation 1 or back calculate
the model effective FFratio? Does it reproduce the model POA (hydrophobic OC)?

h. BC is chemically aged. Do measurement techniques measure BC effectively at all atmospheric lifetimes?

i. Line 650: The observation-method is not necessarily an upper-bound limit on the fraction of POA as it does not consider OA emitted in sea spray, dust, and may not consider all anthropogenic forms of POA. Consider rephrasing and/or demonstrating it is a limit by adding error bars by using more conservative (higher) FFratios.

2. Line 355: Can you elaborate on why the PMF doesn't work to separate SOA and POA? When (what timescale, location, altitude, other factor) does the PMF stop working?

3. Perhaps the strengths of DYN could be better isolated/highlighted (e.g. what lifetimes, OA regimes, POA levels, etc does DYN perform better for?). The abstract statement: "concept of a more dynamic OA...with enhanced removal of primary OA, and a stronger production of secondary OA in global models needed to provide a better agreement with observations" could use more support.

Evaluation seems mixed: Figure 8, indicates GC12 and CESM1-CARMA have reasonable POA (while CESM2-DYN has overestimated POA). Line 803 indicates DYN evaluates better, but that effect seems marginal or secondary to other issues in Figure 9 b at least for GC. RMSE in Table 2 also tends to increase when going from base to DYN treatment in CESM2 and GC12. Figure 7 improvements seem mixed.

Minor comments:

1. Abstract states "OA predictions from AeroCom Phase II...span a factor of 400-1000..." should that be the inter-model variability spans a factor of 400-1000 or the concentrations predicted span that range?

2. Did all models assume nonvolatile POA?

3. Do OA/OC ratios include consideration of S and N in the form of organosulfates and organonitrates?

4. At times, it was a bit confusing if the AeroCom-II ensemble referred to circa 2014 models or the actual AeroCom II results paired with measurements. On line 103, the "AeroCom results" could be clarified as "AeroCom models" or "models used in AeroCom." Line 277 might be better as "monthly average results of 28 global models" instead of "results of 28 global models."

5. Figure 1: Is "author affiliations" supposed to be in the caption?

6. Line 163: While marine production of OA may be smaller than continental production, it's possible marine production contributes more to concentrations over the oceans than it's global production rate would suggest. I recommend rewording sentence. What concentrations of SOA are predicted for oceanic isoprene sources?

7. How does averaging data allow below detection values to be used?

8. Line 605: qualify the OA/OC of 1.4 as "fossil fuel" combustion as biomass combustion tends to have OA/OC of 1.7.

---

## Author Comment (AC1) · 4 Feb 2020

**Responses to the Reviews of "Characterization of Organic Aerosol across the Global Remote Troposphere: A comparison of ATom measurements and global chemistry models"**

**Anonymous Referee #1**

This is a well-written paper. I recommend accepting it, but clarifying as noted below:

We thank the reviewer for interesting suggestions. We have modified the manuscript to address all of his/her concerns.

R1.1) Line 46-47: highest levels measured at what altitudes?

A1.1) The highest levels were measured in the lower troposphere (below 4km). This is now explained in the revised manuscript:

"*OA concentrations have a strong seasonal and zonal variability, with the highest levels measured in the lower troposphere in the summer and over the regions influenced by the biomass burning from Africa (up to 10 ug sm$^{-3}$).*"

R1.2) Line 75: You might consider adding Zhu et al., 2019 to the list of references here or in line 79. Zhu, J., Penner, J. E., Yu, F., Sillman, S., Andreae, M., and Coe, H., 2019: Organic aerosol nucleation, climate and land use change: Decrease in radiative forcing, Nature Communications, 10, Article No. 423, https://www.nature.com/articles/s41467-019-08407-7.

A1.2) The suggested reference has been added.

R1.3) Fig 1b: lines 1402-1407: Is the distribution shown for the AEROCOM models at the ground or at altitudes sampled by aircraft? What is meant by "distribution of studies" when referring to the models? (explain in caption, please, not just text)

A1.3) We have revised the figure axis label, legends, and figure caption to better describe the data shown on the figure. For the models, what we are showing is the geographical distribution of the institutions hosting/running the GCMs that participated in the AEROCOM-II comparison, which, very much like the measurements, have a very strong bias towards the Northern Hemisphere (NH). While GCMs certainly try to cover the global troposphere, both the bias in constraining measurements and funding sources will lead to more optimization of these models for the mid-latitude NH. The updated figure and caption are shown below:

[Figure]

*Caption Fig. 2b: "(b, right) Geographical distribution of institutions at which the AeroCom-II models were ran/developed (based on the affiliation of all authors) and of the field measurements included in two major literature overview studies (Zhang et al., 2007; Heald et al., 2011) for the OA ground and aircraft AMS as a function of latitude. For the aircraft campaigns, the average latitude for the full deployment was taken."*

R1.4) Line 136: Is there something that distinguishes "ATom models" from other models? Strange terminology.

A1.4) The term "Atom models" is shorthand used to refer to current models (current as of beginning of 2019) that have been ran for the ATom field project as explained in the text:

*"ATom measurements were compared with results of eight global models that simulated the time period of the ATom-1 and 2 campaigns (August 2016 and February 2017), using reanalysis meteorology (and a spin-up time of at least six to twelve months). These are referred hereafter as ATom models [..]"*

We have updated the title of sections 2.1 and 2.2 to read "*ATom model simulations*" and "*AeroCom-II model climatology*".

R1.5) Line 178-179: what fraction of hydrophilic organic material is incorporated into precipitation in GOCART? i.e. what is the Kappa value used in this model?

A1.5) The GOCART model emits 50% of POA in hydrophilic and 50% in hydrophobic mode. The model allows a conversion from a hydrophobic to hydrophilic mode with an e-folding time of 2.5 days. All SOA from biogenic, anthropogenic, and biomass burning sources are treated as hydrophilic. The hydrophilic OA is removed by large-scale and convective warm clouds, while the hydrophobic OA is removed by ice clouds. The hydrophilic particles undergo hygroscopic growth according to the equilibrium parameterization of Gerber (1985). This is now explained in the manuscript as:

*"The primary emitted OC and SOA are separated into hydrophobic (50%) and hydrophilic (50%) species, with a 2.5 days e-folding time conversion from hydrophobic to hydrophilic organic particles. All SOAs from biogenic, anthropogenic, and biomass burning sources are treated as hydrophilic particles. Both types of organic particles are dry deposited. The hydrophilic OA is removed by large-scale and convective warm clouds, while hydrophobic OA is removed by ice clouds. The hydrophilic particles undergo hygroscopic growth according to the equilibrium parameterization of Gerber (1985)."*

*Gerber, H. E.: Relative-humidity parameterization of the Navy Aerosol Model (NAM), Tech. Rep. NRL Report 8956, Naval Research Laboratory, 1985.*

R1.6) Line 237: add reference for CMIP6 global inventory.

A1.6) The reference has been added:

*"The two simulations with the GEOS-Chem 12.0.1 global chemistry model (Bey et al., 2001) use emissions based on CMIP6 global inventory (Hoesly et al., 2018; Feng et al., 2019) with regional improvements for anthropogenic sources,.."*

*Hoesly, R. M., Smith, S. J., Feng, L., Klimont, Z., Janssens-Maenhout, G., Pitkanen, T., Seibert, J. J., Vu, L., Andres, R. J., Bolt, R. M., Bond, T. C., Dawidowski, L., Kholod, N., Kurokawa, J.-I., Li, M., Liu, L., Lu, Z., Moura, M. C. P., O'Rourke, P. R., and Zhang, Q.: Historical (1750–2014) anthropogenic emissions of reactive gases and aerosols from the Community Emissions Data System (CEDS), Geosci. Model Dev., 11, 369–408, https://doi.org/10.5194/gmd-11-369-2018, 2018.*

*Feng, L., Smith, S. J., Braun, C., Crippa, M., Gidden, M. J., Hoesly, R., Klimont, Z., van Marle, M., van den Berg, M., and van der Werf, G. R.: Gridded Emissions for CMIP6, Geosci. Model Dev. Discuss., https://doi.org/10.5194/gmd-2019-195, in review, 2019.*

R1.7) Line 426-429: The averaging procedure you used is not clear. If the values are < 3 sigma detection limit, shouldn't you replace the value by zero (so as not to bias the average high)?

A1.7) No! This is an important misconception among some modelers. The data have the correct statistical behavior, i.e. the average of a period of zero concentrations is near zero, as is verified frequently by measuring filtered ambient air in flight. Thus both negative and positive values below DL need to be retained in the dataset. Concentrations cannot be negative, but measurements can be thought of as the sum of concentrations and statistical noise, and can be negative when the real concentrations are zero or very low. A bias would be created if we removed measurements below <0 (or below DL), which we are not doing and generally caution against. This is already explained in the manuscript (L428-430 of the ACPD version), but we have expanded this discussion to make it clearer:

*"Note that a large fraction of the 1-minute OA values in the remote free troposphere were below the local 3σ detection limit. The data of periods of zero concentration (sampling ambient air through a particle filter) do average to zero. Some negative measurements are present, and this is normal for measurements of very low concentrations in the presence of instrumental noise. Averaging of longer periods, as done for the figures in this paper, reduces the detection limit. We therefore caution future data users that the reported data should be averaged as needed, as replacing below-detection limit (or negative) values by other values introduces biases on averages."*

We have also included an additional figure into the SI that evaluates possible biases in the fractional data by filtering the data based on an independent measurement (the NOAA aerosol volume measurement on ATom, Brock et al 2019) and included some additional discussion in the main text:

*"For fractional ratio analysis, measurements were averaged to 5-minute time resolution to reduce the noise in the ratios due to noise in the denominator. The results are not very sensitive to the 5-minute averaging (compared to 1-minute) as shown in Figure S12 for OA to sulfate ratios. The same figure also illustrates that excluding ratios affected by negative concentrations (the non-bracketed case, overall these are about 15% of the dataset) does not really affect the fractional distribution, with the variance between the two cases diminishing as the averaging interval increases. To further confirm that there is no inherent bias in the fractional products regardless of the treatment of low concentration values, an additional sensitivity analysis was performed where data was filtered by an independent measurement proxy for aerosol mass, the aerosol volume measured in ATom (Brock et al, 2019). Using a range of value that encompasses the regime where the AMS calculated volume to aerosol measured volume exhibited increased noise (Guo et al, 2019), no systematic bias was found (Figure S13), with variations of about 10% in fractional volume for different filtering conditions."*

[Figure]

***Caption Fig. S13****: Exploring the impact of thresholding the 5-min averaged data by a minimum detectable aerosol volume in the PM1 range (from the NOAA SD product, see Guo et al 2019 for details) when computing the the OA/(OA+SO4) distributions in Figure 9.*

R1.8) Line 550-551: Other than the reduction in spread of the AerocomII-sub models compared to full AeroCom II ensemble, this statement is not supported by comparing Fig S2 with Fig 3.

A1.8) We have revised this text to make the point clearer:

"*This reduction in the ensemble spread may partially be explained by a smaller size of the ATom model ensemble (see Fig. S2), which also includes models with a more up-to-date OA representation. In order to explore this point further, results for a subset of AeroCom II models (using earlier versions of models in the ATom ensemble) show only a slight reduction (~10%) in the model spread, with however some regional differences i.e. an improved agreement with observations in the MBL, but an increase in the model bias and spread in the LS (Figure S2). Thus, model improvement for the more recent models appears to be the main reason for the reduced spread.*"

R1.9) Line 558-559: you should plot these profiles on a linear scale. It is hard to judge how different the models are using a logarithmic scale.

A1.9) Given that the modeled and observed values span more than one order of magnitude we have used the log scale to visually facilitate the model/obs comparisons but also to allow us to keep the same x-axis span for various regions. We have also added a new supplementary figure (similar to figure 4) using a linear scale in the updated manuscript, as Figure S5 (shown below). This is now explained in the revised manuscript:

[Figure]

*"Note that the use of a wide logarithmic scale (to be able to span all the observations) may make the observed differences appear small, although they often reach factors of 2-10 and larger (Figure S5 shows the same results on a linear scale)."*

*Caption Fig. S5: Similar to figure 4 but on a linear scale.*

R1.10) Line 587-588: I would reference Fig S6 here, since it is on a linear scale. And you should change S7 to linear scale.

A1.10) We have revised the paper to include both the log (Figure S7a) and linear scale figures (Figure S7b) as part of Figure S7.

a)

[Figure]

b)

[Figure]

*Caption Fig. S7: As Figure S6 for ATom-2 shown both on a logarithmic (a) and linear (b) scales.*

R1.11) Line 766-769: what is meant by POA/OA being shifted rightward? Makes no sense to me.

A1.11) The predicted POA/OA ratio in GC12-REF is overpredicted compared to measurements in Figure 7 which is consistent with the results shown in Figure 8 for GC12-REF that have the right amount of POA and underpredict total OA.

This is now clarified in the manuscript as:

*"It should be noted that these results are consistent with the POA/OA frequency distribution shown in Figure 7 (the POA/OA ratio predicted by GC12-REF is larger than the measured ratio, which is consistent with the fact that POA is about the right amount, and OA is underpredicted in Figure 8)."*

---

## Author Comment (AC2) · 4 Feb 2020

**Responses to the Reviews of "Characterization of Organic Aerosol across the Global Remote Troposphere: A comparison of ATom measurements and global chemistry models"**

**Anonymous Referee #2**

General Comments) This is an interesting study that makes comprehensive use of a unique dataset (ATom) to evaluate a series of models. The multi-model approach is particularly valuable for pinpointing model deficiencies in these remote environments. The authors present a thorough series of comparisons, however the conclusions are not well supported. This is primarily due to the reliance on an analysis to separate POA from SOA in the measurements which is not very well justified. More work is needed to expand this analysis (see below for suggestions), or remove it and alter the text accordingly, before the manuscript would be acceptable for publication.

We thank the reviewer for valuable suggestions. We hope that we have addressed all the concerns in a satisfactory manner. In particular, we have improved the POA analysis and associated discussions in the revised manuscript. Additional simulations have been performed with the GEOS-Chem model to document the sensitivity of our results to the simulated non-volatile vs. semi-volatile properties of POA.

R2.1) Figure 1: This figure is unclear and not sufficiently discussed in the main text. What does "distribution of studies" used as the x-axis of Figure 1b mean? – a more exact definition of what is plotted should be provided. In addition, the quantitative discussion of these AeroCom results in the abstract is unclear (line 37) – what does "factor of 400-1000" imply – that the spread of the means is of this range? This could more clearly be given as a percentage of the mean or median model, or as phrased in lines 100-103 as "model dispersion" in orders of magnitude. The manuscript does not fully discuss what is shown in Figure 1b.

A2.1) We already addressed the points about Figure 1b in A1.3 since the first reviewer had a similar comment, and refer reviewer 2 to the changes discussed there.

The factor of 400-1000 refers to the results in Figure 1a, which are described in L103-107 of the ACPD version, as well as in the caption of Figure 1. We have reworded the main text for clarity as:

*"Our own analyses of the AeroCom-II results shown in Figure 1a indicate that model dispersion (quantified as the ratio of the average concentration of the highest model to that of the lowest one, in each region) increases not only with altitude but also with distance from the northern mid-latitude source (and data-rich) regions. The model spread is a factor of 10-20 in the free troposphere between the equator and northern mid-latitudes, and increases to a factor of 200-800 over the Southern Ocean and near the tropopause."*

The caption of Figure 1a has been revised to read:

*"Figure 1: (a, left) The ratio between the average OA concentrations of the highest to the lowest models (for each region) as predicted among 28 global chemistry transport models participating in the AeroCom phase II intercomparison study (Tsigaridis et al. 2014)."*

And the abstract has been revised to read:

*"OA predictions from AeroCom Phase II global models span two to three orders-of-magnitude"*

R2.2) Section 2.1 would benefit from a bit more discussion of the methodology in selecting these models and the differences in their configurations. Are they all standard configurations (i.e. as downloaded), including emissions used, if not why were different parameters chosen? The level of detail in the description of the various models is quite uneven –the authors should ensure that the same information is provided for all models. Finally, are simulations performed and sampled to match the spatial location of the ATom aircraft (with emissions and meteorology matched to the year of the measurements)?

A2.2) The considered models span a range of complexity in terms of aerosols parameterizations, and some of the models have several OA schemes or aerosol modules (like CESM or GEOS-Chem). For each model we have referenced the publication that describes the baseline configuration, and the modifications that have been used in the runs included here. In the revised manuscript, we have made clear when a standard configuration is being used e.g. for the GEOS-Chem GC12-REF configuration:

*"Note that this GEOS-Chem REF simulation is similar to the version 12 default "complex option" which includes non-volatile POA and semi-volatile SOA (semi-volatile POA is an optional switch within this version used in Pai et al. 2020)."*

*Pai, S. J., Heald, C. L., Pierce, J. R., Farina, S. C., Marais, E. A., Jimenez, J. L., Campuzano-Jost, P., Nault, B. A., Middlebrook, A. M., Coe, H., Shilling, J. E., Bahreini, R., Dingle, J. H., and Vu, K.: An evaluation of global organic aerosol schemes using airborne observations, Atmos. Chem. Phys. Discuss., https://doi.org/10.5194/acp-2019-331, in review, 2019.*

We have also provided more details for some models and made sure that the description includes information on the emissions, aerosol module (composition, size representation), OA formation and removals.

The following description has been added for ECHAM6-HAM:

*"Aerosol particles are removed by dry and wet deposition. The wet deposition includes the below cloud scavenging by rain and in-cloud cloud scavenging for large-scale and convective systems (Croft et al., 2010)."*

*Croft, B., Lohmann, U., Martin, R. V., Stier, P., Wurzler, S., Feichter, J., Hoose, C., Heikkilä, U., van Donkelaar, A., and Ferrachat, S.: Influences of in-cloud aerosol scavenging parameterizations on aerosol concentrations and wet deposition in ECHAM5-HAM, Atmos. Chem. Phys., 10, 1511–1543, https://doi.org/10.5194/acp-10-1511-2010, 2010.*

Removal has been better described for GEOS-Chem:

*"The removal of gases and aerosols are treated similar to the GEOS-Chem 12.0.1 model (GC12-REF, see above)."*

The following was added for CESM2:

*"Simulations based on the CESM2.0 Earth system model use the standard version of the Whole Atmosphere Community Climate Model (WACCM6, Gettelman et al., 2019, Emmons et al., 2019)."*

ATom model simulations were performed with the emissions and meteorology matching the year of the measurements. This is now better explained in the manuscript:

*"ATom measurements were compared with results of eight global models that simulated the time period of the ATom-1 and 2 campaigns (August 2016 and February 2017), using the emissions and reanalysis meteorology corresponding to this period (and a spin-up time of at least six to twelve months)."*

In addition, a column has been added to Table 1 specifying the meteorological reanalysis used for each model.

R2.3) The manuscript is missing any discussion of the role of POA treatment in these comparisons. It's not 100% clear from Section 2.1 (e.g. no info provided on POA for ECHAM-HAM, GC10-TOMAS, or any of the CESM configurations), but it appears that all of these simulations use non-volatile POA. A number of modeling studies have implemented a semi-volatile treatment of POA since Robinson et al. (2007). It seems like a major weakness to draw general conclusions on OA model performance when using a series of models which do not represent the semi-volatile nature of POA. It would be nice to see the authors add such a simulation to their suite, but if this proves impractical at this stage of the work, the manuscript should be altered considerably to acknowledge the gaps in the POA treatment and how this may have a substantial impact on the comparisons and

conclusions drawn here. The lack of discussion of the simulation (and emissions) of POA also somewhat undercuts the discussion of Section 4.3. It's clear that models are underestimating the observed OM:OC during ATom, but if the models are over-estimating the POA to begin with (perhaps because it's all assumed to be non-volatile?) then this could be a compensating bias related solely to how POA is treated.

A2.3) As suggested by the reviewer we have clarified in the revised manuscript that models used in this study only include non-volatile POA parameterizations. Please see section 2.1:

*"In all models POA is treated as a non-volatile directly emitted species. In most models (see below) the primary emitted organic aerosol is artificially aged to transition between hydrophobic to hydrophilic POA."*

This non-volatile treatment of POA in the models is consistent with the way the estimated POA has been derived from the ATom measurements. Indeed, the estimated POA is calculated from the POA/BC ratios representative of the ambient air values close to the emission sources, after most evaporation has occurred, but before substantial chemistry of POA has taken place. As a consequence, the estimated POA can be approximately considered to be non-volatile. As discussed in response A2.24, the model and measurement emission ratios are not significantly different. Therefore the comparison with the non-volatile POA representation from models is more appropriate than a comparison with a semi-volatile POA representation. This is now more clearly explained in the manuscript in section 4.4:

*"POA concentrations were estimated from the BC measurements by using an emission ratio appropriate to the airmass origin (biomass burning vs. anthropogenic), and using the f(BB) mass fraction from the PALMS single particle instrument (see Section 3.2). By using the POA/BC ratio at the source regions after most evaporation, but before POA chemical degradation has taken place, we implicitly assume POA to be chemically inert, while in reality it can slowly be lost to the gas-phase by heterogeneous chemistry (e.g. George and Abbatt, 2010; Palm et al., 2018). Thus, the observation-based method provides an upper limit to the fraction of POA. The model/measurement comparison is only shown for the CESM and GEOS-Chem model variants, as other participating models do not separate or did not report their POA and SOA fractions. In all simulations, POA was treated as a chemically inert directly emitted primary aerosol species that only undergoes transport, transformation from hydrophobic to hydrophilic state with ageing (1-2 days typically), coagulation, and dry and wet deposition. Importantly, the treatment of POA as non-volatile (rather than semi-volatile) in models is fully consistent with the assumptions for POA estimation from the measurements."*

And in the conclusion:

*"The non-volatile POA treatment in models is consistent with the assumption of inert POA particles used to estimate POA from the measurements, and cannot explain the model bias. Indeed, sensitivity simulations with semi-volatile POA lead to a much larger model bias for OA in the upper troposphere and remote regions."*

In addition, we have performed sensitivity simulations to estimate the effect of the non-volatile vs. semi-volatile POA assumption in the models on POA predictions. We have performed an additional simulation (GC12-REF-SVPOA) for ATom-1 based on GC12-REF, in which the non-volatile treatment of POA has been replaced by the semi-volatile POA parameterization based on Pye and Seinfeld, 2010 and using a two-product reversible partitioning model. This is a similar model configuration as used in Pai et al. (2020) under "the complex scheme" (though different emissions were used between their study and here). The comparison of POA vertical profiles between GC12-REF (non-volatile) and GC12-REF-SVPOA (semi-volatile) over various regions is shown in the figure below. The comparison indicates that the POA concentrations are larger in most regions when the semi-volatile POA parameterization is used.

[Figure]

***Caption Fig. S16****: Sensitivity simulations to estimate the importance of the non-volatile vs. semi-volatile POA treatment in GEOS-Chem. The semi-volatile POA in GC12-REF-SVPOA (GC12-DYN-SVPOA) model configuration should be directly compared with the corresponding GC12-REF (GC12-DYN) non-volatile POA.*

This is now discussed in the revised manuscript:

*"Finally, we have examined whether the non-volatile treatment of POA in models could lead to these unrealistically high POA fractions in the remote regions. Figure S16 shows a comparison of POA vertical profiles as predicted by the GC12-REF simulations that use non-volatile POA and a sensitivity simulation GC12-REF-SVPOA that uses semi-volatile POA similar to the standard treatment in GEOS-Chem as described in Pai et al. (2020). Note, however, that Pai et al. (2020) included marine POA emissions, used different reanalysis meteorology, and a different model version (12.1.1 rather than 12.0.1 here), so their resulting comparisons to ATom measurements are somewhat different than found here for GC12-REF-SVPOA. The comparison indicates that the POA concentrations increase substantially in most regions when the semi-volatile POA parameterization is used. These results suggest that non-volatile treatment of POA is not responsible of the model bias."*

R2.4) Some information on model configurations is missing that would be important for comparing model performance (could potentially be added to Table 1): what is the assumed OM:OC ratio, what are the global emission totals for key precursors (isoprene, monoterpenes, POA, etc.)?

A2.4) Information on OA/OC ratios was already provided in the ACPD manuscript. Please see the description: *"OA/OC of 1.4 is used in ECHAM6-HAM, whereas 1.8 is used in GEOS5 and GC10-TOMAS simulations for both POA and SOA. Other models calculated directly SOA concentrations without applying this conversion (CESM1-CARMA, CESM2-SMP, CESM2-DYN, GC12-REF and GC12-DYN), but for POA used the ratio of 1.8 (CESM1-CARMA, CESM2-DYN) and 2.1 (GC12-REF and GC12-DYN). Most of the AeroCom-II models used the ratio of 1.4 for all primary and secondary OA (Tsigaridis et al., 2014)."* This information is also shown again in Figure 5 when comparing with the measurements.

As suggested by the reviewer we have added the OA/OC ratios also to Table 1.

We do not have the total amount of precursors saved for all models, so that information has not been added. However, we reference the emission inventories that are used for each model.

R2.5) The estimation of the POA fraction in Section 3.2 is not well supported. First, the manuscript is missing a discussion of the uncertainty on the fBB from PALMS (lines 340-342). Second, the numbers in Table S1 do not support the averages used in the text, for example EFs for urban sources range over an order of magnitude (0.16-15.4) and the authors appear to have simply averaged these values, which seems highlight inappropriate. The example provided by the authors of using a single ratio of BB from Andraea (2019) leading to a POA fraction of > 100% in African plumes also illustrates the inappropriate application of a single number. EFs range significantly with fuel type, combustion conditions, and location; use of any single value is likely to lead to

uncertainties that would vastly outweigh the value of the analysis. A more appropriate approach might be to take a lower limit set of EFs and an upper limit set of EFs, and bracket the POA estimation using first one and then the other. Absent such an analysis, this POA estimate seems unreliable and the results of Section 4 are highly questionable. The analysis of Figure S9 seems to go in this direction, but the range in EFs in this Figure do not represent the full range of values shown in Table S1. Given that all the conclusions in Section 4.4 hinge on this analysis, perhaps the authors could expand this discussion: describe the range in fBB values, and then the calculated POA contributions (from FF and BB separately) estimated for all the ATom data.

In order to explore the uncertainties in their methodology, the authors could also apply the same analysis to the model output of [BC] and assumed EFs (use first the same EFs as used in the measurement analysis and then the EFs used in the model) to see how an estimated POA_model would compare to the simulated POA. This could pinpoint whether flaws in methodology for estimating POA or flaws in the model simulation of POA dominate.

A2.5) The range quoted by the reviewer for urban sources is not correct. The ratio of 15.4 is for rural agricultural biomass burning, not for urban sources. In addition, we only used in our average the ratios for mixed urban air, as discussed in response A2.3, while the ratios for emission sources (e.g. individual cars) were only shown to support their consistency with the mixed urban air ratios. We have clarified Table S1 (shown below) to make clear which values are used in our averages (marked now in **bold**) and which are presented only for reference, and which apply to urban vs. BB sources (shown now in *italic*). In reality, the range of measured ratios for urban pollution is 0.5-2.4, and the uncertainty due to this effect is minor. In fact Figure S9 in the ACPD version already showed a sensitivity study with the urban ratio varying between 0.5 and 3, and showed that the effect of this ratio on the plots is minuscule, especially when compared to the model-measurement disagreement.

We have now clarified in the text how the averages for urban sources were calculated:

*"Based on Table S1 data, we assume POA to be co-emitted with BC for anthropogenic fossil fuel / urban region POA (herein called FF$_{ratio}$ for simplicity, even though much of it is non-fossil, Zotter et al., 2014; Hayes et al., 2015) at a ratio of 1.5 (average of all urban ambient air studies that report POA and BC for best intercomparability to the ATom dataset; including all urban studies results in a very similar number, 1.48)."*

Furthermore, upon revisiting Andreae (2019) review for these responses, we noticed that using an OA/OC ratio of 1.8 for his data as we have done for all other studies compiled in Table S1 was incorrect, since he based his review on a universal value of OA/OC of 1.6 for biomass burning sources (see Section 2.1 in that review), which results in a small correction to the BB$_{ratio}$ to 11.8 (instead of 13.5). Hence we have updated Figure 7 as well as Figure S8-S10 (all shown below) to

reflect this change (which has minimal impact on f(POA)), and have modified the text in the manuscript accordingly:

*"For biomass burning sources, we use a value of POA/BC = 11.8 (BB$_{ratio}$), based on the average of the recent review by Andreae (2019), which included over 200 previous determinations for a variety of fuels and burning conditions (since Andreae (2019) used and OA/OC ratio of 1.6 in his work, we have used that value to calculate POA/BC; we note that this is different from the 1.8 OA/OC ratio used for other studies listed in Table S1)."*

We have also slightly revised the range of FF$_{ratio}$ and BB$_{ratio}$ that we consider in the sensitivity analysis shown in Figure S9. We cover a range of 0.5-2.4 for FF$_{ratio}$, consistent with the discussion above (and add one additional scenario). For the range of BB$_{ratio}$ , we are using the lower and upper uncertainty ranges (in both OC and EC emissions) from Andreae (2019) for the major contributors to global BB (2-60), which also covers all the suggested averages for the individual sources (except peat) as well as the range of BB emissions used in the models (Table S2). Aerosol emissions from peat are a clear outlier, but their global contribution is small (about 5%) and, as a recent analysis shows, the peat sources with very large BB$_{ratio}$, are very localized (Watson et al, 2019), so they mostly contribute during the height of the South East Asian Fire season (September to October, Reddington et al, 2014), hence outside the sampling period for ATom-1 and 2.

*Watson, J. G., Cao, J., Chen, L. W. A., Wang, Q., Tian, J., Wang, X., Gronstal, S., Ho, S. S. H., Watts, A. C. and Chow, J. C.: Gaseous, PM2.5 Mass, and Speciated Emission Factors from Laboratory Chamber Peat Combustion, Atmos. Chem. Phys. Discuss., 1–39, doi:10.5194/acp-2019-456, 2019.*

It should be clear now that the sensitivity study in Figure S9 *does represent the full range of the literature emission ratios* shown in Table S1. And that illustrates the robustness of the POA results: even with the most extreme assumptions for the emission ratios, the POA/OA distribution changes little. The key is that BC is very low in most of the remote troposphere, and thus there are no realistic ratios of POA/BC that could possibly produce POA concentrations similar to those in most models. We have added Figure S11a to the SI (shown below) to illustrate the skewness of the BC/OA distribution.

Regarding the uncertainty in f(BB)$_{PALMS}$, while it should be clear that any uncertainty in this factor will have only a limited impact on f(POA), we have conducted an extra sensitivity study with the uncertainty estimated by the PALMS team (+/-5%), and have added Figure S11b to the SI.

We have also revised Figure S10, which explores the impact of very low OA values on the f(POA) distribution. In addition to showing the sensitivity of f(POA) to the choice of averaging interval (which reduces the percentage of points below detection limit) we also explore the impact of capping POA to OA (e.g. not allowing the estimated POA to be larger than OA). This new analysis

shows that not capping POA results in very similar f(POA) profiles, with the exception of f(POA)=1. The 10-20% fractions calculated for the standard, capped case are actually a combination of data close to sources where POA estimated from the measurements was indeed larger than OA (and which in Figure S10 would show up at values >1) and cases where BC and hence POA was close to zero (BC<0.1 ng sm$^{-3}$) but OA was negative due to noise. As expected, this effect is somewhat less apparent at longer averaging times (and more apparent for ATom-2, where there was a higher fraction of very low OA values). Since the non-capped case underestimates f(POA)=1, by not including the data close to sources, using the capped data is clearly better. However, due to the limitations in our ability to estimate POA when both BC and OA are very low our analysis likely overrepresents the amount of POA found in ATom. We have modified the discussion in Section 4.4 to reflect this:

*"The differences are so large that they are pretty insensitive to details of the POA estimation method from the measurements, mostly because for the vast majority of the ATom track BC/OA ratios were extremely low and hence the exact magnitude of the multiplicative factor is secondary to the estimation of POA (Figure S11). As Figure S9 illustrates, the choice of FFratio has very little impact on the overall distribution of POA. On the other hand, while the BBratio does impact the overall distribution of POA, it mostly affects the points in the vicinity of the large Atlantic plumes. Since the POA/BC ratio in those plumes is fairly low, (see Section 3.2), using a very large BBratio mostly leads to an increase of the fraction of the points where POA > 100%. While the large range of published BB$_{ratio}$ for different sources precludes a more accurate estimation by our method, for the purposes of the comparison with the model results we emphasize that even using the largest BB$_{ratio}$, f(SOA) is still significantly larger in the ATom dataset that in any of the models.*

*Additional sensitivity tests were performed to investigate the impact of noisy data and uncertainties of f(BB) on the estimation of POA. Figure S11 clearly shows that the impact of a misattribution of the aerosol type by the stated PALMS uncertainty (Froyd et al, 2019) is completely negligible. Figure S10 details how the choice of averaging interval (with longer averaging times reducing both the fraction of OA measurements under the DL and below zero) impact the distribution of POA. Overall, no large changes are observed for averaging times >5 min, and hence a 5 min averaging interval was used for the analysis in Figure 7. Figure S10 also illustrates how capping the histogram impacts the POA distribution. To capture the most realistic f(POA) distribution, the data in Fig 7 was capped at the extremes (so f$_{(POA)}$<0 is taken as f$_{(POA)}$=0, and f$_{(POA)}$>1 is taken as f$_{(POA)}$=1). As Fig S10 shows, data with f$_{(POA)}$<0 is almost exclusively due to very small (and always positive, since BC cannot go negative) POA values being divided by small, negative noise in total OA, and hence treating that fraction of the histogram as essentially f$_{(POA)}$~0 is justified. On the other end of the distribution, data where POA is larger than OA is mostly due to our average BB$_{ratio}$ being larger than the one encountered in most of the BB plumes in ATom. Choosing a lower BB$_{ratio}$, as Fig S9b and S9d illustrate, leads to f(POA)>1 basically trending to zero, confirming our interpretation. This is a limitation of the dataset, and it does not seem appropriate to remove*

*these points, since some fraction are likely dominated by POA. However, it shows that the POA estimation, especially for this part of the distribution likely overstates the importance of POA."*

It is not possible to apply the measurement methodology to model outputs, as none of the models track separately BC from various emission sectors. In any case, we have now clearly shown that the measurement-based estimates are very robust against a wide range of assumptions.

*Table S1: POA/BC ratios determined in previous field and laboratory emission studies. Studies that reported well constrained urban non-BB POA based on AMS PMF determinations (highlighted in bold) were averaged to determine the value used for (POA/BC)$_{anthro}$ . Studies that reported (POA/BC)$_{BB}$ are shown in italics. For the average of (POA/BC)$_{BB}$ the weighted average reported by Andreae, 2019 was used.*

| Source | Technique | Type of emissions | POA/BC ratio (OA measured) | POA/BC ratio (OC measured, OA/OC of 1.8 used) |
|---|---|---|---|---|
| Zhang et al. 2005 | AMS PCA for POA EC from TOCA | Urban background | **1.41** | |
| Szidat et al. 2006 | 14C source apportionment for EC and OC | Urban mobile sources Residential burning | | 2.65 *11.3* |
| Ban-Weiss et al. 2008 | OC: Filters (TOA) Aethalometer and filters for BC | Mobile sources: Light Duty Vehicles Diesel | | 2.5 1.3 |
| Aiken et al. 2009 | AMS PMF for POA, SP2 for BC | Urban background | **0.8** | |
| Christian et al. 2010 | TOT EC/OC analyzer | Cooking Stoves Trash Burning Brick Klinn Charcoal Klinn AG Burn | | *6.3* *7.75* *0.27* *78* *200* |
| Chirico et al. 2010 | AMS PMF for POA SP2 for BC | Tailpipe emissions, gas vehicle | 0.16-0.3 | |

| Reference | Method | Source/type | | |
|---|---|---|---|---|
| Minguillon et al. 2011 | 14C source apportionment for EC and OC, combined with AMS PMF | Urban backg. Rural backg. Biomass burning | *15.4* | 1.7 4 |
| Huang et al. 2013 | AMS PMF for POA, SP2 for BC | Urban backg. winter Urban backg. summer | **0.82** **1.27** | |
| Hayes et al. 2013 | AMS PMF for POA, SP2 for BC | Urban background | **1.82** (average) 1.51 (more diesel influenced) | |
| Crippa et al. 2013 | AMS PMF for POA, Aethalometer for BC | Urban mobile sources Cooking aerosol Residential burning | **0.5 (ave)** **0.5 (ave)** 3.4 (ave) | |
| Huang et al. 2015 | Offline AMS and TOT OC/EC analyzer, ME2 analysis | Traffic Cooking BB | **0.5** **2.5** 11 | |
| Zhang et al. 2015 | 14C source apportionment for EC and OC | Fossil fuel, coal burning Residential burning | | 1.6 *8.5* |
| Hu et al. 2016 | AMS PMF for POA, SP2 for BC | Urban Background | **1.4** | |
| Kim et al. 2018 | AMS PMF for POA, SP2 for BC | Urban background (70% HOA, 30% COA) | **2.2** | |
| Whatore et al. 2017 | TOT EC/OC analyzer | African traditional stoves | | *4.8* |
| Nault et al. 2018 | AMS PMF for POA, SP2 for BC | Urban background | **2.38** | |
| Chen et al. 2018 | AMS PMF for POA, SP2 for BC | BB urban BB rural | *6.25* *5* | |
| Chirico et al. 2011 | AMS OA SP2 for BC | Tunnel mobile emissions | 0.4 | |

| Kim et al. 2017 | AMS PMF for POA, SP2 for BC | Total urban POA (40% BB, 27% HOA, 33% COA) | 3.2 | |
|---|---|---|---|---|
| Andreae, 2019 | Review (OA/OC of 1.6 used per the methodology of the review) | Savanna
Tropical forest
Temperate forest
Boreal forest
Peat
AG
Dung
Biofuel
Charcoal
Average (this work) | | *9.1*
*13.8*
*31.7*
*22*
*227*
*18.7*
*9.9*
*52.6*
*13*
***11.8*** |

[Figure]

***Caption Fig. 7:*** *Frequency distribution of observed and simulated ratio of POA to total OA in the free troposphere during ATom-1 and ATom-2 as computed by the GC12-, CESM2-, and CESM1-CARMA models.*

[Figure]

*Figure S8: POA/OA distributions (free troposphere only) from Figure 7 shown as cumulative distributions (CDF). Note that for the OA/BC ratios observed for ATom specifically, the green curves in Fig S9b and S9d (BB$_{ratio}$=2) are closer to the real distribution.*

[Figure]

***Caption Fig. S9:*** *Sensitivity of the overall measured POA/OA distribution to different estimates of POA/BC ratios for both urban and BB sources covering the range of values shown in Table S1 and S2, both for the frequency and cumulative frequency distribution (left/right) and ATom-1 and 2 (top/bottom). Note that for the choice of BBratio ranges, we used the range (within uncertainties) for the main global BB contributors and excluded one clear outlier, peat. This is justified since peat is a small source, mostly localized to SE Asia, and the main emissions of peat BB aerosol are outside the sampling periods of ATom-1 and 2 (Reddington et al, 2014).*

[Figure]

***Caption Fig. S10****: Exploring the impact of OA data below detection limit (DL) by increasing the averaging interval on the POA/OA distributions in Figure 7 for ATom 1 and 2 (a 5 min averaging interval was used throughout the analysis discussed in Section 4.4). Also shown is the comparison of a capped (so $f_{(POA)}$=0 includes $f_{(POA)}$<0, and $f_{(POA)}$=1 includes $f_{(POA)}$>1) vs. an unconstrained histogram, for the same set of averaging intervals. In the manuscript, 5-minute averaging (capped) is used*

[Figure]

***Caption Fig. S11:*** *(left) Distribution of BC/OA ratios that are used as the basis of the estimation of $f_{(POA)}$ for all ATom deployments, shown using different averaging intervals (right) Effect of the 5% uncertainty in the f(BB) reported by the PALMS instrument on the estimation of $f_{(POA)}$, using both bracketed and not bracketed data (cf. Figure S10).*

Added SI reference:

*Reddington, C. L., Yoshioka, M., Balasubramanian, R., Ridley, D., Toh, Y. Y., Arnold, S. R. and Spracklen, D. V.: Contribution of vegetation and peat fires to particulate air pollution in Southeast Asia, Environ. Res. Lett., 9(9), doi:10.1088/1748-9326/9/9/094006, 2014.*

R2.6) Lines 813-821: Figure 10 seems interesting, but it feels like an aside. The details of how these models treat inorganics (including nitrate, ammonium, sea salt, and dust) and the relevant emissions, which would be necessary to understand these differences are not included in the manuscript. Thus, the authors should either eliminate this text in favor of a more focused discussion of the OA results (as suggested in point #5 above), or substantially enhance the model description section to include the relevant details.

A2.6) While this paper focuses on OA, it is still of interest to document the relative importance of OA and other species, and how these vary substantially across different models. Several papers from our groups and others have been published that address some of those components, and others are in preparation. We believe it is still of broad interest to keep this figure to provide context for the OA results. We have added further explanations to the text with suitable references for ATom analyses and modeling of the other chemical components:

*"The discrepancies between the observed and predicted composition of submicron aerosol over remote regions can be quite large for other constituents as well. Figure 10 shows the comparison of measured and predicted composition of the submicron aerosol over the Southern Ocean (during the NH winter) where the disagreement in simulated sea salt, nitrates, ammonium, and MSA often exceeds the contribution of OA. While the observations show a more uniform distribution of non-marine aerosol with higher values in the mid and upper troposphere, respectively, most models tend to simulate highest fractions of OA (and sulfate) towards the tropopause. This may also be explained by the uncertainties in modeled wet removal of aerosol that has been discussed above. Specific studies have discussed and continue to investigate the ATom measurements and simulations of different components in more detail, including black carbon (Katich et al., 2018; Ditas et al., 2019), MSA (Hodshire et al., 2019), sulfate-nitrate-ammonium (Nault et al., 2019), and sea salt (Yu et al, 2019; Bian et al., 2019; Murphy et al., 2019)."*

For consistency with the treatment in Figure 2b, we have also included both the modeled and measured submicron dust to Figure 10. The measurements only reflect the low end of the dust distribution (< 500 nm), and do not fully match the size range of the model-reported submicron dust (as shown in Table 1). Hence it is expected that observations will have lower dust concentrations than the models.

[Figure]

We have updated the Figure caption to read:

*Caption Fig. 10: "[..] Note that while the modeled and measured submicron sea-salt size ranges agree fairly well (Table 1), this is not quite the case for dust. Given that the accumulation mode dust in the models presented contains larger sizes than the AMS range (< 500 nm), it is expected for the modeled dust concentration to be larger than measured."*

*Bian, H., et al. (2019), Observationally constrained analysis of sea salt aerosol in the marine atmosphere 3, Atmos. Chem. Phys., doi:10.5194/acp-2019-18.*

*Ditas, J., et al. (2018), Strong impact of wildfires on the abundance and aging of black carbon in the lowermost stratosphere, Proc. Natl. Acad. Sci., 811595-11603, doi:10.1073/pnas.1806868115.*

*Hodshire, A., et al. (2019), The potential role of methanesulfonic acid (MSA) in aerosol formation and growth and the associated radiative forcings, Atmos. Chem. Phys., 19, 3137-3160, doi:10.5194/acp-19-3137-2019.*

*Katich, J., et al. (2018), Strong Contrast in Remote Black Carbon Aerosol Loadings Between the Atlantic and Pacific Basins, J. Geophys. Res., 123, 13,386-13,395, doi:10.1029/2018JD029206.*

*Nault, B., et al. (2019), Global Observations of Ammonium Balance and pH Indicate More Acidic Conditions and More Liquid Aerosols than Current Models Predict, Abstract A52C-08, presented at 2019 Fall Meeting, AGU, San Francisco, CA, 9-13 Dec.*

*Murphy, D. M., Froyd, K. D., Bian, H., Brock, C. A., Dibb, J. E., DiGangi, J. P., Diskin, G., Dollner, M., Kupc, A., Scheuer, E. M., Schill, G. P., Weinzierl, B., Williamson, C. J., and Yu, P.: The distribution of sea-salt aerosol in the global troposphere, Atmos. Chem. Phys., 19, 4093-4104, https://doi.org/10.5194/acp-19-4093-2019, 2019.*

*Yu, P., Froyd, K. D., Portmann, R. W., Toon, O. B., Freitas, S. R., Bardeen, C. G., Brock, C., Fan, T., Gao, R.-S., Katich, J. M., Kupc, A., Liu, S., Maloney, C., Murphy, D. M., Rosenlof, K. H., Schill, G., Schwarz, J. P. and Williamson, C.: Efficient In-Cloud Removal of Aerosols by Deep Convection, Geophys. Res. Lett., 46(2), 1061–1069, doi:10.1029/2018GL080544, 2019.*

**Minor Comments**

R2.7) The mixed capitalization in the title is a bit odd.

A2.7) The mixed capitalization has been removed: "*Characterization of organic aerosol across the global remote troposphere: A comparison of ATom measurements and global chemistry models*".

The mixed capitalization for the mission name (ATom, Atmospheric Tomography mission) is in accordance with the official mission acronym and description: https://espo.nasa.gov/atom

R2.8) Line 69: The authors might consider rephrasing. The word "major" implies a larger role in RF than OC contributes in the AR5 assessment cited (i.e. GHG dominate the RF, and even amongst aerosols, the effect of OC is considerably less than the inorganics or BC).

A2.8) We agree with the reviewer. The sentence has been changed to read:

"*They are associated with adverse health effects (Mauderly and Chow, 2008, Shiraiwa et al., 2017) and contribute radiative forcing in the climate system (Boucher et al., 2013).*"

R2.9) Line 92: Hodzic et al (2016) do not use the "same field campaigns" – rather they use a subset of those previously analyzed by Heald et al. (2011) with some additional campaigns.

A2.9) We agree with the reviewer's comment, and have updated the text to read:

"*For a subset of 9 recent aircraft campaigns, Hodzic et al. (2016) showed that OA is likely a more dynamic system than represented in chemistry-climate models, with both stronger production and stronger removals.*"

R2.10) Lines 87-91: Pai et al., ACPD, 2019 provides a more recent evaluation of the standard GEOS-Chem model configurations (including comparisons with ATom) which should be discussed here and perhaps elsewhere in the manuscript, particularly as they do not see the same bias away from source that was highlighted in previous studies (Heald et al., 2011; Hodzic et al., 2016).

A2.10) This paper has not yet been accepted as of this writing, so we refrained from discussing it in detail, based on previous guidance from journal editors and reviewers. Now that it is accepted, we have added a reference in the revised paper, see response to R2.3.

R2.11) Table 1: Why are dust and seasalt sizes included here, and why are they listed in the sub-micron only? Dust and sea salt go well into the 10's of um in model simulations.

A2.11) We agree that sea-salt and dust are mostly present in the coarse mode, but they do have a tail in the submicron mode. It is their contribution to submicron aerosols only that is included in Figures 2 and 10, to provide a complete representation of all the chemical components present in submicron particles. Figure 10 in the submitted manuscript did not include dust, as explained in the response A2.6 we have added it in the revised version. We have also adjusted the caption accordingly, as documented in that response.

R2.12) Lines 198-199 and 756-757: Marais et al. (2016) replace the isoprene VBS with their mechanism for isoprene SOA. Please clarify whether isoprene SOA in your simulations follows this or whether it includes both that from the VBS of Pye et al. (2010) as well as that produced using the mechanism of Marais et al. (2016), which might lead to double-counting of isoprene SOA.

A2.12) For isoprene, there is no double-counting as the VBS has been replaced by the parameterization from Marais et al. (2016).

This is now more clearly explained:

"*The first configuration (called hereafter GC12-REF) includes the default (http://wiki.seas.harvard.edu/geos-chem/index.php) representation of SOA formation based on Marais et al. (2016) for isoprene-derived SOA, and on the volatility basis set (VBS) of Pye et al. (2010) for all other precursors.*"

This is also clarified lines 756-757:

"*It should be noted that in both cases, isoprene-SOA is formed in aqueous aerosols following Marais et al. (2016).*"

R2.13) Line 201: does "with the exception of the treatment of isoprene SOA" imply that photolytic removal does not apply to isoprene SOA in GC12-DYN?

A2.13) That is correct. This has been clarified in the revised manuscript: "*As in Hodzic et al. (2016) the GC12-DYN model version includes updated VBS SOA parameterization, updated dry and wet removal of organic vapors, and photolytic removal of SOA (except for isoprene-SOA).*"

R2.14) Line 249-250: does this imply that CESM2-DYN uses the same SOA yield parameters, photolytic loss, and updated Henry's law constants as GC12-DYN? If not, please clarify which differ.

A2.14) The treatment is similar in both models for the most part following the parameterization of Hodzic et al., 2016, at the exception of i) the isoprene-SOA formation (GC12-DYN used Marais et al., 2016); ii) the low-NOx yields (in CESM2-DYN only low-NOx yields are used). This is now more clearly explained in the manuscript:

"*This is a similar SOA scheme as used in GC12-DYN (with differences in the treatment of isoprene-SOA based on Marais et al. 2016 in GC12-DYN, and the use of both low- and high-NOx VBS yields in GC12-DYN).*"

R2.15) Line 275: "in a climatological way" is not defined here. Suggest remove as the later text describes how the model is sampled.

A2.15) We have removed this text as suggested by the reviewer.

R2.16) Figure S1 should be included in the main text given that it shows a central comparison of ATom-2 with the models.

A2.16) We respectfully disagree. Figure S1 shows that the trends discussed for ATom-1 hold for ATom-2 as well.

R2.17) Section 3.1: The measurement description section should include the detection limits and uncertainties on the AMS data during ATom and how this might impact the comparisons. I noted that some of this is given in lines 415-423, but it seems like this belongs earlier in the measurement description section, or at least that the authors could refer the reader to this later discussion in their manuscript, so that they know it will be addressed.

A2.17) These items are discussed at length in the references provided (Schroeder et al, 2018; Nault et al, 2018; Jimenez et al., 2019), but we agree with the reviewer that a brief summary and referral

to Section 3.3 would improve the readability of the manuscript. Hence we have added the following to Section 3.1:

*"AMS data was acquired at 1 Hz time resolution and independently processed and reported at both 1 s and 60 s time resolutions (Jimenez et al., 2019a). The later product, with more robust peak fitting at low concentrations was exclusively used as the primary dataset in this work. Detection limits at different time resolutions/geographical bins relevant to this study are discussed in Section 3.3. The overall 2σ accuracies of the AMS measurement (38% for OA, 34% for sulfate and other inorganics) are discussed in Bahreini et al. (2008) and Jimenez et al. (2019b)."*

*Bahreini, R., Ervens, B., Middlebrook, A. M., Warneke, C., de Gouw, J. A., DeCarlo, P. F., Jimenez, J. L., Brock, C. A., Neuman, J. A., Ryerson, T. B., Stark, H., Atlas, E., Brioude, J., Fried, A., Holloway, J. S., Peischl, J., Richter, D., Walega, J., Weibring, P., Wollny, A. G. and Fehsenfeld, F. C.: Organic aerosol formation in urban and industrial plumes near Houston and Dallas, Texas, J. Geophys. Res., 114, D00F16, doi:10.1029/2008JD011493, 2009.*

*Jimenez, J.L., P. Campuzano-Jost, D.A. Day, B.A. Nault, D.J. Price, and J.C. Schroder. ATom: L2 Measurements from CU High-Resolution Aerosol Mass Spectrometer (HR-AMS). ORNL DAAC, Oak Ridge, Tennessee, USA. https://doi.org/10.3334/ORNLDAAC/1716, 2019a.*

*Jimenez, J.L., et al.: Evaluating the Consistency of All Submicron Aerosol Mass Measurements (Total and Speciated) in the Atmospheric Tomography Mission (ATom), Abstract A31A-08, presented at 2019 Fall Meeting, AGU, San Francisco, CA, 9-13 Dec., 2019b.*

R2.18) Line 329: could the authors be more explicit? Does this imply that biomass burning OA from Africa is larger in size than typical?

A2.18) Not at all, this just refers to the fact that a linear regression is quite sensitive to high points, and that on average the African BB plumes have 10x higher concentrations than the data outside of them. As discussed in Brock et al. (2019) and Jimenezet al. (2019), for the measurements there is no systematic bias apparent in the comparisons with the particle sizing instruments in this range.

For the models discussed, both GC10-TOMAS and CESM1-CARMA do show about 15%-20% contribution of coarse aerosols contribution to OA in the BB plumes, and removing those improves the correlation for GC10-TOMAS (0.97). This is not the case for the standard version of CESM1-CARMA, since without the convective fix it also shows a substantial contribution of large aerosols in the UT.

We have modified the text to clarify this:

*"(Slopes for ATom-1 linear regressions: CESM-1CARMA:0.91, GC10-TOMAS: 0.94, ECHAM6-HAM 1.00) mostly influenced by the high concentration points in the biomass plumes off Africa that have a large effect on the linear regressions, since they are about 10 times larger than the bulk of the dataset)"*

R2.19) Lines 330-339: what is the size range of the aerosols detected by the PALMS instrument?

A2.19) The PALMS instrument reports mass products in the range 100-3000 nm geometric based on the NOAA size distribution data (Brock et al, 2019, Froyd et al, 2019). All the PALMS data included in this work has been computed to match the size range of the AMS (so $D_{aero}$ 40...1250 nm, see Knote et al., 2011, and Jimenez et al., 2019) using the measured density. Hence the PALMS data reported here is consistent with the AMS data, with the possible exception of (less frequent) particle growth events in the upper troposphere where a significant mass fraction is below the optical detection limit of the PALMS (roughly 100-150 nm $D_{geo}$, see Froyd et al, 2019). The text in the manuscript has been modified to explain this more clearly:

*"For all PALMS data used in this work (biomass burning fraction and dust) the AMS transmission function was applied to ensure that both instruments were characterizing approximately the same particle size range."*

R2.20) Line 339: unclear. Why is the AMS transmission function applied to the PALMS data?

A2.20) See the response to the previous comment (A2.19).

R2.21) Line 343: Given that fBB from PALMS is a derived quantity and not a direct measurement the statement that the PALMS fBB "is more useful as a particle tracer" is a bit bold and requires a citation. Or the language should be softened to "may be more useful".

A2.21) We have revised this text to further explain what we meant and why this is the best choice for our analyses, with the available dataset. Note that the next sentence in the manuscript (L343-345 in the ACPD version, not referred to by the reviewer, but very important for this choice) provides an additional, and likely more important reason for the usefulness of this parameter for our purposes. The revised text reads:

*"This parameter correlates quite well with other gas-phase BB tracers, and is more useful as a particle tracer since its lifetime follows that of the particles. Importantly, it is not impacted by the long lifetimes of the gas-phase tracers (e.g. 9 months for $CH_3CN$) and unrelated removal processes (e.g. ocean uptake for $CH_3CN$ and HCN) that result in highly variable backgrounds. Hence $f(BB)_{PALMS}$ has a much higher contrast ratio and linearity for particle BB impacts, compared to the available gas-phase tracers in the ATom dataset. An airmass was classified as non-BB influenced*

*when f(BB)$_{PALMS}$ was lower than 0.30 (Hudson et al, 2004) as shown in Figure 2b.  f(BB)$_{PALMS}$ was also used to assess the impact of POA on the total OA burden (next section); note that no thresholding was applied in that case."*

R2.22) Lines 346-349: These sentences seem to conflate primary and biomass burning, which are not necessarily the same thing. If the implication is that the analysis assumes no SOA from biomass burning (such as suggested by Hodshire et al., 2019), that assumption should be stated explicitly here.

A2.22) The order of these 3 sentences was confusing and we have reorder them to first explain how we separate BB and non-BB airmasses, and then how we calculate the POA fraction in OA. See the revised manuscript:

*"An airmass was classified as non-BB influenced when f(BB)$_{PALMS}$ was lower than 0.30 (Hudson et al, 2004) as shown in Figure 2b. For both ATom-1 and 2, about 76% of measurements were classified as not influenced by biomass burning. f(BB)$_{PALMS}$ was also used to assess the impact of POA on the total OA burden (next section); note that no thresholding was applied in that case."*

Furthermore, *f(BB)$_{PALMS}$* is an important variable on that estimation process, as explained in the next section. At this point in the text, this has no bearing on the SOA formation ability of BB sources.

Later, in the next section, we do state (L375-377): *"We note the measured total OA/BC of ~3.5 (conservatively assuming that all OA is POA) observed on both ATom missions for the large African sourced BB plumes over the Equatorial Atlantic"* Indeed we are assuming here that **on those strong African BB plumes measured near the source region** all OA is POA. However, there is no explicit assumption applied that all BB OA is POA in our POA estimation method. Depending on the plumes encountered in the global atmosphere and their OA/BC ratio, some of their OA can be classified as SOA by our method.

The key point is that, since **the main result is that POA is surprisingly low compared to models, we are trying to make conservative assumptions that maximize POA**. In this way the measurement-based estimate cannot be criticized as being biased low and providing too low POA.

R2.23) Line 382: what are the units on the POA? In units of carbon or was an OM:OC applied?

A2.23) We only use OA (Organic Aerosol), POA (Primary Organic Aerosol) and SOA (Secondary Organic Aerosol) in units of μg sm$^{-3}$, as stated in Section 3.1, in this manuscript. By definition these include carbon and any other elements that are part of the organic molecules constituting OA. As described in Section 4 and Table 1, some of the older models still use OC, but this is a less

useful metric (mostly left over from a time in which only OC could be measured) that we have tried to avoid for the discussion of concentrations in this work.

Importantly, as described above, to derive the $FF_{ratio}$ from Table S1 we have relied exclusively on studies that actually report OA, and not OC, since the uncertainty in those determinations is substantially larger and also less applicable to the instrument payload on ATom. We have modified the text to clarify this point:

*"Based on Table S1 data, we assume POA to be co-emitted with BC for anthropogenic fossil fuel / urban region POA (herein called $FF_{ratio}$ for simplicity, even though much of it is non-fossil, Zotter et al., 2014; Hayes et al., 2015) at a ratio of 1.55 (average of all studies that report POA and BC)"*

R2.24) Line 391: EFs range orders of magnitude and these ranges in both the model and measurements are being compared. It's not clear that "no significant bias is apparent", they could easily differ by a factor of two on average – perhaps the authors rather mean something like "the ranges in values are consistent".

A2.24) We have calculated the averages for measurements (3.5-4 for residential, and 1-1.8 for traffic) and the emission inventories (4.6-5.9 for residential, and 1.1-1.4 for traffic) to confirm that they are similar. We have reworded this text to address this comment as:

*"The averages and ranges of the measurement and model ratios are similar, and thus no significant model bias on the ratios is apparent."*

R2.25) Section 3.3: The authors have focused on the means for their model-measurement comparisons. It would be useful to examine whether this is an appropriate metric – are the distributions skewed? Can the models capture the shape of the distribution? Might a comparison of medians in Section 4 provide different results?

A2.25) For the model-measurement comparisons of the AeroCom-II and ATom model ensembles, we have compared both the box plots for various regions using medians (Figure 3) and the vertical profiles using the means of each ensemble (Figure 4). The results of those analyses are consistent, and show a factor of 2-3 overestimation by the AeroCom-II model ensemble of the measured OA in various regions.

The plot below shows the comparison of medians for the observed and predicted OA in various regions for (a) the AeroCom-II model ensemble and for (b) the ATom model ensemble. These plots are to be compared with Figure 5 in the manuscript that showed OA mean concentrations. Here again, the comparison suggests that using medians results in a slightly lower values for all

datasets (as expected), but does not change the conclusions of the model-measurement comparisons. We have added the plots below to the SI as Figure S18.

This is now explained in the revised manuscript:

"*We note that using the ensemble median OA profiles instead of ensemble mean OA profiles (as shown in Figure 5 and S7) results in a slightly lower values of OA but does not change the conclusions of the model-measurement comparisons. (Figure S18).*"

For the evaluation of the individual models the statistics are shown in Table 2. As suggested by the reviewer we have in addition compared distribution plots of OA mass concentrations for the observations and various models.

[Figure]

a)

[Figure]

b)

c)

[Figure]

*Caption Fig. S18: Comparison of OA median vertical profiles as measured during ATom-1 and predicted by the (a) AeroCom-II model ensemble and (b) ATom model ensemble . Panels (c) and (d) show the same for ATom-2, respectively (similar to figure 5 in the paper that compares OA average profiles).*

R2.26) Lines 435-436: could you provide the range of MBL heights and tropopause heights along the flight tracks used here?

A2.26) We have added the NCEP reanalysis values of the PBL and tropopause heights to the Supp. Info. as Figure S17. GEOS5 values are very similar:

[Figure]

[Figure]

*Caption Fig. S17: (top) tropopause heights from the NCEP reanalysis at each Lat/Long flown for ATom-1 (blue) and ATom-2 (red). (bottom) Planetary boundary layer (PBL) heights obtained in the same way. Values from the GEOS-5 model are very similar to these. ATom-1 and 2 flight tracks are included in grey for context.*

The DC-8 ceiling is about 13 km (42000 ft, in practice 39000-41000 ft was the maximum altitude for most flights), which means that we only sampled the stratosphere at latitudes higher than 30 degrees. Based on these data, we modified the manuscript to document this as:

*"The tropopause height varied during ATom between 8 and 16.5 km; given the DC-8 ceiling (42 kft, 12.8 km) the stratosphere was only sampled at latitudes higher than 30 degrees in both hemispheres. The MBL height varied between up to 1.5 km in the mid-latitudes, ~1 km in the tropics, and sometimes <150 m (lowest DC-8 altitude) for some of the sampling in the polar troposphere."*

R2.27) Figure 2b: please provide either total number of points or percentages of total dataset for the categories should here.

A2.27) As requested by the reviewer we have calculated the percentages of data in each category. This information has been added to the caption of Figure 2b:

*"In ATom-1, BB-only represents 24% of the data, clean MBL 8%, clean FT 57% and clean UT 12%, whereas in ATom-2 BB-only represents 24%, clean MBL 9%, clean FT 53%, clean UT 15%."*

R2.28) Line 472: suggest inserting the word "likely" to "less polluted than ATom-1, likely due to a" since you haven't definitely compared emissions or source contributions.

A2.28) We have modified the text as requested.

R2.29) Line 497-498: The statement "It should be noted.." is surprising. The authors haven't shown any analysis for this and Figures 2a and S1 clearly show elevated OA in the North Pacific which seems likely associated with Asian source. Could the authors explain?

A2.29) This may have been unclear as originally written. We were trying to say that we did not see large extended plumes. But we do agree that the elevated OA in the North Pacific is likely associated with the Asian outflow. To clarify this point, we have modified the sentence and referenced the corresponding figures in the revised manuscript:

*"It should be noted that Asian pollution was likely an important contributor to the North Pacific Basin, especially between 2 and 6 km, in both ATom deployments (see figures 2a and S1)."*

R2.30) Line 537: The NMB in Table 2 for CESM1-CARMA is given as -33.2%, so the -20-30% range seems incorrect.

A2.30) The range has been corrected. See the updated text in the response A2.31 below.

R2.31) Table 2 indicates that the NMB for all models is positive for ATom-2. I didn't see this surprising result discussed in the text.

A2.31) See response in A2.32.

R2.32) Line 539-541: This statement is incorrect as it only appears to apply to CESM. According to Table 2, while GC12-DYN is slightly less biased than GC12-REF for ATom1, the reverse is true for ATom-2.

A2.32) To address reviewer's comments (2.31 and 2.32) we have separated the discussion into ATom-1 and ATom-2 (NH summer and NH winter) periods. The revised manuscript has been updated to read:

*"During the NH summer (ATom-1), models using the VBS parameterization from Pye et al. (2010) tend to underpredict the OA concentrations by 43% for GC12-REF and 33% for CESM1-CARMA for ATom-1, most likely due to the excessive evaporation of the formed SOA in remote regions and low yields for anthropogenic SOA (Schroder et al., 2018; Shah et al., 2019). Models using the VBS parameterization from Hodzic et al. (2016) (CESM2-DYN and GC12-DYN) where OA is less volatile and also OA yields are corrected for wall losses show an improved agreement with observations especially for CESM2-DYN (with NMB of ~5%), and to a lesser extent for GC12-DYN (NMB of ~33%). During the NH winter (ATom-2) characterized by a lower production of SOA, both VBS approaches lead to an overestimation of the predicted OA. This is likely caused by excessively high levels of primary emitted OA as discussed in section 4.4."*

R2.33) Section 4.4: The POA to OA ratio is derived, not directly measured (e.g. line 685).

A2.33) We believe that this is already very clear to a reader of this section, since the method is described in detail. However, to reduce possible confusion we have changed the text at this location to read:

*"Most models fail to reproduce the overwhelming dominance of SOA that is inferred from the measurements during ATom-1, while the discrepancies are less severe during NH winter (ATom-2)."*

R2.34) Lines 726-728, and 737-738: given these statements why does Figure 8 not include a comparison of BC with and without in-cloud removal?

A2.34) The reason the BC was not shown is because we do not have CESM1-CARMA results for BC for both simulations with and without in-cloud removal improvements. Figure for BC is shown below and includes only CESM1-CARMA simulations with in-cloud removal improvements (as described in Table 1 of the manuscript). Thus, this figure has not been included in the main section of the paper, but we have added it to the SI (Figure S15).

[Figure]

*Caption Fig. S15: Measured and predicted BC concentrations during ATom-1 as a function of the number of days since the air mass was processed through convection.*

---

## Author Comment (AC3) · 4 Feb 2020

**Responses to the Reviews of "Characterization of Organic Aerosol across the Global Remote Troposphere: A comparison of ATom measurements and global chemistry models"**

**Anonymous Referee #3**

Hodzic presents a comparison of a large set of global models and observations from the free troposphere with a focus on organic aerosol. I applaud the authors on this large undertaking in terms of number of models and synthesis of observations. The manuscript contains a large amount of information. My major comments are regarding the POA estimation and better illustrating the utility of DYN configuration (which may just be a bit buried).

We thank the reviewer for the encouraging evaluation. We have modified the manuscript to address his/her concerns on the POA and DYN model configurations and other topics.

**Major comments:**
R3.1). Section 3.2 and POA estimation: Can the uncertainty in the POA estimates be quantified, ideally with error bars (e.g. in Fig 6 vertical profiles or Fig 8 POA)? Some ideas to consider regarding the POA estimation:

The uncertainty on the POA estimation is dominated by the choice of the biomass burning POA/BC emission ratio. We have documented both in the SI and in A2.5 at length that except for the pure BB points the uncertainty in the distribution of f(POA) is rather small (and for the purer BB points it's biased high toward more POA). Regarding OA, the following figure shows the uncertainty range from Figure S9 in concentration space:

[Figure]

So in absolute terms, the uncertainty in the estimation of total POA is about 20. But this does barely affect the actual fractions shown in Figure 6, since only a very small number of points actually contributes to the larger concentrations. For Figure 8 the uncertainty is likely larger, but will mostly affect the absolute numbers, not the trends with convection, which is the point of that figure.

R3.1-a) There are anthropogenic POA sources that do not have significant amounts of BC. See, for example, Figure 2 of Reff et al. (https://pubs.acs.org/doi/10.1021/es802930x) which indicates large emissions of dust associated with anthropogenic activity (road dust, construction dust) that have high OC relative to BC. While fossil fuel-BC may be a general proxy for anthropogenic activity, is the ratio 1.55 reflective of that general behavior? What does the EPA NEI indicate as FFratio including all anthropogenic PM sources (e.g. Fig 2 of Reff et al.)?

A3.1-a) We thank the reviewer for providing this additional reference. We have included values for the EPA NEI inventory in the supplementary materials (see Table S2). The associated ratios for the traffic and residential combustion sources in the NEI inventory (for traffic = 1.8, and residential = 8.2) are similar to those reported for other inventories and already discussed in the paper (section 3.2). Our ratios do not include emissions associated with fugitive dust from road, tire and construction, which is typically found in larger particles ($D_{aero} > 1$ μm, Zhao et al., 2017).

This is now explained in the manuscript:

*"It should be noted that urban model ratios do not include emissions associated with fugitive dust from road, tire and construction, as those are typically found in larger particles than those studied here (Zhao et al., 2017)."*

*Zhao, G., Chen, Y., Hopke, P.K., Holsen, T.M., Dhaniyala, S.: Characteristics of traffic-induced fugitive dust from unpaved roads, Aerosol Science and Technology, 51:11, 1324-1331, DOI: 10.1080/02786826.2017.1347251, 2017.*

Also see response to R3.1-d. Because we are determining the ratios from urban ambient air measurements, sources that do not emit BC (but do contribute on the submicron range) are also implicitly included.

R3.1-b) Can you plot data (obs or model) as a function of f(BB) influenced to get an intercept at f(BB)=0 and f(BB)=1 for comparison to FFratio and BBratio specified in text?

A3.1-b) Below are the requested quantile plots. On the left we are showing the ratio of the mean OA to mean BC quantiles, while on the right we are using medians. As already shown in Figure S11a and emphasized in A2.5, it highlights for the vast majority of cases along the ATom track,

BC was a very small contributor to particulate carbon. And it also shows, as discussed in the manuscript, that the measured OA/BC in the BB plumes encountered during ATom were actually lower than the global average BB$_{ratio}$ we use in our estimation. These plots have been added to the paper as Fig. S19.

[Figure]

*Figure S19: Distribution of the OA/BC ratio as a function of the fraction of BB influence measured by f(BB)$_{PALMS}$, calculated both as binned averages (left) and binned medians (right) for AToOm-1. Also shown are the OA/BC ratios that we currently assume based on the literature review for both anthropogenic (FF$_{ratio}$) and biomass burning sources (BB$_{ratio}$).*

R3.1-c) For the calculation of observed POA, should there be small amounts of OA associated with sea spray and dust? You have sea salt and dust observations in Figure 2b and elsewhere that could be used to estimate that POA.

A3.1-c) We have added the following text to address this point:

*"The contribution of POA from sea spray is difficult to constrain. As an order-of-magnitude estimate, marine POA is roughly calculated based on preliminary calibrations of OA on mineral dust particles from the PALMS instrument (personal communication K. Froyd). Using this calibration, the average OA by mass on sea salt was <10% for the large majority of MBL sampling (>85%). Since sea salt contributed 4% (11%) of mass in the AMS size range for ATom-1(2) (Figure 2), we estimate that marine POA is on the order of ~1% of aerosol mass in the AMS size range, and possibly much lower. Thus we think that it is reasonable to neglect the contribution of marine POA to this dataset. Future studies will refine this estimate."*

[Figure]

*POA associated with sea-salt particles in the marine boundary layer (<1300m) as reported by the PALMS instrument during ATom-1.*

R3.1-d) Would it be better to label the estimate from equation 1 as "combustion POA"? I would assume models are also more specifically combustion POA?

A3.1-d) It would be inaccurate to label the estimated POA as combustion in the current model outputs as we cannot separate the combustion-emitted POA from other emission sources. This would require adding additional tracers and redoing the model simulations.

For the measurements, it would again be inaccurate to consider the estimated POA as only due to combustion. We have added the following text to the manuscript to clarify this point:

*"The studies used to derive the emission ratio used ambient data in urban air, where all sources mix together and impact the POA/BC ratio, and thus the ratios include the impact of POA sources that may not emit BC."*

R3.1-e) Is non-differential removal of BC and POA appropriate if POA is semivolatile? Consider that near a fire, concentrations could be high enough for IVOCs to be partitioned into the particle. As the airmass is diluted, POA will decrease more rapidly than a conserved tracer. This may explain why the larger BBratio gives POA >100% (line 378).

A3.1-e) This would be true if we were using the directly measured POA. But we are deriving POA from BC, and our method to estimate POA implicitly assumes that POA is non-volatile, and that it does not evaporate (and that also it is not lost to other processes such as photolysis or

heterogeneous oxidation, even though those processes are known to be active in the atmosphere). This is done to obtain an upper limit for the measured POA, so that the key result that POA is too high in the models is reinforced. Therefore, POA > 100% in African plumes cannot be due to POA evaporation (and is instead likely due to uncertainties associated with the use of the global BB average ratios for those specific plumes; in fact the measured OA/BC value is within the combined uncertainty range for tropical forest given by Andreae (2019) 2.9..24.7).

The sensitivity of our results to the semi-volatile nature of POA has already been discussed in A2.3.

R3.1-f) For model estimates of POA, is hydrophilic OC considered POA or SOA? Can models just label the hydrophilic OC as "SOA" and get the right properties for endpoints (health, climate) of interest?

A3.1-f) Primary emitted hydrophilic OC is considered as POA. Indeed, the accepted definition is that for carbon emitted in the particle phase, even if it reacts in the particle phase, remains POA. For SOA to be produced from POA, POA needs to evaporate, and the evaporated organics gases to undergo gas-phase oxidation and then recondence into the particle phase. This is not happening in the model for the hydrophilic POA. This is consistent with earlier studies e.g. Pye and Seinfeld, 2010; Tsigaridis et al., 2014; Pai et al., 2019.

R3.1-g) Did you consider using a CTM to verify the method in equation 1 or back calculate the model effective FFratio? Does it reproduce the model POA (hydrophobic OC)?

A3.1-g) As already explained in the response A2.5, we did not apply this methodology to the model outputs as models do not track separately BC for various sources.

R3.1-h) BC is chemically aged. Do measurement techniques measure BC effectively at all atmospheric lifetimes?

A3.1-h) There is no evidence (that we are aware of, nor supplied by the reviewer) in the literature or in our measurements of refractory BC (such as measured in ATom) undergoing chemical loss to the gas-phase in the atmosphere. However, there are well-known aging effects (including coagulation, condensation, cloud processing) that do change the microphysical arrangement of BC by causing its physical shape to change and associating it with increasing amounts of internally mixed materials. Extensive testing as published in the literature (for example, Cross et al., (2010)) has shown that the measurement technique used in ATom to measure BC concentrations is insensitive to these aging effects. We have added the following text to the manuscript to address this point:

*"Note that BC can physically age but it is not lost in any significant amount to the gas-phase due to chemical processes in the atmosphere."*

*Cross et al., (2010) Soot Particle Studies—Instrument Inter-Comparison—Project Overview, Aerosol Science and Technology, 44:8, 592-611, DOI: 10.1080/02786826.2010.482113.*

R3.1-i) Line 650: The observation-method is not necessarily an upper-bound limit on the fraction of POA as it does not consider OA emitted in sea spray, dust, and may not consider all anthropogenic forms of POA. Consider rephrasing and/or demonstrating it is a limit by adding error bars by using more conservative (higher) FFratios.

A3.1-i) We respectfully disagree with the reviewer's statement. As explained in response A3.1-d above, our method implicitly accounts for all pollution sources of POA, even if they do not emit BC. As documented in response A3.1-c above, the estimated impact of marine POA in the submicron range during ATom is very small, and does not affect our results. Any amount of POA present in dust would be even smaller, as the dust concentrations in the submicron range (Fig. 2) are even smaller than those of sea salt, and the fraction of OA in dust is also very low.

R3.2) Line 355: Can you elaborate on why the PMF doesn't work to separate SOA and POA? When (what timescale, location, altitude, other factor) does the PMF stop working?

A3.2) Extracting linear factors by PMF always "works" in a technical sense, the question is whether it can provide the information that one seeks. This depends on the information content of the dataset and the questions asked. We have updated the text at this location to document this issue in more detail (since we do get this question relatively frequently), as:

*"This approach is not suitable for ATom. To accurately resolve a minor factor such as POA in an AMS dataset, there needs to be a combination of: (a) Sufficient OA mass concentration, so that the signal-to-noise of the spectra is sufficient; (b) Enough fractional mass for the factor to be resolved (>5% in urban areas per Ulbrich et al. (2009), probably a larger fraction at low concentrations such as in ATom); (c) Sufficient spatio-temporal variability ("contrast") in the relative contributions of different factors, since that is part of what PMF uses to extract the factors; (d) Sufficient difference in the spectra of the different factors (for the same reason as (c)), and (e) relatively invariant spectra for each factor across the dataset (as this is a key assumption of the PMF algorithm). As an example of a near ideal case, in Hodshire et al (2019) we extracted MSA by PMF from the ATom-1 data, and were able to match that factor with our independently calibrated MSA species. A very distinct and nearly invariant mass spectrum was measured repeatedly near sources (MBL) (and was mostly absent elsewhere, thus providing strong spatio-temporal contrast) and accounted for about 6% of the fractional mass and 15% of the variance in time. Thus all the conditions were met. For POA, on the other hand, the air sampled in ATom and*

*coming from e.g. Asia has POA and SOA very well mixed, with little change on their relative mass fractions vs. time (as the aircraft flies through that airmass). POA is very low, as documented later in this paper. Atmospheric aging makes the spectra from all OA sources more and more similar as measured by AMS spectra (Jimenez et al., 2009). Thus most of the conditions above are not satisfied for extracting POA by PMF analysis of this dataset."*

*Ulbrich, I.M., M.R. Canagaratna, Q. Zhang, D.R. Worsnop, and J.L. Jimenez. Interpretation of Organic Components from Positive Matrix Factorization of Aerosol Mass Spectrometric Data. Atmospheric Chemistry and Physics , 9, 2891-2918, 2009.*

*Hodshire, A.L., P. Campuzano-Jost, J.K. Kodros, B. Croft, B.A. Nault, J.C. Schroder, J.L. Jimenez, J.R. Pierce. The potential role of methanesulfonic acid (MSA) in aerosol formation and growth and the associated radiative forcings. Atmos. Chem. Phys., 19, 3137-3160, 2019, https://doi.org/10.5194/acp-19-3137-2019*

*Jimenez, J.L., M.R. Canagaratna, N.M. Donahue, A.S.H. Prevot, Q. Zhang, J.H. Kroll, P.F. DeCarlo, J.D. Allan, H. Coe, N.L. Ng, A.C. Aiken, K.D. Docherty, I.M. Ulbrich, A.P. Grieshop, A.L. Robinson, J. Duplissy, J. D. Smith, K.R. Wilson, V.A. Lanz, C. Hueglin, Y.L. Sun, J. Tian, A. Laaksonen, T. Raatikainen, J. Rautiainen, P. Vaattovaara, M. Ehn, M. Kulmala, J.M. Tomlinson, D.R. Collins, M.J. Cubison , E.J. Dunlea, J.A. Huffman, T.B. Onasch, M.R. Alfarra, P.I. Williams, K. Bower, Y. Kondo, J. Schneider, F. Drewnick, S. Borrmann, S. Weimer, K. Demerjian, D. Salcedo, L. Cottrell, R. Griffin, A. Takami, T. Miyoshi, S. Hatakeyama, A. Shimono, J.Y Sun, Y.M. Zhang, K. Dzepina, J.R. Kimmel, D. Sueper, J.T. Jayne, S.C. Herndon, A.M. Trimborn, L.R. Williams, E.C. Wood, C.E. Kolb, A.M. Middlebrook, U. Baltensperger, and D.R. Worsnop. Evolution of Organic Aerosols in the Atmosphere. Science, 326, 1525-1529, 2009. doi: 10.1126/science.1180353.*

R3.3) Perhaps the strengths of DYN could be better isolated/highlighted (e.g. what life-times, OA regimes, POA levels, etc does DYN perform better for?). The abstract statement: "concept of a more dynamic OA. . .with enhanced removal of primary OA, and a stronger production of secondary OA in global models needed to provide a better agreement with observations" could use more support.
Evaluation seems mixed: Figure 8, indicates GC12 and CESM1-CARMA have reasonable POA (while CESM2-DYN has overestimated POA). Line 803 indicates DYN evaluates better, but that effect seems marginal or secondary to other issues in Figure 9 b at least for GC. RMSE in Table 2 also tends to increase when going from base to DYN treatment in CESM2 and GC12. Figure 7 improvements seem mixed.

A3.3) We agree with the reviewer that given the extreme complexity of the dataset and the models, it is difficult to identify the model configuration that works the best all the time. We do not

conclude that DYN is providing better results than other models under all conditions (e.g. CESM2-DYN still has a problem with the POA removal in convection as the fix implemented in CESM-CARMA has not yet been implemented in this version). We have reworded the text in section 4.1 to better describe the model behavior in the NH summer and winter, and indicate more clearly when and where the stronger SOA production provides better results:

*"During the NH summer (ATom-1), models using the VBS parameterization from Pye et al. (2010) tend to underpredict the OA concentrations by 43% for GC12-REF and 33% for CESM1-CARMA for ATom-1, most likely due to the excessive evaporation of the formed SOA in remote regions and low yields for anthropogenic SOA (Schroder et al., 2018; Shah et al., 2019). Models using the VBS parameterization from Hodzic et al. (2016) (CESM2-DYN and GC12-DYN) where OA is less volatile and also OA yields are corrected for wall losses show an improved agreement with observations especially for CESM2-DYN (with NMB of ~5%), and to a lesser extent for GC12-DYN (NMB of ~33%). During the NH winter (ATom-2) characterized by a lower production of SOA, both VBS approaches lead to an overestimation of the predicted OA. This is likely caused by excessively high levels of primary emitted OA as discussed in section 4.4."*

More importantly (not specific to DYN), in this study we have shown as stated in the conclusions (and abstract) that *"the OA system seems to be more dynamic with a need for an enhanced removal of primary OA, and a stronger production of secondary OA in global models to provide a better agreement with observations."* And that is supported by the fact that models that have improved in-cloud removal of POA tend to perform better with regard to POA concentrations and vertical profiles in the upper troposphere than those that don't have it (e.g. CESM1-CARMA with improved in-cloud removal vs. without improved in-cloud removal in Figure 8 and S6; GC12 in Figure 8 and S6 compared to CESM2). We have also shown that making POA semi-volatile instead of non-volatile in model simulations aggravates the model bias in the upper troposphere, and that removal by deep convective clouds and possibly photolysis are needed to address model bias.

The need for a stronger SOA production in models is supported by the fact that models that have correct POA (CESM1-CARMA or GC12) need a stronger production of SOA to match the measured concentrations. The comparison of GC12-REF with two SOA formation mechanisms (using the same removals) in Figure 8 illustrates that a stronger production of SOA would lead to an improved agreement with measurements.

**Minor comments:**

R3.4) Abstract states "OA predictions from AeroCom Phase II...span a factor of 400- 1000. . ." should that be the inter-model variability spans a factor of 400-1000 or the concentrations predicted span that range?

A3.4) The first one. This has already been clarified in response A2.1 above.

R3.5) Did all models assume nonvolatile POA?

A3.5) Yes, see response A2.3.

R3.6) Do OA/OC ratios include consideration of S and N in the form of organosulfates and organonitrates?

A3.6) The following text has been added to the manuscript to address this point:

*"Note that for organosulfates (R-O-SO$_2$H and organonitrates (R-O-NO$_2$, pRONO$_2$ in the following) only one oxygen is included in the reported OA/OC, as the fragments of these species are typically the same as for inorganic species in the AMS (Farmer et al., 2010). However in ATom organosulfates are estimated to account for ~1% of the total sulfate (based on PALMS data, see Liao et al., 2015 for the methodology). Since sulfate and OA concentrations are comparable, organosulfates would only increase the OA/OC by ~1% on average. Organonitrates are reported from the AMS for ATom. Their impact on OA/OC is not propagated for the default values, to maintain consistency with a large set of OA/OC measurements by AMS in the literature, and since they would increase OA/OC on average by only 4.5% (ATom-1) and 2.2% (ATom-2), which is smaller than the uncertainty of this measurement. However, we show the results with both methods in Fig. 5 to fully document this topic.*

*Farmer, D.K., A. Matsunaga, K.S. Docherty, J.D. Surratt, J.H. Seinfeld, P.J. Ziemann, and J.L. Jimenez. Response of an Aerosol Mass Spectrometer to Organonitrates and Organosulfates and implications for Atmospheric Chemistry. Proceedings of the National Academy of Sciences of the USA, 107, 6670-6675, doi: 10.1073/pnas.0912340107, 2010.*

*Liao, J., K.D. Froyd, D.M. Murphy, F.N. Keutsch, G. Yu, P.O.Wennberg, J.St. Clair, J.D. Crounse, A. Wisthaler, T. Mikoviny, T.B. Ryerson, I.B. Pollack, J. Peischl, J.L. Jimenez, P. Campuzano Jost, D.A. Day, B.E. Anderson, L.D. Ziemba, D.R. Blake, S. Meinardi, G. Diskin. Airborne organosulfates measurements over the continental US. Journal of Geophysical Research-Atmospheres, 120, 2990–3005, doi:10.1002/2014JD022378, 2015.*

[Figure]

*Caption Fig. 5: Distribution of the OA / OC ratio as measured during ATom-1 and -2. Also included (in dashed lines) are the distributions of OA/OC values that included the contribution of organic nitrates (pRONO₂). Values for the recent aircraft campaigns (SEAC4RS, DC3 and WINTER) that took place over continental US regions closer to continental source regions are also shown (Schroder et al., 2018). The bars (right axis) show the OA/OC used for SOA and POA by the models included in the AeroCom and ATom ensemble, with OA/OC=1.4 being the modal value for the former and 1.8 for the latter.*

R3.7) At times, it was a bit confusing if the AeroCom-II ensemble referred to circa 2014 models or the actual AeroCom II results paired with measurements. On line 103, the "AeroCom results" could be clarified as "AeroCom models" or "models used in AeroCom." Line 277 might be better as "monthly average results of 28 global models" instead of "results of 28 global models."

A3.7) We have modified the text as suggested by the reviewer: "*Our own analyses of the AeroCom-II models..*" and "*We consider the monthly average results of 28 global models,..*".

R3.8) Figure 1: Is "author affiliations" supposed to be in the caption?

A3.8) We have added *"Model contributor affiliations"* to the figure label and caption. See also response A1.3.

R3.9) Line 163: While marine production of OA may be smaller than continental production, it's possible marine production contributes more to concentrations over the oceans than it's global production rate would suggest. I recommend rewording sentence. What concentrations of SOA are predicted for oceanic isoprene sources?

A3.9) SOA production from oceanic isoprene sources is not included as already indicated in the manuscript.

As suggested by the reviewer, we have modified the sentence to read: "*None of the models includes the marine production of OA which is estimated to be ~3 orders of magnitude smaller than the continental production of OA from both isoprene and monoterpene precursors (Kim et al., 2017), but could be important in the MBL.*"

See also the response to A3.1-c above.

R3.10) How does averaging data allow below detection values to be used?

A3.10) This seems to be a very widespread issue with modelers' understanding of our measurements. We have included the following text to section 3.3. to clarify and document this point:

"*Per standard statistics, the precision of a measurement decreases (i.e., gets better) with the square root of the number of points (or time interval) sampled. I.e. the precision of an average can be approximated by the standard error of the mean ($\sigma/sqrt(n)$, where n is the number of measurements averaged), and it is better than the precision of the individual data points ($\sigma$). This also applies to the detection limit, since it is just 3 times the precision. Note that a detection limit is not meaningful unless the averaging time is specified. For example, let's assume that the detection limit is 20 ng m$^{-3}$ (1-second), and the data points over 60 consecutive seconds are all 10 ng m$^{-3}$. All 1-second measurements are below the 1-second DL. However the average (10 ng m$^{-3}$) is now above the DL for 1-minute averages, which is $20/sqrt(60) = 2.6$ ng m$^{-3}$.*"

R3.11) Line 605: qualify the OA/OC of 1.4 as "fossil fuel" combustion as biomass combustion tends to have OA/OC of 1.7.

A3.11) We have added this qualifier as requested: "*A low OA/OC ratio is indicative of freshly emitted OA from fossil fuel combustion (typically ~1.4),..*"

---

## Author Response (AR2)

**Responses to Round 2 of Review for Hodzic et al. ACPD**

The authors have done considerably work to address the points raised in review, substantially improving the manuscript. A few points remain to be addressed prior to publication:

We thank the reviewer for the appreciation of the work that went into the previous revision. We have modified the manuscript further to address these additional comments.

R2.1. A2.3: A likely explanation for the overestimate of POA in models (when treated as NV) is that emissions of POA were likely overestimated (i.e. SV emissions reported as POA when estimated under high loading conditions, Robinson et al., 2007), so one possible solution for these models is the need to re-balance this (downgrade POA emissions, increase SV SOA sources). It would be nice to see this history (i.e. the incorrect assumption that emissions were all NV) acknowledged in Section 4.4.

We have modified the text in L806 to read:

"*These seasonal differences suggest that model errors could be partially due to inefficient production of SOA and/or too high POA emissions, although removal errors also probably play a major role (see next section).*"

R2.2. A2.5: The updated tables are clearer, but given that these appear in SI, I suggest that the authors include in the text either the standard deviation or range of values averaged to obtain 1.5 and 1.48 in lines 1950 and 1952.

Neither the manuscript (ending in L1713) nor the SI (ending in L147) have lines 1950 or 1952. Perhaps the reviewer is referring to lines in the posted response to R2, but that file does not have line numbers. In any case, the only mention of "1.48" is on L422, while 1.5 is mentioned two lines before in L420. Therefore we assume the reviewer is referring to that section of the text, which already contains a discussion of the range of ratios shown in the table (L422): "*Mobile source measurements in general exhibit lower ratios (POA/OA ratio 0.5-1.5) while COA determination typically ranges from 2 to 3.*"

For the sake of clarity, we have modified this text to include the standard deviations and modes, and rephrased the last sentence to make the ranges we are discussing more apparent:

"*Based on Table S1 data, we assume POA to be co-emitted with BC for anthropogenic fossil fuel / urban region POA (herein called FFratio for simplicity, even though much of it is non-fossil, Zotter et al., 2014; Hayes et al., 2015) at a ratio of 1.5±0.82 (average ±1σ of all*

*urban ambient air studies that report POA and BC for best intercomparability to the ATom*
*dataset; including all urban studies results in a very similar number, 1.48±0.65, median: 1.41).*
*Measurements where mobile sources are the main contributor in general exhibit lower ratios*
*(POA/OA ratio 0.5-1.5), while studies with strong COA contributions typically ranges from 2*
*to 3."*

R2.3. A2.5: It remains unclear exactly how equation 1 was applied to the data. I believe that the
authors applied the average POA/BC EFs that were discussed in Section 3.2 to ALL data, but the
text "an emission ratio appropriate to the airmass origin.." on L2301 seems to suggest that there
is some sort of back trajectory to identify the origin of air masses applied to determine the EF.
Please clarify this process in the text.

Line 370 of the revised manuscript reads: *"For a particular airmass, the mass fraction of*
*biomass burning (BB) aerosol reported by the PALMS instrument f(BB)$_{PALMS}$ (Thompson and*
*Murphy, 2000; Froyd et al., 2019) was then used to evaluate the degree of BB influence."*

And then again when stating Eq (1) (Line 440):
*"The PALMS determined mass fraction of biomass impacted aerosol (f(BB)$_{PALMS}$) can then*
*be used to determine a total POA contribution from both types of sources"*

They both make clear that f(BB)$_{PALMS}$ is used as a proxy for airmass origin. Furthermore, the
sentence that the reviewer refers to (L761, not L2301) states this again:

*"POA concentrations were estimated from the BC measurements by using an emission ratio*
*appropriate to the airmass origin (biomass burning vs. anthropogenic), and using the f(BB) mass*
*fraction from the PALMS single particle instrument (see Section 3.2)."*

We have modified the last sentence to make this clearer:

*"POA concentrations were estimated from the BC measurements by using an emission ratio*
*appropriate to the airmass origin (biomass burning vs. anthropogenic), as quantified by the*
*f(BB) mass fraction from the PALMS single particle instrument (see Section 3.2), with*
*f(BB)=1 taken as a BC and OA being of pure BB airmass origin and f(BB)=0 exclusively from*
*a non-biomass burning source."*

R2.4. A2.5: The authors state that they could not perform a similar analysis on model output to
test the approach of equation 1 because they do not have model output for BC from various
emission sectors – I presume that what they mean here is that they would need fBB from the

model. It seems that the authors could have chosen to output this information as part of the SV-POA simulation that they performed during the review process. Given that the authors have not verified their method with model data, they should explicitly acknowledge that the validity of their analysis approach has not been tested with simulated data, and that future work should verify this approach.

The suggested simulation is of interest, but would have required much additional work as one (or several) new BC species would have to be added to the model and modified in multiple locations (emissions, transport, convection, wet and dry deposition etc.). And emission fields would have to be broken down by source. The model would then have to be tested to gain confidence on the simulation and make sure errors were not introduced from the extensive code modifications. Therefore this is a considerable amount of work, that would be justified for a manuscript on source apportionment of BC, but not for the revision of an OA manuscript that already includes extensive experimental data and output from many models. On the other hand, running a case with another standard GEOS-Chem configuration (as for the SV-POA case) was straightforward as it only required changing the model configuration and input files.

In addition, the models have a more accurate internal method to track POA and SOA by applying mass conservation within the model. The measurement-based estimate is well-supported in the revised manuscript, including extensive sensitivity studies, which shows that its conclusions are robust against the uncertainties. It does not seem necessary (or too useful) to us to simulate the measurement-based method with a model. Therefore we refrain from recommending this research.

R2.5. A2.6: The authors have not sufficiently described the inorganic aerosol simulations in the model description section (2.1), including a description of relevant thermodynamic schemes (highlighting which models did not include nitrate and ammonium) and emissions input and schemes (i.e. sea salt and dust). Without such a description, the results of Section 4.6 should not be included as the reader has not been provided sufficient basis for assessing these schemes.

References have already been provided that describe each of the models in section 2.1. We have also indicated which inorganic species are included in each model, and it is clear whether a specific model has nitrates. There is still high value in including an overall comparison including all the species for reference, even if the details of their simulation in all the models cannot be described in great detail, as this provides important context for the relative importance of OA (the main topic of this paper) and other species, and the fact that there is high variability in the simulations of other components as well. The following text provides key references for an interested reader:

*"Specific studies have discussed and continue to investigate the ATom measurements and simulations of different components in more detail, including particle number (Williamson et al., 2019), black carbon (Katich et al., 2018; Ditas et al., 2019), MSA (Hodshire et al., 2019), sulfate-nitrate-ammonium (Nault et al., 2019), and sea salt (Yu et al, 2019; Bian et al., 2019; Murphy et al., 2019)."*

R2.6. A2.16: Are the authors suggesting that the spatial distribution and observed OA magnitudes are the same between ATom-1 and ATom-2?? Visual inspection of Figure 2 and S1 suggest that these are quite different, and thus the original review comment that Figure S1 should be included in the main text should be re-considered.

There are many similarities between the spatial distributions, e.g. a clean Southern hemisphere, high concentrations in the lower equatorial Atlantic, more pollution in the Northern hemisphere etc. Of course there are also some differences. However the data are already shown in the paper in arguably more useful ways (Fig. 2b, 2e, and 4) and Fig. S1 is easily accessible for anyone interested. We therefore prefer to keep Fig. S1 in the supplementary information.

R2.7. A2.13: Could the authors explain the logic of why photolytic removal is not included for isoprene-SOA in the text?

For isoprene SOA formed through explicit chemistry and heterogeneous uptake in aqueous aerosols, we follow the implementation of Marais et al. (2016) in GEOS-chem, where the photolytic removal is not included. Further research is needed to quantify how efficient photolytic removal for these isoprene SOA species is. This is now mentioned in the revised manuscript:

*"As in Hodzic et al. (2016) the GC12-DYN model version includes updated VBS SOA parameterization, updated dry and wet removal of organic vapors, and photolytic removal of SOA (except for isoprene-SOA that is formed in aqueous aerosols, where we follow Marais et al. 2016)."*

R2.8. A2.21: the text "correlates quite well with other gas-phase BB tracers" is not justified in the current manuscript. Please provide plots for the SI, or a reference that shows this (or remove this statement).

A comprehensive description of f(BB) is currently under review on a separate manuscript (in which the main authors of this paper are not coauthors), so we cannot include it here. In lieu of it, we have added the following plots to the SI that shows the correlation:

[Figure]

Figure S20: Correlation of f(BB) from the PALMS instrument with colocated, well characterized gas phase tracers for BB for the full ATom mission (1-4). HCN (top panel) provided by the Caltech CIMS instrument, $CH_3CN$ (bottom panel) provided by the NCAR TOGA GC-IE instrument (Wofsy et al, 2018).

We are also referring to this Figure now when discussing f(BB)PALMS (Line 371):

*"This parameter correlates well with other gas-phase BB tracers (Figure S20), and is more useful as a particle tracer since its lifetime follows that of the particles."*

R2.9. Line 1934: "documented later" suggests observational confirmation, whereas the POA estimate here is a derived quantity with substantial uncertainties. Recommend this language be modified to "suggested by later analysis.."

Again L1934 does not exist in the manuscript. Those words do appear in L404. We respectfully disagree with the reviewer, and have shown that the low POA is strongly supported by the evidence, as documented extensively in the previous round of responses and the revised manuscript. Therefore we prefer to keep this text as it is.

R2.10. Line 1965: Please discuss in the text how the new numbers from Andreae 2019 compares to other numbers from literature shown in Table S1.

Table S1 only includes urban-focused studies. While three of these did include BB sources (and are listed as such in the table), we see no value in comparing this very small subsample with a review that includes these 3 studies and about 200 more. Especially since, as mentioned in the paper text and illustrated in Figures S9 and S19, the exact value of $BB_{ratio}$ has virtually no impact on the conclusions of our analysis over the range of values reported in Andreae 2019 (which again, includes the studies in Table S1).

Moreover, we do compare Andreae 2019 with the model emission inventories (Table S2), which in our view is the relevant comparison (L447):

*"On the other hand, for biomass burning, the emission inventories ratios range from ~5 for crop, to ~15 for forest, and up to ~50 for peatland. While generally consistent with the values discussed by Andreae (2019), they are on the lower end of the ranges discussed in that work. The averages and ranges of the measurement and model ratios are similar, and thus no significant model bias on the ratios is apparent"*

Therefore we prefer to keep this text as it is and not discuss Table S1 further.